# RBM39 degrader invigorates innate immunity to eradicate neuroblastoma despite cancer cell plasticity

The cellular plasticity of neuroblastoma is defined by a mixture of two major cell states, adrenergic and mesenchymal, which may contribute to therapy resistance. However, how neuroblastoma cells switch cellular states during therapy remains largely unknown, and how to eradicate neuroblastoma regardless of its cell state is a clinical challenge. To better understand the cellular plasticity of neuroblastoma in chemoresistance, we define the transcriptomic and epigenetic map of adrenergic and mesenchymal types of neuroblastomas using human and murine models treated with indisulam, a selective RBM39 degrader. We show that cancer cells not only undergo a bidirectional switch between adrenergic and mesenchymal states, but also acquire additional cellular states, reminiscent of the developmental pliancy of neural crest cells. These cell state alterations are coupled with epigenetic reprogramming and dependency switching of cell state–specific transcription factors, epigenetic modifiers, and targetable kinases. Through targeting RNA splicing, indisulam induces an inflammatory tumor microenvironment and enhances the anticancer activity of natural killer cells. The combination of indisulam with anti-GD2 immunotherapy results in a durable, complete response in high-risk transgenic neuroblastoma models, providing an innovative, rational therapeutic approach to eradicate tumor cells regardless of their potential to switch cell states.

Pediatric cancers give rise to unique clinical challenges. First, the number of potentially "druggable" targets with specific and selective inhibitors remains low[1,2]. Second, the alterations of cancer signaling pathways have not proven to be viable therapeutic targets[3–6]. Third, the low mutation burden of pediatric tumors leads to a low spectrum of neoantigens[7], which may greatly limit the effect of immunotherapy, including immune checkpoint inhibitors. Molecular and cellular heterogeneity within a tumor causes further challenges in designing and selecting effective therapies, and curtailing treatment resistance[8]. Using neuroblastoma as an example, this study provides evidence that heterogeneous tumor cells can be eliminated by harnessing the power of the immune system through targeting the RNA splicing factor, RBM39.

Arising in early fetal development[9], neuroblastoma is an extracranial, embryonal tumor derived from the neural crest lineage, transformed by *MYC* oncogenes[10–13]. 50% of high-risk neuroblastomas have *MYCN* amplification, while the other half express high levels of *C-MYC*. With current intensive multimodal therapies (combined cytotoxic chemotherapies, surgery, stem cell transplantation, radiotherapy, differentiating agents and anti-GD2–based immunotherapy), 5-year survival rates for patients with high-risk neuroblastoma remain less than 50%[14–17]. In addition, survivors of high-risk disease have a significant risk of developing long-term side effects, including subsequent malignant neoplasms due to exposure to cytotoxic chemotherapy and radiotherapy[18,19]. Unfortunately, developing safer and more effective targeted therapies against high-risk neuroblastoma has

✉ e-mail: Jun.Yang2@stjude.org

been challenging due to the low incidence of targetable recurrent mutations[20–22], although a small fraction of patients with ALK mutations are highly responsive to ALK inhibitors[23]. While functional genomic screens have identified many essential survival genes in neuroblastoma cells[24–29], translating these discoveries into effective therapies has been technically challenging because most of these essential genes, including MYC, are considered to be "undruggable". To address the unmet clinical need for high-risk neuroblastoma therapy, we recently identified a therapeutic target, RBM39, a MYC target that regulates pre-mRNA splicing and appears to be essential to neuroblastoma survival[30]. We and others further found that indisulam, a "molecular glue" drug that selectively recruits RBM39 to the CRL4-DCAF15 E3 ubiquitin ligase (DCAF15) for proteasomal degradation[31,32], is highly effective against neuroblastoma by inducing a wide range of splicing anomalies that affect a number of essential genes[30,33]. It is known that MYC−driven neuroblastoma has deregulated splicing[34–36]. However, high-risk, patient-derived xenograft (PDX) models implanted in immunodeficient mice eventually relapse when treated with this RBM39 degrader, prompting us to investigate the mechanism of therapy resistance of neuroblastoma to indisulam treatment.

Lineage plasticity[37] (or transdifferentiation[38], pathway indifference[39]) is one key mechanism of drug resistance by which cancer cells acquire an alternative cellular state in response to treatment to sustain their survival[37–41]. Neuroblastoma cell lines contain morphological variants that contribute to cell state heterogeneity[42–44], with cells residing in one of two major cellular states, committed adrenergic (ADRN) and neural crest migratory or mesenchymal (MES), largely defined by epigenetic and transcriptomic programs[45–47]. These distinct populations exhibit differential responses to chemotherapy[46]. While there is a theory that MES and ADRN states are interconvertible, consequently leading to therapy resistance, this is largely supported by forced overexpression or genetic deletion of transcriptional factors in cell lines[46,48,49]. Whether interconversion of these cell states occurs in response to tumor environmental changes such as exposure to therapeutic agents is still not very clear, albeit these studies indicate that the resistant tumor cells acquired MES gene features. Neuroblastoma results from differentiation arrest of neural crest−derived sympathoadrenal progenitor cells[10,50]. Recent single-cell RNA-seq and lineage tracing studies show transcriptomic similarity between neuroblastoma and normal cell types along the developmental trajectory of neural crest cells[51–54]. The differentiation status of human adrenal medulla at different developmental time points is associated with neuroblastoma outcome risk[55]. Unlike cell lines, which can be clearly defined by ADRN and MES signatures, the in vivo cellular state of neuroblastoma cells is less clear, although most resemble an ADRN state[51,54,56]. However, malignant cells of high-risk neuroblastomas show increased MES signatures and reduced ADRN signatures in one study[52]. Other studies further showed that ADRN tumors with MES features have molecular traits of Schwann cell precursors (SCP), bridge cells, and early neuroblasts (all of which are neural crest cell progeny)[51,52], indicating intratumoral plasticity in those tumors. However, the path from one cellular state of neuroblastoma to another (or others) during in vivo therapy has not been defined. Our study addresses one key knowledge gap in therapy resistance: how cellular plasticity of cancer cells confers drug resistance in neuroblastoma. Addressing this question is a key step towards developing more effective combination therapies against refractory malignancies.

Here, by using multiple high-risk neuroblastoma models that developed resistance to indisulam, we show that cancer cells undergo a multi-directional switch of cell states, reminiscent of the cellular pliancy of neural crest cells. The cell state switch of neuroblastoma cells is associated with epigenetic changes, and a dependency switch of cell state−specific transcription factors, epigenetic modifiers, and targetable kinases. Notably, indisulam treatment induces an inflammatory tumor microenvironment characterized by infiltration of immune cells such as T and NK cells. In the NK cell competent mice that are deficient in adaptive immunity (T cells and B cells), implanted c-MYC−driven murine neuroblastomas and human neuroblastoma xenograft models are completely eradicated by indisulam, suggesting that NK cells or innate immunity may play a critical role in indisulam-mediated anticancer activity. Complete and durable responses are also achieved in an immunocompetent MYCN/ALK^F1178L neuroblastoma mouse model and two different MYCN syngeneic models when treated with the combination of indisulam and anti-GD2 mAb. It is known that anti-GD2 mAb exerts its antitumor effect through NK cell-mediated ADCC (antibody-dependent cell−mediated cytotoxicity)[57]. Our study indicates that targeting RBM39 in combination with anti-GD2 therapy may eradicate neuroblastoma cells irrespective of their cell state switching potential, providing a rationale to combine indisulam and anti-GD2 mAb as a therapy for high-risk neuroblastoma patients.

## Results

### Indisulam resistance in a transgenic *MYCN/ALK^F1178L* mouse model is associated with cell state switch to MES and Schwann cell precursor phenotypes

Human *ALK^F1174L* (analogous to mouse *ALK^F1178L*) is the most frequent somatic mutation in neuroblastoma and is associated with *MYCN* amplification, conferring a worse prognosis than *MYCN* amplification alone[58]. Our previous study showed that indisulam treatment of *MYCN/ALK^F1178L* mice with a regular dosing (25 mg/kg, 5 days on/2 days off for two weeks) led to durable, complete responses even when tumor sizes reached over 1000 mm3 prior to treatment initiation[30]. When testing the dosing schedule by treating the *MYCN/ALK^F1178L* mice with only one dose of indisulam (25 mg/kg) per week for 3 weeks in our current study, we found variable responses (Fig. 1a). Two weeks after therapy discontinuation, we resumed the regular dosing schedule for an additional two weeks. Among the 5 treated mice, three showed complete and durable responses, one responded to the first dose of therapy (tumor volume reduced from 1100 mm³ to 300 mm³) but then developed resistance to the following treatment (Fig. 1a), while another one showed a complete response but eventually relapsed after second therapy discontinuation (Fig. 1a, b). To understand the mechanism of therapy resistance, we performed RNA-seq analysis followed by gene set enrichment analysis (GSEA) to compare the naïve tumors with the relapsed and resistant tumors (Fig. 1c). In comparison with the naïve tumors that showed a high "E2F" gene signature, indicative of the high proliferation rate of naïve tumor cells, the "human 20q11 amplicon" was the top upregulated gene signature in the resistant/relapsed tumors (Fig. 1c, d). Human *RBM39* is located in 20q11. This was consistent with the high levels of RNA-seq reads of *Rbm39* in the relapsed and resistant tumors (log2 fold change = 0.73), which is located in Chr2qC of mouse genome (Fig. 1e). Surprisingly, the RBM39 protein was actually downregulated (Fig. 1f), which was in contrast to its mRNA expression. These data suggest that cells strived to survive by producing more mRNAs to compensate for the RBM39 protein degradation. This was further verified by the correlation of indisulam resistance with the RBM39 copy number and mRNA expression levels across 534 cell lines in DepMap (depmap.org) (Fig. 1g, h).

Additionally, we noticed that the relapsed/resistant tumor cells acquired a high "neural crest stem cells" gene signature (Fig. 1c,i), suggesting that these therapy-resistant cancer cells underwent de-differentiation. We then examined the ADRN and MES signatures[46] and found that the relapsed/resistant cells exhibited a significant shift to the MES state (Fig. 1j). Particularly, we also found that the relapsed/resistant tumor cells expressed a signature of "Schwann cell precursors (SCP)"[51,52], characterized by high expression of *S100b*, *Sox10*, *Plp1* and other genes (Fig. 1k). Then we colored the

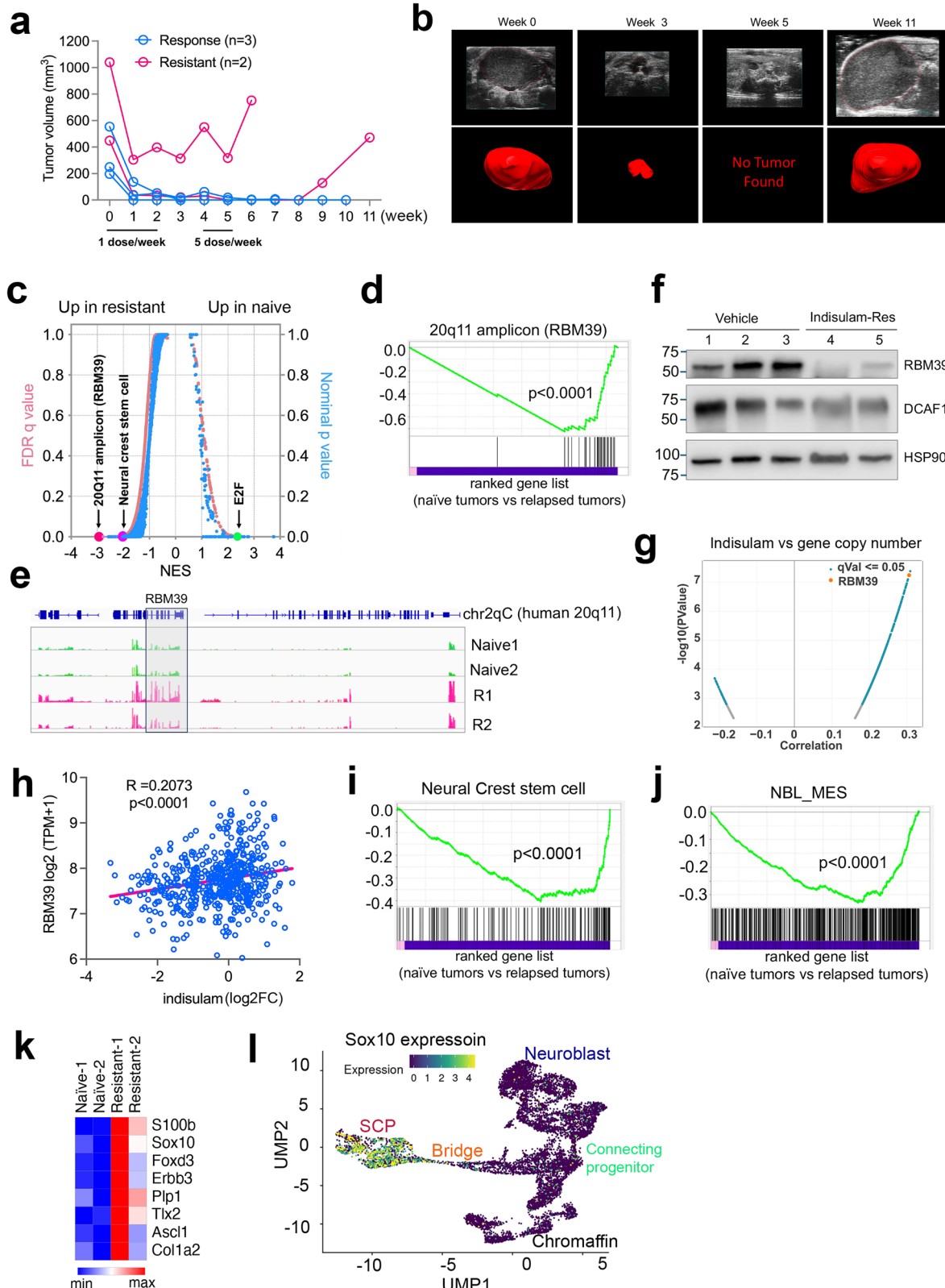

UMAP projection by the expression of *Sox10*, the SCP transcription factor, into the single cell analysis of human adrenal medulla, which demonstrated high levels of *SOX10* expression in SCPs but not in chromaffin and neuroblast cells (Fig.1l), the downstream progenies of SCP and origin of neuroblastoma. We further examined the SCP signature, which was highly expressed in the resistant/relapsed *Th-MYCN/ALK*[F1178L] tumors, in two human neuroblastoma cohorts

(Target NBL[21] and St Jude PCGP[59]). The SCP markers such as *TLX3*, *SOX10*, *S100b*, and *FOXD3* appeared to be enriched in non-*MYCN* amplified tumors (Supplementary Fig. 1a, b), suggesting that the resistant/relapsed *MYCN*-driven tumors acquired gene features similar to the non-*MYCN* amplified tumors. Taken together, these data indicate that the resistant/relapsed neuroblastomas acquired new cell state properties of neural crest progenies.

**Fig. 1 | Indisulam resistance in transgenic *MYCN/ALK^F1178L* mouse models is associated with lineage switch to cell states with MES and Schwan precursor cells. a** Tumor growth curve for *MYCN/ALK^F1178L* mice treated with 25 mg/kg of indisulam. Blue indicates sustained complete response (biological replicates *n* = 3). Pink indicates indisulam resistance and relapse (biological replicates *n* = 2). **b** Ultrasound imaging of tumor relapse. Top panel shows the two-dimensional ultrasound images over time and the bottom panel shows the volume reconstructions. **c** GSEA plot analysis of naïve (biological replicates *n* = 2) vs relapsed tumors (biological replicates *n* = 2). The normalized RNA read counts are transformed to log2 counts per million reads (log2CPM) for GSEA analysis. Right y-axis =one-sided, nominal *p*-values. Left y-axis = false discovery rate (FDR) *q* values. X-axis = normalized enrichment score (NES). **d** GSEA shows "20q11 amplicon" gene set upregulated in indisulam-resistant tumors. Nominal *p*-value < 0.0001 calculated by one-sided Fisher's exact test. **e** The normalized RNA-seq reads at *Rbm39* gene locus by IGV program. **f** Western blot analysis for vehicle-treated tumors (biological replicates *n* = 3) and indisulam-resistant tumors (biological replicates *n* = 2). The experiment was done once. **g** Pearson correlation between gene copy number vs indisulam response from DepMAP data (www.depmap.org). *p*-value (left y-axis, two-sided) calculated using t-distribution based on the Pearson correlation coefficient (x-axis). The z-score calculated by permutation test. **h** Pearson correlation between gene expression vs indisulam response from DepMAP data (www.depmap.org). The *p* value is calculated using two-sided t-distribution. Y-axis = expression of *RBM39* with log2TMP plus 1 (TPM + 1). X-axis = log2 fold change of indisulam in cell viability measurement. GSEA shows "neural stem cell" (**i**) and "NBL_MES" (**j**) gene sets significantly upregulated in resistant tumors. Nominal *p* value < 0.0001 calculated by one-sided Fisher's exact test. **k** Heatmap shows the expression of Schwan cell progenitor (SCP) markers in naïve (biological replicates *n* = 2) vs. relapsed tumors (biological replicates *n* = 2). The Morpheus program is used to generate z-score-based interactive heatmap after log2-transformed expression data. **l** *Sox10* labels the SCP cell population from the study[9]. Source data are provided as a Source Data file.

## Indisulam resistance is associated with a cell state switch from ADRN to MES and melanocytic state in human *MYCN*-amplified neuroblastomas

To further understand the cell state alterations in neuroblastoma resistance, we treated *MYCN*-amplified patient-derived xenografts (SJNB14) implanted subcutaneously into CB17/SCID mice with indisulam (25 mg/kg, 5 days on/2 days off for two weeks). In comparison to the vehicle control group, all five tumors treated with indisulam had complete responses, but eventually they relapsed. However, these tumors remained responsive to indisulam treatment until the eighth cycle of therapy when tumors developed full resistance (Fig. 2a). RNA-seq followed by GSEA showed that the resistant tumors had a significant reduction of the "ADRN" signature, followed by a "hypoxia" signature (Fig. 2b, c). However, the resistant tumors acquired a high "interferon alpha and beta" signature (Fig. 2d), in line with the feature of MES neuroblastoma cells, which were reported to express high levels of interferon pathway genes[60,61]. The master transcriptional factors of MES state cells, such as *c-MYC*, *PRRX1* and *NOTCH1* were highly upregulated in the resistant tumor cells, together with the SCP marker *S100B* and stem cell markers *KIT* and *SALL4* (Fig. 2e). Wnt ligands were also highly upregulated in the resistant tumors. A recent study based on network analysis of neuroblastoma expression data proposed that Wnt signaling is a major determinant of regulatory networks that underlie mesenchymal/neural crest cell-like cell identities through PRRX1 and YAP/TAZ transcription factors[62]. Surprisingly, melanocytic markers, including *MITF*, which encodes the master transcriptional factor of melanocytes and melanomas, were also highly upregulated. It is known that SCPs are melanocyte progenitors[63]. The scRNA-seq analysis in the developmental trajectories of SCP to chromaffin cells and neuroblasts showed no expression of *MITF* (Supplementary Fig. 2a), supporting the hypothesis that neuroblastoma cells either directly transdifferentiated to a melanocyte-like state or dedifferentiated to SCPs which then differentiated to a state with melanocytic features. We further examined the expression of Wnt ligands and melanocytic gene signatures in human neuroblastoma cohorts. The Wnt ligand signature was enriched in neuroblastomas without *MYCN* amplification in both the TARGET NBL and St Jude PCGP cohorts[21,59] (Fig. 2f; Supplementary 2b). Interestingly, the melanocytic gene signature was enriched in TARGET neuroblastomas without *MYCN* amplification but not in St Jude PCGP data (Fig. 2f; Supplementary 2b), probably due to the confounding factors such as different stages, disease risk classification, and pre-treatment and post-treatment from both cohorts. It was known that the TARGET NBL dataset was enriched with high-stage and high-risk neuroblastomas[21]. Distinct from the *MYCN/ALK^F1178L* resistant murine tumors, the *RBM39* expression showed no great induction in the resistant SJNB14 tumors (Fig. 2g), and Sanger DNA sequencing showed no mutations in the indisulam binding motif (Supplementary Fig. 2c), suggesting that the cell state switch plays a major role in mediating the indisulam therapy resistance in this *MYCN*-amplified PDX model.

Similarly, mice implanted with SIMA (with MYCN amplification) cell line–based xenografts also showed excellent outcomes after the completion of a 10-day treatment, but the neuroblastoma eventually relapsed (Supplementary Fig. 3a). To test if the relapsed SIMA tumors remained sensitive to indisulam, we resumed treatment after tumor recurrence. Like SJNB14 PDX, SIMA xenografts had 100% complete response after two additional repeated treatment cycles (Supplementary Fig. 3a). Western blot analysis of xenografts harvested at 24-hours after the final dosing from the third cycle of treatment confirmed that RBM39 expression was nearly completely abrogated in the treatment group (Supplementary Fig. 3b). RNA splicing analysis of tumor xenografts showed that indisulam induced very similar events to those induced by RBM39 knockdown and in vitro indisulam treatment, including genome-wide splicing anomalies in skipped exons[30] (Supplementary Fig. 3c). RNA-seq and GSEA analysis revealed that tumor cells displayed upregulated gene signatures of neural crest migration and type I interferons (Supplementary Fig. 3d), although the changes in ADRN and MES signatures were not significant. Nevertheless, manual examination of master transcriptional factors of ADRN and MES revealed that the tumors treated with indisulam expressed higher levels of *C-MYC*, *PRRX1*, and *FOSL2* but reduced levels of *PHOX2A*, *PHOX2B*, and *MYCN* (Supplementary Fig. 3e), indicating that the cell state underwent a transition from ADRN to MES during the indisulam therapy. Like SJNB14, a significant induction of melanocytic markers and Wnt ligands was observed in the SIMA tumors treated with indisulam (Supplementary Fig. 3d, e).

Changes in neuroblastoma cell states are associated with epigenetic reprogramming[45,46,48]. To test if this occurred in the indisulam-resistant tumors, we performed Cleavage Under Targets and Tagmentation (CUT&Tag) to map the genome-wide H3K27Ac in SJNB14 naïve versus resistant tumors. H3K27Ac is an epigenetic mark indicating active promoters and enhancers in gene transcription. We identified 8580 differential H3K27Ac peaks (Fig. 2h). The enrichment of H3K27Ac at *PHOX2B* (a master transcription factor in ADRN) locus was greatly reduced in the resistant tumors, while the H3K27Ac levels were upregulated at the locus of *S100B* (a marker of SCP and MES) (Fig. 2i). S100B is also a melanoma marker[64]. We also found that H3K27Ac peaks at the *TWIST1* locus were among the top downregulated in resistant tumors (Fig. 2h, i), in line with its reduction in mRNA levels (Fig. 2e). In neuroblastoma, TWIST1 co-occupies enhancers with MYCN and is required for MYCN–dependent proliferation[65]. SJNB14 is a *MYCN*-amplified tumor. Motif analysis for the H3K27Ac peaks revealed that transcription factors, including those determining MES state (JUN-AP1, FOSL2) could bind at the H3K27Ac loci in both naïve and resistant tumors (Supplementary Fig. 4a). Then, we performed GSEA analysis for the genes with altered H3K27Ac peaks.

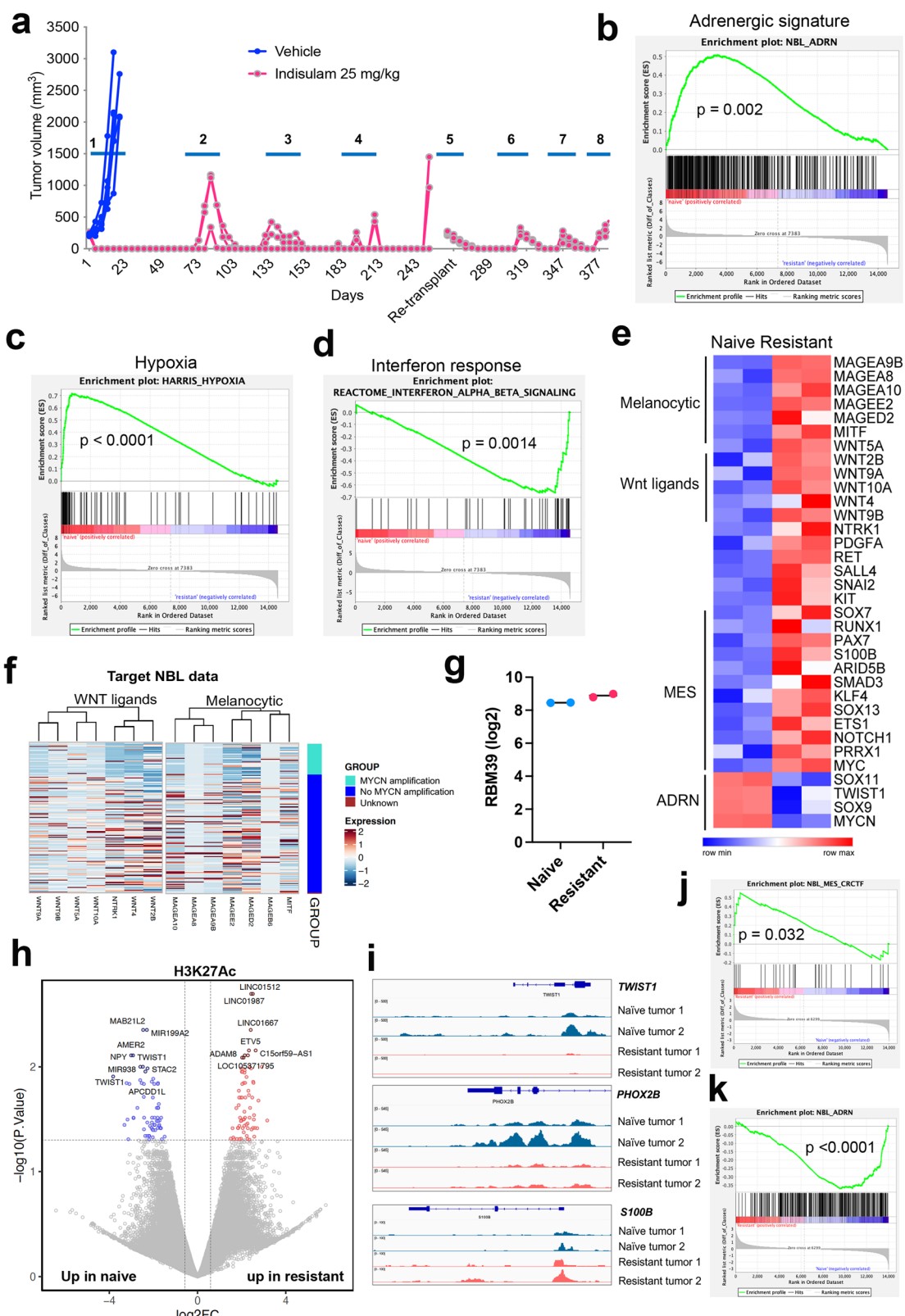

Consistent with the RNA-seq results, the indisulam-resistant tumor cells showed a significant enrichment of H3K27Ac at gene loci of MES transcriptional factors (i.e., SMAD3, TEAD4, MYC, Fig. 2j; Supplementary Fig. 4d), mesenchymal genes such as those involved in muscle contraction, HIPPO (YAP/TAZ) signaling pathway, and interferon alpha and beta signaling (Supplementary Fig. 4b). However, there was a significant downregulation of H3K27Ac peaks at the genomic loci of

ADRN genes, and those involved in G2/M phase of cell cycle (Fig. 2k, Supplementary Fig. 4c). These data support that the epigenetic and transcriptional landscapes in the naïve tumors have been reprogrammed when tumor cells developed therapy resistance.

We investigated if both Th-MYCN/ALK[F1178L] and SJNB14 models share common gene signatures, including the SCP gene signature, after tumors developed resistance to indisulam. However, the

**Fig. 2 | Indisulam resistance is associated with cell state switch from ADRN to MES and melanocytic state in human *MYCN*-amplified neuroblastomas.**
**a** Individual tumor volume for SJNB14 PDXs implanted in CB17/SCID mice undergoing repeated cycles (2 weeks as one cycle) of treatment with vehicle (*n* = 5) and 25 mg/kg indisulam (*n* = 5), 5 days on, two ways off. Note: relapsed tumors after the 4th cycle were re-implanted to new mice before the 5th cycle treatment due to the aging of the primary mice. GSEA shows the ADRN (**b**) and hypoxia (**c**) gene signatures are significantly downregulated in resistant tumors vs. naïve tumors, while the interferon (**d**) gene signature is significantly upregulated. Nominal *p*-values are calculated by one-sided Fisher's exact test. **e** Heatmap showing that the expression changes of melanocytic markers, Wnt ligands, MES, and ADRN transcriptional factors in naïve (biological replicates *n* = 2) vs. resistant tumors (biological replicates *n* = 2). The z-score-based heatmap after log2-transformed expression is generated by using the SRplot program. **f** The expression of melanocytic markers and Wnt ligands in a human neuroblastoma cohort (TARGET study)[21]. The expression scale bar indicates z-score based heatmap after log2-transformed expression. **g** Log2 transformed *RBM39* expression from normalized RNA-seq in naïve vs. resistant SJNB14 tumors (biological replicates *n* = 2). **h** Volcano plot shows the differential peaks of H3K27Ac in naïve vs. resistant tumors by CUT&Tag analysis. Y-axis is −log10 transformed p values, x-axis is log2 transformed fold changes. *P*-value is calculated by the Benjamini-Hochberg method. **i** IGV plots show the peak changes of at the loci of *TWIST1*, *PHOX2B*, and *S100B*. **j** GSEA shows the upregulation of H3K27Ac at the genomic loci of MES TFs. Nominal *p*-values are calculated by one-sided Fisher's exact test. **k** GSEA shows the downregulation of H3K27Ac at the genomic loci of ADRN signature genes. Nominal *p*-values is calculated by one-sided Fisher's exact test. Source data are provided as a Source Data file.

upregulated gene signatures in both models were barely overlapping. This could be due to multiple reasons. First, neuroblastoma is a very heterogeneous disease and could occur at different developmental stages of neural crest cell lineage. Neural crest cells are a transient stem cell population that develops into different cell lineages under different development cues. Second, the species specificity (mouse vs. human) could also make a difference. Third, the tumor drivers are different (MYCN/ALK^F1178L in murine GEMM vs. the PDX model with MYCN amplification derived from a relapsed patient that received intensive chemotherapy). Nevertheless, both models exhibited gene signatures that can be projected to some stages of neural crest lineage during development, as characterized by Schwann cell precursor and melanocytic markers. These data support the lineage plasticity of neuroblastoma cells in therapy resistance, and this may be associated with differentiation and de-differentiation of neural crest lineages. One recent study characterized chemotherapy-resistant high-risk neuroblastoma persister cells and found that these persisters were not a uniform population[66]. Rather, these cells were composed of 4 different groups and exhibited distinct gene signatures. Interestingly, this study identified some persister tumors that had elevated Schwann cell signatures[66]. Taken together, these data provided additional evidence that tumor cells not only switched their state from ADRN to MES, but also acquired additional molecular features such as those expressed in melanomas once they developed resistance, further emphasizing the cellular plasticity of neuroblastoma cells, which is reminiscent of cellular pliancy of neural crest or SCP cells.

### Indisulam resistance is associated with RBM39 upregulation and cell state switch from MES to ADRN in a C-MYC−overexpressing xenograft model

Both *Th-MYCN/ALK^F1178L* and the *MYCN*-amplified SJNB14 PDX models demonstrated the cell state switch from ADRN to MES when they developed therapy resistance, although a more complex and multidirectional transdifferentiation underlies the de facto mechanism (Figs. 1, 2). To some degree, our data are in line with the hypothesis that the conversion of ADRN to MES may account for the chemoresistance of neuroblastoma cells. However, the conversion of MES to ADRN of neuroblastoma cells in therapy resistance is less clear. To explore whether the neuroblastoma cells in an MES state could convert to an ADRN state under therapy selection, we treated SK-N-AS xenografts (C-MYC overexpression by a translocated super-enhancer[13], known as MES dominant[45,46]) with indisulam. Again, SK-N-AS xenografts also showed 100% tumor regression after a 10-day dosing with 25 mg/kg of indisulam (Supplementary Fig. 5a) and remained sensitive to indisulam until the 4th repeated treatment cycle. RNA-seq and GSEA analysis revealed that indeed the resistant tumor cells acquired ADRN and neuronal gene signatures with a significant reduction of MES and interferon signatures (Fig. 3a−e), supporting the hypothesis of interconversion of ADRN and MES states of neuroblastoma cells. Like the *MYCN/ALK^F1178L* resistant tumors, *RBM39* mRNA was increased by about

4-fold in the resistant SK-N-AS xenografts (Fig. 3f), but not its paralog, *RBM23* (Fig. 3f), which has previously been shown to have no effect on RNA splicing[31]. These data further verified that RBM39 was the specific target of indisulam that was responsible for the therapeutic efficacy, leading to a selective pressure on RBM39 in resistant tumors. At the same time, cells underwent cell state alterations from MES to ADRN.

### Epigenetic reprogramming of indisulam−resistant c-MYC−overexpressing neuroblastoma cells

We hypothesized that the resistant SK-N-AS neuroblastoma cells had undergone an epigenetic reprogramming to alter their MES cell state and thus enhancing the transcription of *RBM39*. To test this hypothesis, we derived cell lines from three independent SK-N-AS indisulam-resistant tumors. These cell lines indeed expressed high levels of RBM39 and were resistant to indisulam treatment in vitro (Supplementary Fig. 5b, c). To test if the resistant cells were still dependent on RBM39, we introduced exogenous DCAF15 into the indisulam-resistant SK-N-AS cells, which led to a remarkable degradation of RBM39 and cell death in comparison with the parental cells (Supplementary Fig. 5d−g). These data indicate that the indisulam-resistant cells are still dependent on RBM39. To understand the mechanism of cell state switching, we performed Assay for Transposase-Accessible Chromatin with high-throughput sequencing (ATAC-seq) for mapping genome-wide chromatin accessibility of resistant versus naïve cells. Many ADRN genes such as *PHOX2A* and *MYCN* as well as the *RBM39* locus showed increased chromatin accessibility in the resistant cells (Fig. 3g), in line with the GSEA results from RNA-seq analysis. Motif analysis for the predicted transcriptional factor binding at the genes with altered chromatin accessibility demonstrated that the MES transcriptional factors, such as FOSL1, FOSL2, JUNB, JUND, were enriched in naïve SK-N-AS cells, but not in the resistant cells (Fig. 3h, i). However, the ADRN transcriptional factors such as ASCL1 and NF1A were enriched in the resistant cells. Interestingly, we found that the activity of SP transcriptional factor family members SP2 and SP3 was also enriched in the resistant SK-N-AS cells, together with MYCN (Fig. 3h, i). While it is unclear what the role of SP transcription factors is in neuroblastoma identity, one early study indicates that they are involved in driving expression of *MYCN* in neuroblastoma cells[67]. To further corroborate our ATAC-seq study, we carried out CUT&Tag to assess the global alterations of H3K27Ac in naïve and resistant cells (Fig. 3j), an epigenetic mark indicating active promoters and enhancers in gene transcription. GSEA of the differential peaks near the annotated genes revealed that the 20q locus (where RBM39 resides) ranked at the top in the resistant cells, while the c-MYC targets ranked at the top in the naive SK-N-AS cells (Fig. 3k). Again, motif analysis of the altered H3K27Ac peaks showed that the binding of MES transcriptional factors, such as in the FOS and JUN families, was specifically reduced in the resistant

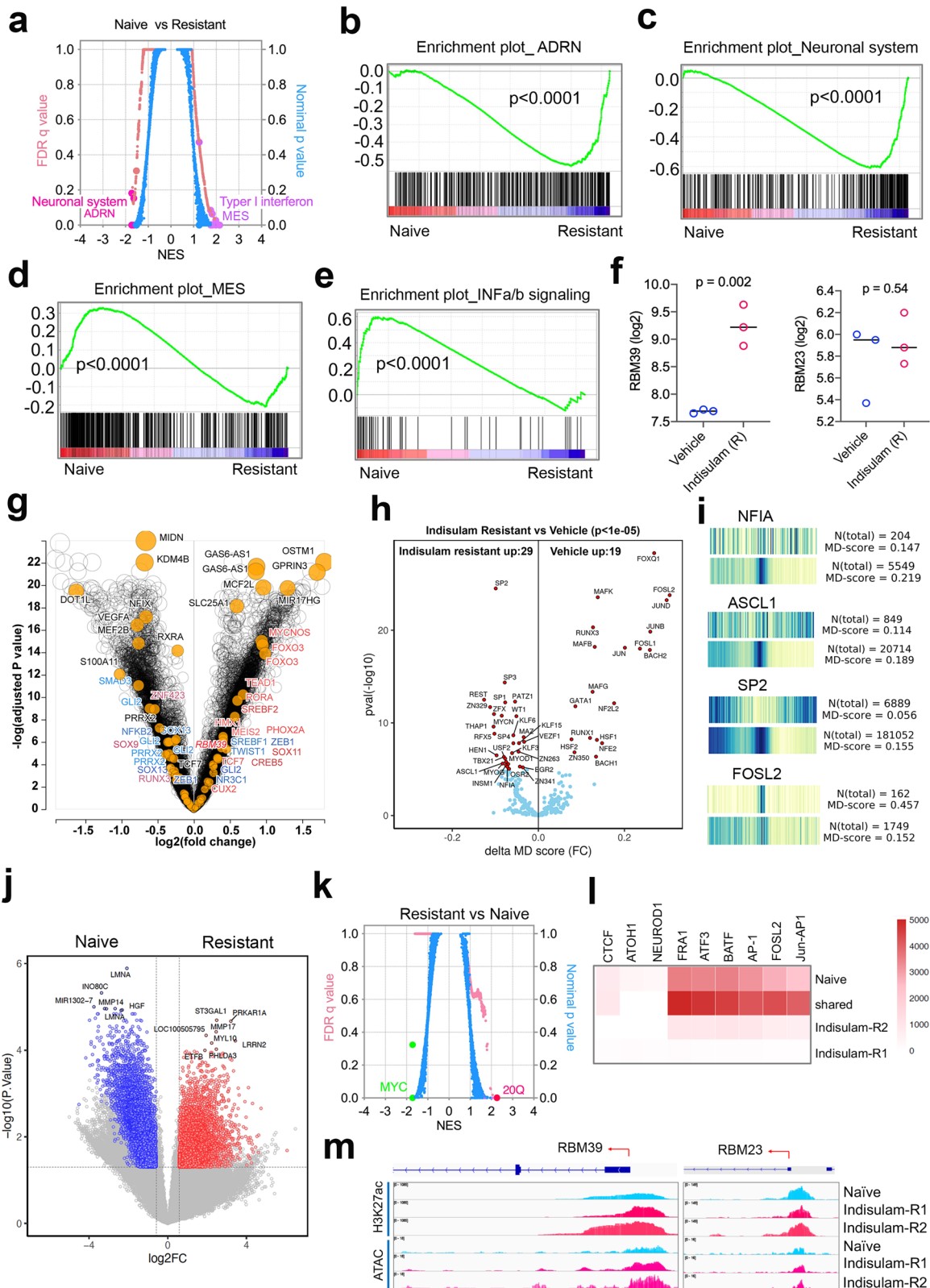

cells (Fig. 3l), similar to the results from ATAC-seq. The *RBM39* locus showed a greater increase in H3K27Ac than the *RBM23* in the resistant cells, in line with the elevated ATAC-seq peak (Fig. 3m), which explains why the resistant cells expressed higher levels of RBM39. These data indicate that the resistant SK-N-AS cells acquired a new cellular state under the selection of repeated indisulam treatment through epigenetic reprogramming.

**Cell state alterations are coupled with a switch of dependency on epigenetics and cell state–specific transcription factors**

We surmised that the distinct epigenetic landscapes of naïve and resistant tumor cells may result in dependency on specific epigenetic modifiers and cell state–specific transcription factors in these cells, as cell differentiation and de-differentiation is regulated by a cascade of cell state–specific transcriptional factors (TFs) in cooperation with

**Fig. 3 | Indisulam resistance is associated with RBM39 upregulation and cell state switch from MES to ADRN in a *C-MYC*-overactive xenograft model. a** GSEA analysis of naïve (biological replicates *n* = 3) vs. relapsed tumors (biological replicates *n* = 3). Right y-axis indicates nominal p values (one-sided) and left y-axis indicates false discovery rate (FDR) *q*-values (one-sided with Benjamini-Hochberg method), and normalized enrichment score (NES) (x-axis). **b-e** GSEA shows "ADRN" and "neuronal system" gene sets. Nominal *p* values are calculated by one-sided Fisher's exact test. **f** *RBM39* and *RBM23* expression in naïve (biological replicates *n* = 3) vs. resistant (biological replicates *n* = 3) SK-N-AS tumors. *p* values are calculated by two-sided Student *t*-test. **g** Volcano plot for differential peaks of ATAC-seq on or at nearby genes in resistant (biological replicates *n* = 4) vs naïve tumors (biological replicates *n* = 2). Y-axis = p values (two-sided, corrected by Benjamini-Hochberg method) transformed by -log10. X-axis = fold changes transformed by log2. Red =ADRN genes, Blue = MES genes. **h** Volcano plot for differential binding motifs of transcriptional factors. Y-axis indicates *p* values (two-proportion z-test)

transformed by -log10 while x-axis indicates the fold changes of delta MD score. **i** Heatmap showing the motif displacement (MD) distribution of NFIA, ASCL1, SP2 and FOSL2, MD-score, and the number of this motif within 1.5 kb of an ATAC-seq peak naïve vs resistant tumors. **j** Volcano plot for differential peaks of H3K27Ac in naïve vs resistant tumors by CUT&Tag analysis. *P* value is two-sided, corrected by the Benjamini-Hochberg method. **k** GSEA analysis for genes with differential peaks of H3K27Ac in resistant (biological replicates *n* = 4) vs. naïve (biological replicates *n* = 2) tumors by CUT&Tag analysis. Blue indicates nominal *P* values (one-sided) and pink indicates false discovery rate (FDR) *q* values (two-sided, corrected by Benjamini-Hochberg method). **l** Heatmap indicates the binding motifs for transcriptional factors enriched in naïve, resistant and both based on Homer Motif analysis of H3K27Ac peaks. **m** IGV snapshots displaying the peaks in naïve vs resistant SK-N-AS cells by H3K27Ac CUT&Tag and ATAC-seq at *RBM39* and *RBM23* genomic loci. Source data are provided as a Source Data file.

epigenetic modifiers. To test this hypothesis, we performed CRISPR-Cas9 screening (human epigenetic library with 8 gRNAs/gene and transcription factor library with 4 gRNAs/gene) to knock out these genes for a dropout screen with MAGeCK analysis in naïve versus resistant cells (Fig. 4a, Supplementary data 1, 2, 4, 5), in which reduction of gRNA reads (dropout) indicates that cells are dependent on it for survival. In the transcription factor library screening, we identified more pre-mRNA splicing factors in the resistant cells, particularly when they were cultured under indisulam selection (Fig. 4b), which is consistent with the function of RBM39 being critical to splicing in neuroblastoma[30], which also verified the screening robustness. Then we specifically examined the ADRN and MES transcriptional factors (Fig. 4c). While a general transcriptional regulator CDK9 was essential to both naïve and resistant cells (Fig. 4d), c-MYC and FOSL2 of MES TFs in the naïve cells were ranked above the ADRN TF HAND2, TBX2 and PHOX2B (Fig. 4c, left). However, in the resistant cells with or without indisulam selection, c-MYC and FOSL2 dropped down while HAND2, PHOX2B, and TBX2 became more essential to the survival of the resistant cells (Fig. 4c, middle and right, 4e, 4f). These data demonstrate that naïve cells and indisulam-resistant cells have specific dependencies on transcriptional factors that determine the cell state of neuroblastoma.

Then, we compared the epigenetic dependency of naïve and resistant cells, in which the epigenetic modifiers formed unique protein-protein interaction network modules (Fig. 4g), consistent with the fact that epigenetic modifiers usually form protein complexes to function. We identified chromatin remodeling INO80 complex was more critical to the survival of naïve SK-N-AS cells, while the SWI/SNF complex components (PBRM1, SMARCE1) and SIN3A complex components (PHF12, ARID4B) were more important in the resistant cells (Fig. 4g-i). Notably, we observed that INO80C was one of the top genes with active H3K27Ac in naïve SK-N-AS cells (Fig. 3j), providing additional evidence that the INO80 complex might be specifically important to the naïve SK-N-AS cells dominated by an MES state, while the SWI/SNF complex could be more essential to resistant SK-N-AS cells that acquired an ADRN state.

### Selective and shared dependency of naïve and resistant neuroblastoma cells on targetable kinases

The dependency switch of neuroblastoma cells on epigenetic modifiers and cell state–specific TFs suggests that additional dependencies may correspondingly change once neuroblastoma cells convert to another state. As most epigenetic regulators, such as ARID4B and transcription factors, are difficult to target, we tested if we could identify targetable vulnerabilities of indisulam-resistant neuroblastoma tumor cells by focusing on kinases. Again, with a similar approach, we screened naïve SK-N-AS and indisulam–resistant SK-N-AS cells cultured with or without 250 nM indisulam, using the human kinase CRISPR library (Fig. 5a; Supplementary data 3, 6 gRNAs/gene).

We identified 43, 31, and 25 kinases that were essential for either naïve, or indisulam-resistant cells under selective pressure or no selective pressure, respectively; and 13 of them were commonly shared (Fig. 5b; Supplementary data 6). We found that *CLK3*, *DYRK1A* and *DYRK1B*, members of the dual-specificity serine/threonine and tyrosine kinase that play an important role in pre-mRNA splicing by phosphorylating serine- and arginine-rich splicing factors in the spliceosomal complex, are essential to naïve SK-N-AS cells but not for the indisulam-resistant cells (Supplementary data 6). This is in line with the mechanism of indisulam that targets splicing. However, the changes in expression levels of these kinases in resistant vs naïve cells were not correlated with the dependency switch (Supplementary data 7).We then specifically examined the targetable kinases with inhibitors available in clinical trials and found 9 kinases are essential to SK-N-AS cells regardless of indisulam resistance, and 2 kinases (*FGFR4* and *CDK2*) are specifically essential to indisulam-resistant SK-N-AS cells (Table 1). These kinases play critical roles in the G2/M phase cell cycle (i.e., *AURKA*, *AURKB*, *PLK1*), DNA repair (*ATR*, *CHEK1*) and gene transcription (*CDK7*, *CDK9*). We hypothesize that the combination of indisulam with any of these kinase inhibitors may enhance the efficacy and blunt disease relapse. To test this, we chose gartisertib, a selective ATR inhibitor currently in clinical trials[68], in combination with indisulam at 10 mg/kg, for a two-week treatment. Indeed, gartisertib in combination with indisulam significantly extended mouse survival (Fig. 5c) and showed very limited toxicity based on the results of body weight gain over time (Fig. 5d), suggesting that this combination is safe and effective. Recent studies have shown that ATR inhibition is able to reverse the chemoresistance of ALT (alternative lengthening of telomere) neuroblastoma due to telomere dysfunction–induced ATM activation[69], and significantly enhances the efficacy of ALK inhibition in transgenic neuroblastoma models driven by *MYCN* and *ALK*[70]. These promising preclinical data, together with our preliminary results, provide a rationale to continue testing the combination of indisulam with ATR inhibitors in more mouse models in the future.

### Cell state switch induced by indisulam treatment can lead to upregulation of GD2 expression through epigenetic reprogramming

GD2 is a disialoganglioside that is biosynthesized from the precursor gangliosides GD3/GM3 by β−1,4-N-acetylgalactosaminyltransferase (B4GALNT1, GD2 synthase) (Fig. 6a). The expression of GD2 in normal tissues is restricted to the brain, peripheral pain fibers, and skin melanocytes, but it is abundantly expressed in neuroectodermal tumors, including neuroblastoma[71]. The application of anti-GD2 immunotherapy has greatly improved the survival of high-risk neuroblastoma patients when it is combined with a differentiating agent and chemotherapy[72–74]. Recent advances in the development of GD2–based CAR T and CAR NKT therapies have provided additional evidence

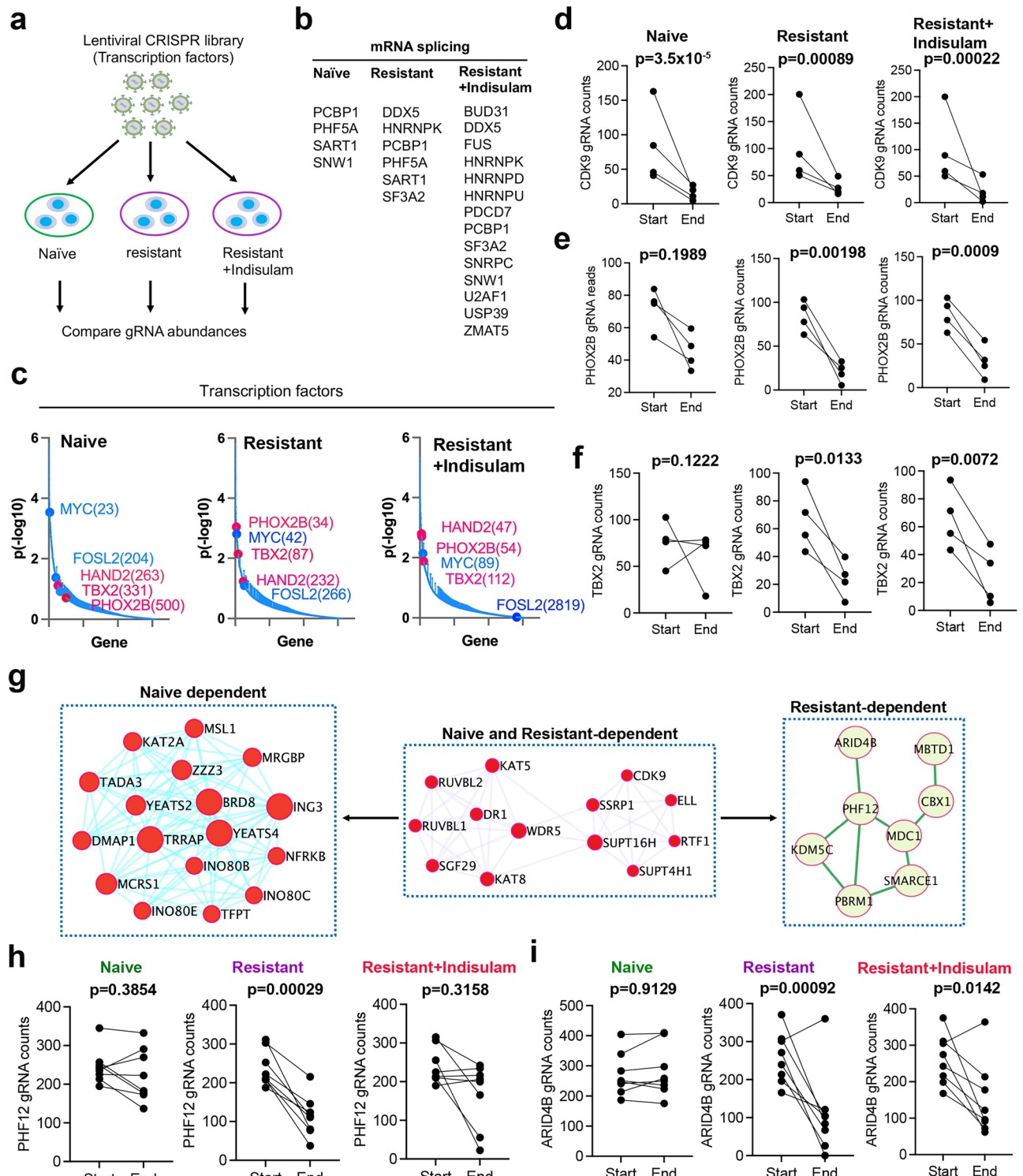

**Fig. 4 | Cell state alterations are coupled with a switch of dependency on epigenetics and lineage-specific transcription factors. a** Focused CRISPR library screening procedure using SK-N-AS cell lines derived from naïve and indisulam-resistant SK-N-AS tumor cells cultured for 3 weeks in the presence of absence of 250 nM of indisulam. **b** Dependency genes related to pre-mRNA splicing in naïve and resistant SK-N-AS cells with or without 250 nM of indisulam culture. **c** Ranked dependency genes encode transcriptional factors in naïve and resistant SK-N-AS cells with or without 250 nM of indisulam culture. Blue indicates MES TFs, and pink indicates ADRN TFs. *P* values (two-sided) generated by the MAGeCK-VISPR method.

**d–f** Normalized counts of gRNAs for CDK9, PHOX2B and TBX2 in naïve and resistant SK-N-AS cells with or without 250 nM of indisulam culture (biological replicates, *n* = 4). *p* value calculated by the two-sided MAGeCK method. **g** Protein interaction network analysis (STRING: https://string-db.org) of epigenetic regulators that are selectively essential to naïve or indisulam-resistant tumors. **h, i** Normalized counts of gRNAs for PHF12 and ARID4B in naïve and resistant SK-N-AS cells with or without 250 nM of indisulam culture (biological replicates, *n* = 6). *p*-value calculated by the two-sided MAGeCK method. Source data are provided as a Source Data file.

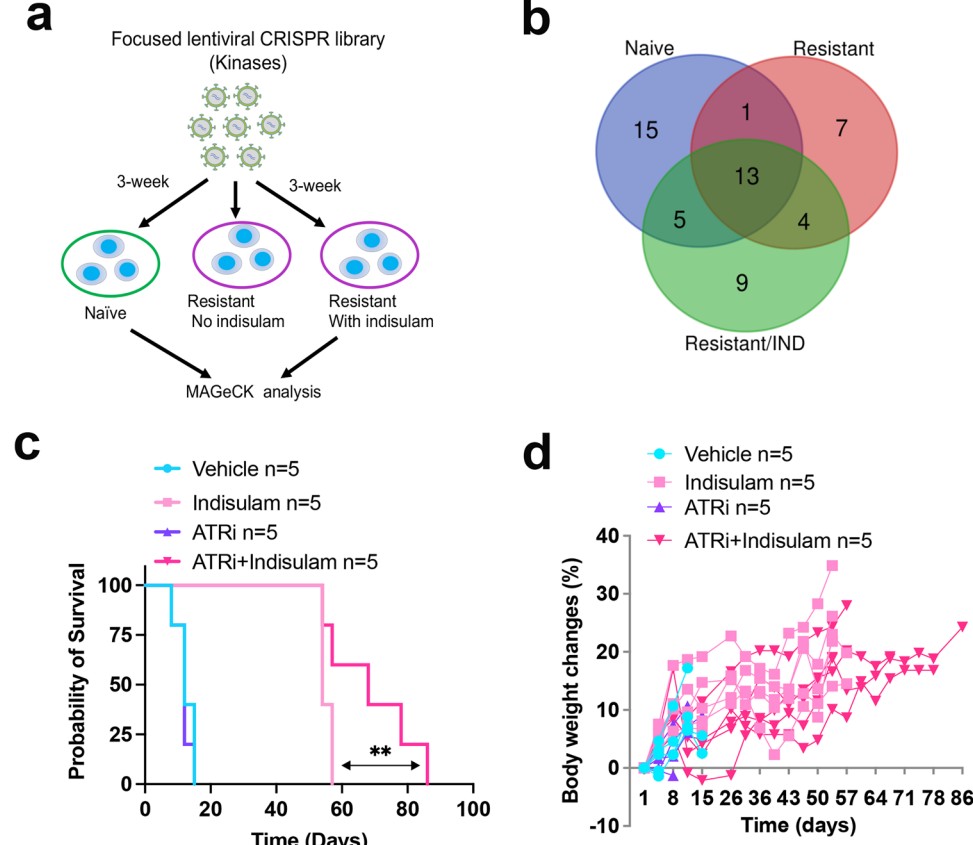

**Fig. 5 | CRISPR identification of targetable essential kinases leads to a more effective combination therapy consisting of indisulam and an ATR inhibitor. a** Human Kinase CRISPR library screening using SK-N-AS cell lines derived from naïve tumors and indisulam-resistant tumors in the presence of absence of 250 nM of indisulam. **b** Venn diagram showing the shared and cell-type-specific essential kinases. **c** Kaplan-Meier survival for SK-N-AS tumors treated with vehicle ($n = 5$), indisulam ($n = 5$), gartisertib ($n = 5$), or combination ($n = 5$). Indisulam is administered with 10 mg/kg, 5 days on, 2 days off, for 3 weeks, with an additional two weeks when tumors relapse with indisulam alone treatment. Gartisertib is administered with 10 mg/kg, once weekly. \*\*$p = 0.02$, which is calculated by the Log-rank (Mantel-Cox) test. **d** Body weight monitoring over time for each individual mouse. Source data are provided as a Source Data file.

**Table 1 | CRISPR screen identifies kinases with inhibitors available**

| Groups | Gene | Drug |
| --- | --- | --- |
| Shared by naïve, resistant, resistant cultured with indisulam | ATR | Gartisertib |
| | AURKA | MLN8237 |
| | AURKB | AZD2811 |
| | CDK7 | Samuraciclib |
| | CDK9 | BAY1251152 |
| | CHEK1 | LY2880070 |
| | PKMYT1 | RP-6306 |
| | PLK1 | Onvansertib |
| | TTK | CFI-402257 |
| Shared by resistant, and resistant cultured with indisulam | FGFR4 | FGF401 |
| | CDK2 | PF-07104091 |

showing the potential of anti-GD2 therapy for patients with relapsed disease[75,76]. While many reasons could account for the response failure of anti-GD2 immunotherapy for a fraction of patients, one of the hypotheses is that cell state switching may lead to low abundance of GD2 expression due to downregulation of GD2 synthase genes. A recent study has shown that cell lines derived from *TH-MYCN* transgenic tumors lost GD2 expression, accompanied by a switch of cell state from ADRN to MES[77]. Another study further supported that neuroblastoma state transition to MES confers resistance to anti-GD2 antibody via reduced expression of ST8SIA1[78]. To understand the impact of cell state alterations induced by repeated indisulam treatments on GD2 expression, we examined the key enzymes of GD2 synthesis (*B4GALNT1* and *ST8SIA1*) in our indisulam–resistant models. Our RNA-seq results showed that, in the transgenic *TH-MYCN/ALK^F1178L* model, one out of two resistant tumors showed a higher expression of *B4GALNT1* and *ST8SIA1* (Fig. 6b), while the SJNB14 PDX model showed no remarkable changes (Fig. 6c). In the SIMA model, *ST8SIA1* was significantly upregulated in the tumors with three rounds of indisulam treatment while the expression of *B4GALNT1* showed no changes (Fig. 6d). In the SK-N-AS model, the expression of both *B4GALNT1* and *ST8SIA1* was significantly upregulated in the resistant tumors (Fig. 6e), which is in line with the elevated enrichment of H3K27Ac and enhanced chromatin accessibility at the promoter regions of *B4GALNT1* and *ST8SIA1* (Fig. 6f, g), indicating that epigenetic reprogramming of SK-N-AS cells leads to upregulation of GD2 synthesis genes. Then, we applied flow cytometry analysis to profile the GD2 expression of the three resistant cell lines derived from SK-N-AS tumors. The result showed that, in comparison with the naïve control, all three resistant clones expressed higher levels of GD2 (Fig. 6h). These data indicate that cell state alterations induced by indisulam do not lead to a reduction of GD2 expression; rather, it result in enhanced expression, at least in some neuroblastomas. These data provide a rationale for a combination therapy of indisulam in combination with anti-GD2 immunotherapy.

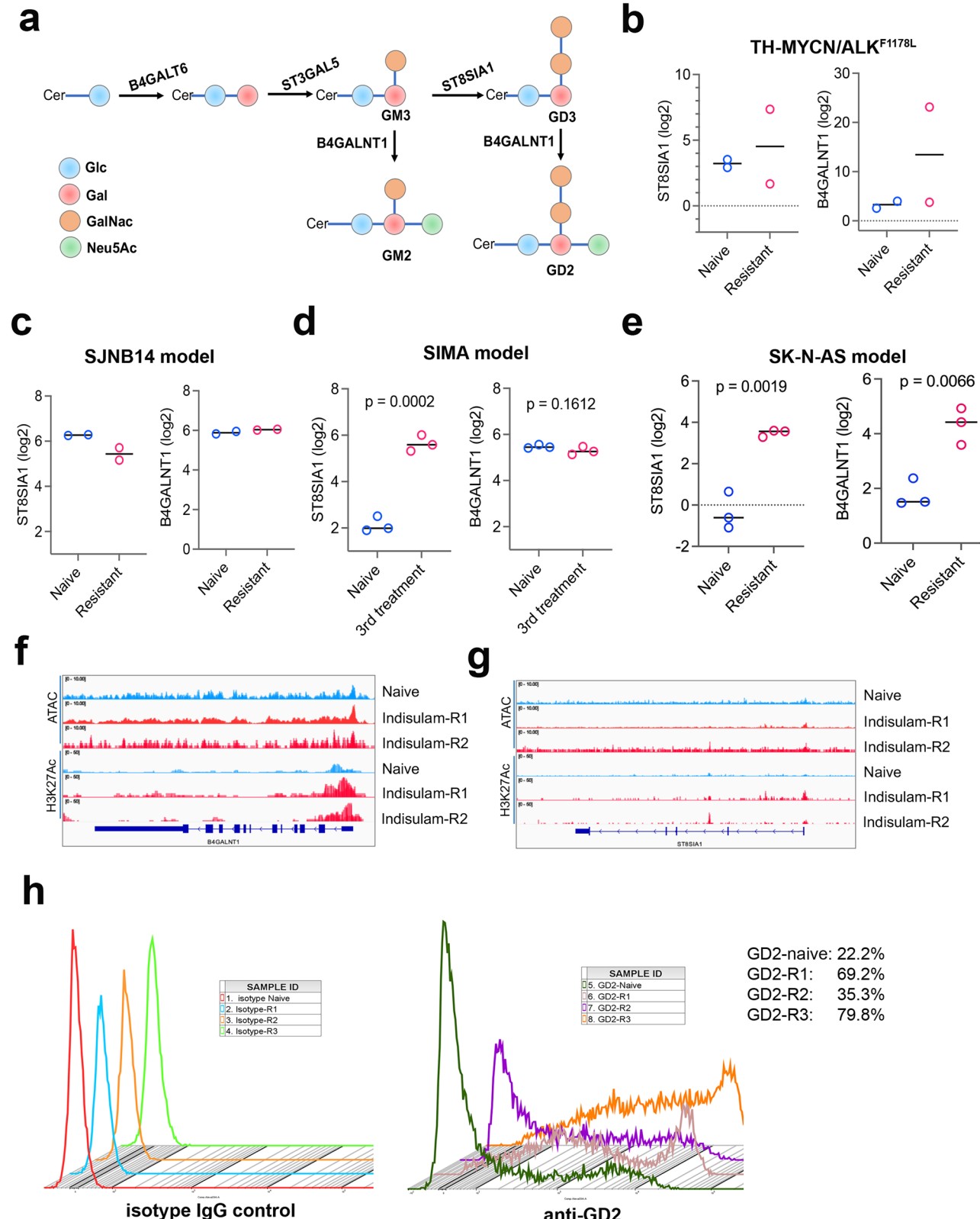

**Fig. 6 | Cell state switch induced by indisulam treatment can lead to upregulation of GD2 expression through epigenetic reprogramming. a** Cartoon depicts the GD2 synthesis and related key enzymes. Cer, ceramide, N-acylsphingosine; Glc, glucose; Gal, galactose; GalNac, N-acetylgalactosamine; Neu5Ac, N-acetyl neuraminic acid. **b–e** Expression of the two key enzymes in GD2 synthesis in 4 different neuroblastoma models. P values are calculated by two-sided, Student t-test when biological replicates *n* = 3. **f**, **g** Snapshots of H3K27Ac CUT&Tag and ATAC-seq at *B4GALNT1* and *ST8SIA1* genomic loci using the IGV program, displaying the peaks in naïve vs resistant SK-N-AS cells. **h** Flow cytometry analysis of GD2 expression in naïve and indisulam-resistant SK-N-AS cells. Left, isotype IgG as control; right, anti-GD2 intensities. The experiment was done once. Source data are provided as a Source Data file.

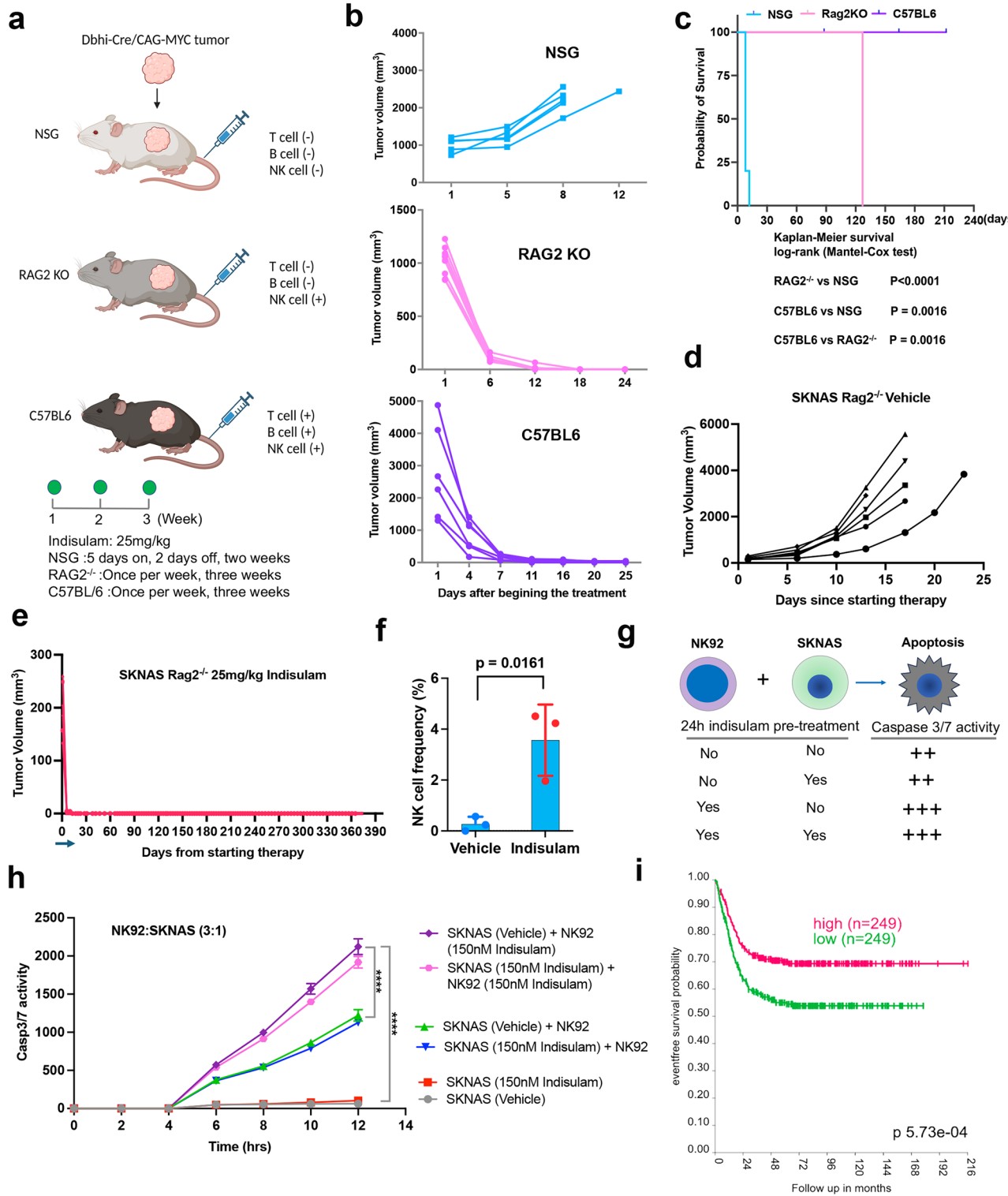

## Complete eradication of C-MYC-driven neuroblastoma in NK cell competent mice

We previously showed that indisulam induced durable complete responses in *C-MYC*− and *MYCN/ALK^F1178L*−driven neuroblastoma models under immune competent settings[30]. One recent study indicates that indisulam induces neoantigens and elicits anti-tumor immunity, augmenting checkpoint immunotherapy in a manner dependent on host T cells and peptides presented on tumor MHC class I[79]. The excellent efficacy of indisulam in immune competent neuroblastoma models leads to one hypothesis that indisulam modulates T cell

immune responses that eliminate cancer cells by making neoantigens through altered pre-mRNA splicing. Thus, despite the cell plasticity of neuroblastoma that may lead to therapy resistance, harnessing the immune system activated by indisulam may lead to a disease cure. To test this hypothesis, we implanted syngeneic *C-MYC*−driven neuroblastoma (derived from *Dbh-iCre/CAG-C-MYC* mouse[30]) into immune competent C57BL/6 mice, Rag2^−/− mice (no T and B cells, intact NK cells), and immune deficient NSG mice without T, B, and NK cells (Fig. 7a), which were then treated with indisulam at the indicated schedules. Given the high potency of indisulam against

**Fig. 7 | Potent efficacy of indisulam in NK cell competent mice. a** Tumors derived from C-MYC transgenic neuroblastoma model implanted into immunodeficient NSG mice, T and B cell-deficient Rag2$^{-/-}$ mice and immune competent C57BL6 mice, with indicated therapeutic schedules. Graph generated by Biorender (www. biorender.com). **b** Individual tumor volume for each group treated with indisulam. NSG ($n = 8$), Rag2$^{-/-}$ ($n = 7$), C57BL/6 ($n = 6$). **c** Kaplan-Meier survival curve for each group of mice treated with indisulam, with indicated $p$ values for comparisons of each group. P value is calculated by Log-rank (Mantel-Cox) test. **d** Individual tumor volume for SK-N-AS xenografts implanted into Rag2$^{-/-}$ mice treated with vehicle ($n = 6$). **e** Individual tumor volume for SK-N-AS xenografts implanted into Rag2$^{-/-}$ mice treated with 25 mg/kg of indisulam ($n = 6$), for 5 days on and two days off for two weeks, indicted by the blue arrow. **f** Percentage of IFNγ-producing NK cells in c-MYC tumors treated with vehicle ($n = 3$) and indisulam ($n = 3$) for 3 days, analyzed by flow cytometry. Data are presented as Mean +/- SEM. $P$ value is calculated by two-

sided, unpaired Student $t$-test. **g** Schematic diagram shows the co-culture of NK92 and SK-N-AS cells in a ratio of 3:1 with or without a 24-hour pretreatment with 150 nM of indisulam. The live SK-N-AS cells are pre-stained by Incucyte Caspase-3/7 Green before mixed with NK92 cells. The release of the DNA dye and fluorescent staining of the nuclear DNA serves as the indicator of apoptosis. **h** Real-time quantification of apoptosis of SK-N-AS cells measured by the Incucyte Caspase-3/7 Green. The y-axis indicates the caspase-3/7 activity (apoptosis) while the x-axis indicates the elapsed time of co-culture of NK92 and SK-N-AS cells ($n = 4$ per group). Data are presented as Mean +/- SEM. $P$ values calculated by two-sided, Student $t$-test for pairwise comparisons at the 12 h timepoint. ***$p < 0.0001$. **i** Kaplan-Meier survival for neuroblastoma patients with high and low expression of *NCR1* gene in SEQC cohort (GSE62564)[83]. $P$ value calculated by Log-rank (Mantel-Cox) test. Source data are provided as a Source Data file.

neuroblastomas[30], we began treatment after tumor sizes reached over 1000 mm³. Again, we observed exceptional efficacy of indisulam in C57BL/6 mice, which showed complete responses (Fig. 7b). Interestingly, all tumors in Rag2$^{-/-}$ mice also underwent complete responses (Fig. 7b, note that the dosing schedule for the two models is one dose per week). However, indisulam showed little efficacy to this *C-MYC*–driven mouse neuroblastoma model in immune deficient NSG mice, and progressive disease occurred even when given 5 days of treatment per week for two weeks. In line with the tumor responses, Kaplan-Meier analysis showed that indisulam treatment led to durable complete responses in both C57BL/6 and Rag2$^{-/-}$ mice (Fig. 7c). While Rag2$^{-/-}$ mice were sacrificed earlier than the C57BL/6 mice, it was not because of disease relapse rather it was due to mouse aging-related illness of this specific strain. Similarly, indisulam treatment led to complete and durable responses in human neuroblastoma xenografts (SK-N-AS) implanted into Rag2$^{-/-}$ mice (Fig. 7d, e). However, all SK-N-AS xenografts eventually relapsed in NSG mice after 2 months of therapy (Supplementary Fig. 5a), suggesting that residual disease is responsible for cancer relapse, which is eliminated in Rag2$^{-/-}$ mice. These data indicate that the innate immunity components, such as NK cell,s might play a critical role in indisulam-mediated anti-cancer activity. NK cells are known to mount rapid responses to damaged, infected or stressed cells, and play a major role in first-line innate defenses against viral infection and tumor growth[80]. To verify the anti-tumor role of NK cells in Rag2$^{-/-}$ mice, we performed FACS analysis to examine infiltration of the interferon gamma (IFNγ) producing NK cells in tumors treated with vehicle and indisulam for 3 days, respectively. Indeed, the IFNγ-producing NK cell frequency among all cells sorted from the tumors was significantly increased in the indisulam group in comparison with the control group (Fig. 7f), suggesting that indisulam treatment leads to an inflamed tumor microenvironment with a high percentage of activated NK cell infiltration. Considering the genetic background difference between NSG mice and Rag2-/- or C57BL/6 mice, we therefore directly assess whether NK cells can be activated by indisulam to enhance neuroblastoma cell killing. We utilized a co-culture system by mixing NK92 cells (a human lymphoma-derived cell line phenotypically similar to activated NK cells)[81] with SK-N-AS cells. Both cell types were pretreated with a suboptimal dose of indisulam (150 nM that does not induce cell death and only modestly degrades RBM39 protein, Supplementary Fig. 6) and co-cultured at a 3:1 effector-to-target (E:T) ratio (Fig. 7g). SK-N-AS cell viability was assessed by measuring apoptosis through caspase 3/7 activity using Incucyte Caspase-3/7 Green (Fig. 7h). The live SK-N-AS cells were pre-stained with Caspase-3/7 Green before being mixed with NK92 cells. The inert, non-fluorescent substrate crosses the cell membrane where it is cleaved by activated caspase-3/7, resulting in the release of the DNA dye and fluorescent staining of the nuclear DNA, serving as an indicator of apoptosis. Interestingly, pretreatment of SK-N-AS cells with indisulam did not enhance NK92-mediated cell killing. In contrast, pretreatment of NK92 cells with indisulam significantly increased their cytotoxic activity

against SK-N-AS cells (Fig. 7h). Our data were consistent with a recent report showing that indisulam enhanced NK cell–mediated killing in an in vitro assay[82]. To corroborate the importance of NK cells in anti-neuroblastoma activity, we examined the association of NKp46 that is the major NK cell-activating receptor involved in the elimination of target cells and mediates tumor cell lysis. Indeed, a higher expression level of the NK cell marker *NCR1* was correlated with a better event-free survival and overall survival of neuroblastoma patient cohorts[83] (Fig. 7i; Supplementary Fig. 7a), regardless of risk and *MYCN* status (Supplementary Fig. S7b–e).

## Durable and complete responses of anti-GD2 immunotherapy in combination with indisulam against three immune-competent neuroblastoma models

Pediatric solid tumors including neuroblastoma are generally immune "cold", making it challenging to develop effective immune checkpoint blockade (ICB) therapies, as evidenced by clinical trials[84]. However, the excellent efficacy of indisulam to *C-MYC* tumors in Rag2$^{-/-}$ mice suggests that the innate immunity, including NK cells can be leveraged to develop more effective therapies against neuroblastoma. To further understand the effect of indisulam on tumor microenvironment in the immune-competent setting, we harvested tumors from the Th-MYCN/ALK$^{F1178L}$ mice after a 3-day treatment with indisulam (25 mg/kg, $n = 1$) and vehicle ($n = 2$), and then performed single cell RNA-seq analysis. The Uniform Manifold Approximation and Projection (UMAP) clustered the tumor samples into two groups (vehicle and indisulam) (Fig. 8a). UMAP identified nine cell clusters (Fig. 8a), which were annotated into five major cell types by gene markers for each cell type established from neuroblastoma scRNA-seq atlas[85], including tumor cells, immune cells, endothelial cells, mesenchyme-like cells, and Schwann-like cells (Fig. 8b). Distinct changes in cellular composition were observed following indisulam treatment (Fig. 8a, b). The tumor cell fraction (Mycn$^+$) was decreased from 96.45% (control) to 51.01% (indisulam), while the mesenchyme-like cell fraction (Prrx1$^+$) was increased from 0.80% to 4.23%, and the Schwann-like cell fraction was increased from 0.29% to 4.26% (Fig. 8b; Supplementary Fig. 8, Supplementary data 8). These Schwann-like cells were positive with *Sox10* (Fig. 8b; Supplementary Fig. 8), consistent with the bulk RNA-seq analysis that showed upregulation of SCP signature (Fig. 1).

Additionally, a remarkable increase in the immune cell population (Ptprc$^+$ or Cd45$^+$) was observed following indisulam treatment, rising from 1.92% in the vehicle control group to 39.19% in the indisulam treatment group (Fig. 8b; Supplementary Fig. 8, Supplementary data 8). The immune cells consisted of lymphoid cells, including NK cells (Klrk1$^+$) and T cells (Cd4$^+$ and Cd8b1$^+$), and myeloid cells, including dendritic cells (Cd11c$^+$) and macrophages (Mrc1$^+$) (Supplementary Fig. 8f-l). Among the 2980 identified immune cells, the majority were myeloid cells, including macrophages and dendritic cells (vehicle 440 vs indisulam 2373), with a smaller fraction representing NK (vehicle 24 vs indisulam 39) and T cell (vehicle 14 vs indisulam 61) populations

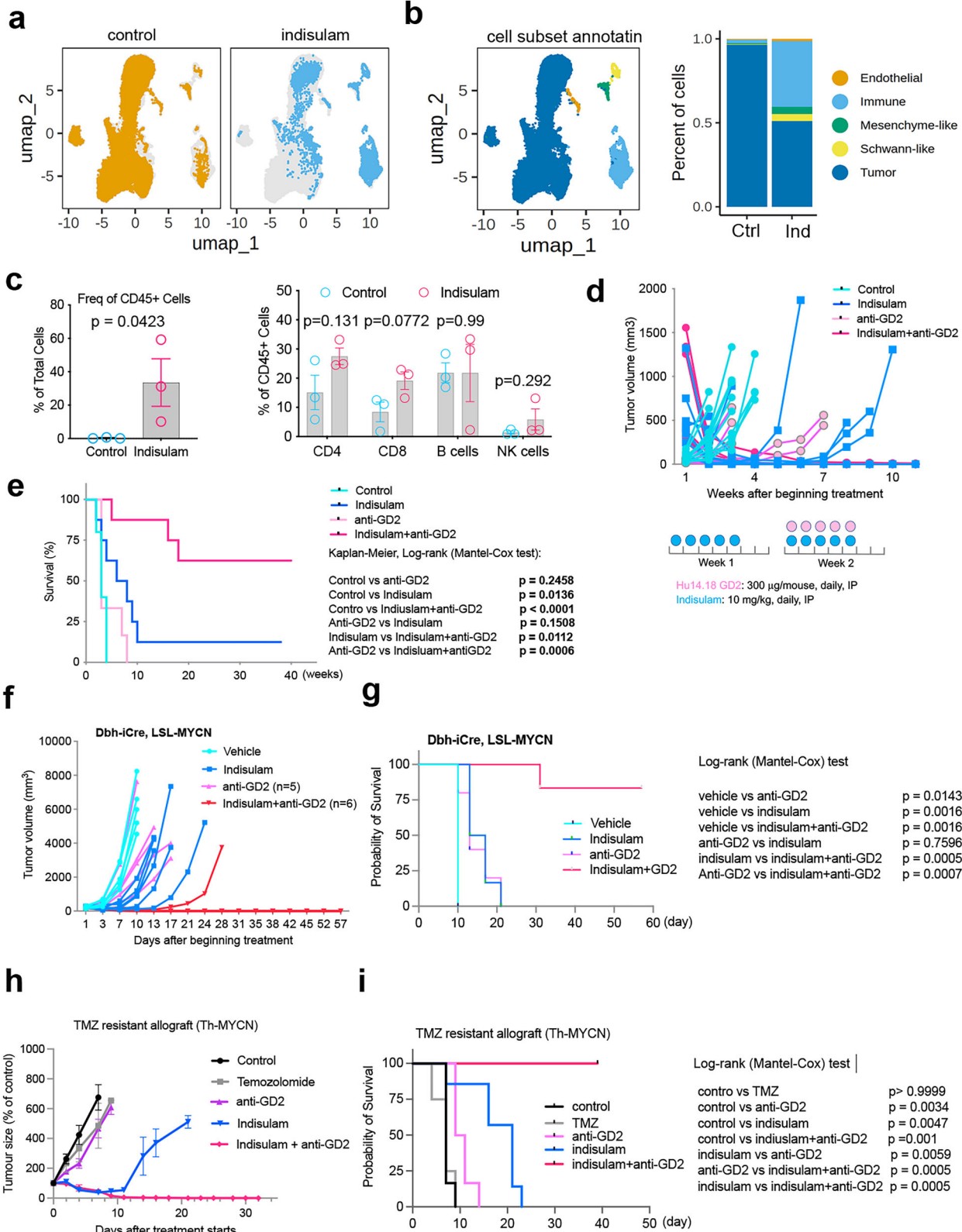

(Supplementary Fig. 9a, b). The macrophages were further divided into 4 clusters based on their differential gene expression (c5-c8, Supplementary Fig. 9a–c). Each cluster of macrophages expressed differential signaling pathways (Supplementary Fig. 9d). The functions of each cluster of macrophages may warrant further investigation. We further validated the immune cell infiltration by profiling immune cell infiltration in the tumors using FACS analysis with cell type-specific

antibodies. In comparison with the controls, tumors treated with indisulam exhibited a significant enrichment of CD45$^+$ cells (0.29 ± 0.35% vs 33.5 ± 24.6%) (Fig. 8c, right), in line with the enrichment of CD4 T cells (15.1 ± 10.2% vs 27.5 ± 4.9%), CD8 T cells (8.4 ± 5.8% vs 19.1 ± 5.2%), and NK cells (1.5 ± 0.88% vs 5.8 ± 6.3%) but not B cells (21.86 ± 5.9% vs 21.8 ± 17%) (Fig. 8c, left). The increase in immune cell infiltration induced by indisulam prompted us to test the hypothesis

**Fig. 8 | Potent efficacy of indisulam in combination with anti-GD2 in immune competent mice. a** Uniform Manifold Approximation and Projection (UMAP) of integrated scRNA-seq data from controls ($n = 2$) and indisulam-treated ($n = 1$) samples from the Th-MYCN/ALK[F1178L] models. **b** UMAP highlights the major cell types (left) and the bar plot shows the proportions of different cell types in control and indisulam samples (right). **c** Left shows the percentage of immune cells (CD45) in tumors treated with vehicle ($n = 3$) and indisulam ($n = 3$) for 5 days, analyzed by flow cytometry. Right shows the percentage of different types of immune cells in tumors treated with vehicle and indisulam for 5 days, analyzed by flow cytometry. Data are presented as Mean +/− SEM. P value calculated with two-sided, unpaired t test. **d** Individual tumor volume for each group treated with vehicle ($n = 11$), indisulam ($n = 8$), anti-GD2 ($n = 6$), combination ($n = 8$). Th-MYCN/ALK[F1178L] mice were treated with the indicated doses and schedule. **e** The Kaplan-Meier survival for transgenic MYCN/ALK[F1178L] mice treated with vehicle, anti-GD2, indisulam, combination as shown in (**d**), with indicated p values (log-rank test) for comparisons of

each group. **f** Individual Dbh-iCre;LSL-MYCN tumor volume treated with vehicle ($n = 5$), indisulam ($n = 5$), anti-GD2 ($n = 5$), combination ($n = 5$). The Dbh-iCre;LSL-MYCN syngeneic mice treated with 100 mg/kg of indisulam via tail vein injection, every 4 days with a total of 4 doses. The anti-GD2 mAb (300 μg/mouse) is given on the next day after indisulam treatment. **g** Kaplan-Meier survival for the Dbh-iCre;LSL-MYCN syngeneic mice as shown in (**f**), with indicated p values (log-rank test) for comparisons of each group. **h** Individual Th-MYCN tumor volume treated with vehicle ($n = 6$), temozolomide (TMZ, $n = 4$), indisulam ($n = 6$), anti-GD2 ($n = 6$), combination of indisulam and anti-GD2 ($n = 6$). The TMZ-resistant Th-MYCN syngeneic mice were treated with indisulam as **d** (10 mg/kg, 5 days on and 2 days off for 2 weeks), and anti-GD2 (15 μg/mouse, on day 1 and day 5), TMZ (6 mk/kg for 5 days). Data are presented as Mean +/− SEM. **i** Kaplan-Meier survival for the Th-MYCN syngeneic mice as shown in (**h**), with indicated p-values (log-rank test) for comparisons of each group. Source data are provided as a Source Data file.

that indisulam may enhance the efficacy of immunotherapy for this genetic model.

The application of anti-GD2 immunotherapy has greatly improved the survival of neuroblastoma patients when it is combined with differentiating agents and chemotherapy[72–74]. The anti-GD2 monoclonal antibody exerts an NK cell–dependent ADCC–mediated antitumor effect[57]. We therefore hypothesized that combining indisulam with anti-GD2 immunotherapy may achieve a long-term remission or cure of neuroblastoma. While our dosing schedule (25 mg/kg, 5 days on, two days off, two weeks) used in most cases had achieved remarkable responses in multiple high-risk neuroblastoma models[30], it was based on the maximally tolerated dose in mice (40 mg/kg) in a previous animal study that lacked pharmacokinetic (PK) rationale[86]. We identified and investigated a more clinically relevant indisulam dose and schedule. Briefly, we evaluated the plasma PK profile of indisulam in normal female CB17/SCID mice, approximately 12 weeks in age. Plasma indisulam was quantified with a qualified liquid chromatography–tandem mass spectrometry (LC-MS/MS) assay. A clinically relevant dose for mice was estimated from unbound plasma PK (Supplementary Fig. 10) and exposure – namely, the predicted indisulam average concentration over 5 days (Cavg,120 h). Assuming linear, dose-proportional PK, and similar plasma protein binding between species, a clinically relevant dose for mice would range from 6.25 to 12.5 mg/kg Dx5 Q3wks (once daily, 5 days per week for 3 weeks). This regimen would approximate the Cavg,120 hr achieved with the clinical 160 mg/m$^2$ Dx5 indisulam regimen[87,88]. We then tested the efficacy of indisulam (10 mg/kg) in combination with anti-GD2 mAb (Hu14.18K322A) produced by St Jude GMP, in immune competent MYCN/ALK[F1178L] transgenic mice. One of the advantages of the version of anti-GD2 antibody (Hu14.18K322A) is it has been modified to abrogate complement binding, thereby reducing pain caused by the antibody but retaining its clinical efficacy[57]. We dosed mice for week 1 with indisulam (10 mg/kg, Dx5, IP route due to the challenge of IV route for younger mice) surmising it may prime an immune response, followed by combination therapy in week 2 (10 mg/kg of indisulam and 300 μg/mouse of Hu14.18K322A given via IP for Dx5) (Fig. 8d). The tumor response was monitored by ultrasound imaging of MYCN/ALK[F1178L] mice that developed neuroblastoma in the abdomen. The results showed that while tumors responded to indisulam with 2-week treatment, eventually they relapsed (Fig. 8d). Anti-GD2 mAb only led to tumor growth delay in 2 out of 6 mice, suggesting this mouse model is recalcitrant to anti-GD2 mAb monotherapy. Strikingly, the combination of indisulam with anti-GD2 mAb for only two weeks of treatment led to a durable complete response in all tested mice, even with the original tumor size over 1.5 cm$^3$. In the combination group, one mouse died for unknown reasons around week 5, and the rest of the mice were culled due to other health conditions but none of them died of disease relapse (Fig. 8e). We further tested the efficacy of indisulam and anti-GD2 mAb in two additional neuroblastoma syngeneic models: Dbh-

iCre:LSL-MYCN model (C57BL6 background)[89] and a temozolomide-resistant Th-MYCN model (129 × 1/SvJ background). For the Dbh-iCre:LSL-MYCN model, indisulam was administered via tail vein injection with 100 mg/kg every 4 days for a total of 4 doses (based on the clinical dosing schedule recommended by Eisai pharmaceuticals). For the Th-MYCN model, indisulam was dosed as the Th-MYCN/ALK[F1178L] model. Regardless of the dosing schedule differences, both models exhibited complete and durable responses to the combination therapy while the monotherapy only achieved transient responses (Fig. 8f–i). These data indicate that the combination of indisulam and anti-GD2 mAb is highly effective against high-risk neuroblastoma models and may have translational feasibility to human patients. Taken together, it is likely that innate immunity mediated by NK cells in cooperation with indisulam plays the major role in eradicating the tumor cells. These data provide a rationale to translate the indisulam to the clinic in combination with anti-GD2 immunotherapy for high-risk neuroblastoma patients.

## Discussion

The cellular heterogeneity of neuroblastoma cells was observed decades ago[42–44]. Recent studies have defined two major populations (ADRN and MES) of neuroblastoma cells according to their transcriptomic and epigenetic features[45–47]. scRNA-seq analysis of neuroblastomas revealed that ADRN exists in nearly all tumors with variable degrees of an MES signature[51,52,54,90,91]. Using our transgenic MYCN/ALK[F1178L] model, we found that once tumor cells developed indisulam resistance, they predominantly acquired SCP features seen during normal adrenal medulla development[51–53]. SCP has high cellular pliancy with the potential to differentiate to different cell lineages depending upon the environmental cues. In an ADRN PDX model (MYCN amplified) that developed indisulam resistance, cancer cells lost their ADRN features and were enriched with expression features of melanoma markers such as MITF. It is well known that the melanocyte is a derivative of neural crest cell or SCP[63,92,93]. Conversely, the MES dominant tumors acquired ADRN features once they developed resistance to indisulam, which was not observed in previous studies. These findings illustrate that neuroblastoma cell states can not only interconvert but could also transdifferentiate to additional cell states from the ADRN and MES states. Thus, our data strongly suggest that the capacity of neuroblastoma cells transdifferentiating to different developmental stages of neural crest progeny is an important mechanism by which neuroblastoma acquires therapy resistance (Fig. 9a, b).

Cancers bear transcriptional features of their original tissue lineages under normal developmental programs, and such features are largely determined by a small number of master transcription factors (TFs)[94]. These master TFs typically regulate their own genes through an autoregulatory loop that forms the core transcriptional regulatory circuitry (CRC) of a cell, and their expression is dominated by superenhancers that drive high rates of transcription[94]. The transcriptional

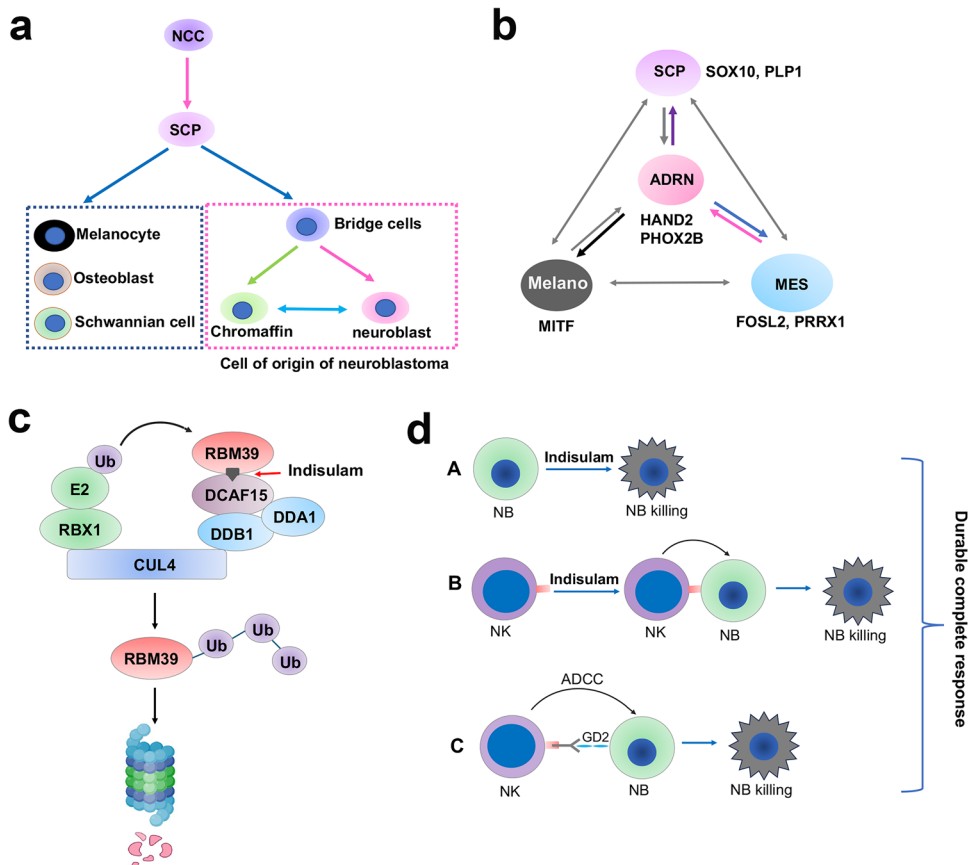

**Fig. 9 | Models for neuroblastoma cell plasticity and the working mechanism of indisulam. a** Schwann cell precursors from neural crest cells could differentiate to distinct lineages, including neuroblasts and chromaffin cells that are presumed to be cells of origin of neuroblastoma. NCC, neural crest cell. SCP, Schwann cell precursors. **b** Neuroblastoma cells have multiple cell states that are potentially able to interconvert. These cell states are determined by lineage–specific master TFs or CRC TFs. ADRN, adrenergic. MES, mesenchymal. Melano, melanocytic. **c** The mechanism of action of indisulam. **d** The anticancer mechanism for the combination of indisulam with anti-GD2 immunotherapy. Indisulam degrades RBM39 in neuroblastoma cells, leading to cell death due to splicing defects (top). Indisulam also directly activates NK cells, leading to NK cell-mediated killing (middle). At the same time, indisulam induced an inflamed state of neuroblastoma cells, causing immune cell infiltration. When GD2 immunotherapy is applied, NK cells elicit antibody-dependent cellular cytotoxicity (ADCC) to kill neuroblastoma cells (bottom). All three mechanisms together lead to a durable complete response.

programs of ADRN and MES neuroblastoma cells are dominated by their respective CRC TFs (Fig. 9b). Our study showed that neuroblastoma cells also switch CRC TF dependency once their cell states change. For example, we noticed that PHOX2B and TBX2, the two ADRN CRC TFs, were less important to the survival of naïve SK-N-AS cells (MES state) but more essential to the indisulam-resistant SK-N-AS cells (ADRN state). Notably, the expression of *HAND2*, *PHOX2B*, and *TBX2* did not show significant upregulation in the resistant SK-N-AS tumors. This raised one hypothesis that cells may lean on ADRN CRC TFs for survival when they switch their cell state from MES to ADRN, even though the expression levels of ADRN CRC TFs showed no significant upregulation. Additionally, the transcriptional activity of PHOX2B, HAND2 and TBX2 could be significantly enhanced during the cell state transition from MES to ADRN, leading to the cell dependency change. However, further work needs to be done to understand the underlying mechanism. The naïve and resistant cells also exhibited distinct dependency on epigenetic modifiers. We found that the naive SK-N-AS cells are more dependent on the INO80 chromatin remodeling complex, while the resistant cells were more dependent on the components of the SWI/SNF complex (PBAF complex), including PBRM1 and SMARCE1. The INO80 and SWI/SNF complexes are ATP-dependent chromatin remodelers. While there is no information about the functions of the INO80 complex in neuroblastoma, a previous study has shown that INO80 occupies >90% of super-enhancers in melanoma, and its occupancy is dependent on

transcription factors such as MITF[95]. However, a recent study has shown that the expression of SWI/SNF is regulated by SOX11 in neuroblastoma, and SOX11 is an ADRN CRC TF serving as a dependency transcription factor in adrenergic neuroblastoma. In the SJNB14 model, the expression of *SOX11* was reduced after the tumors switched their cell state from ADRN to other cell states, such as melanoma-like, which conversely expressed high levels of *MITF*. One study showed that SMARCE1 promoted neuroblastoma tumorigenesis through assisting MYCN-mediated transcriptional activation[96]. The components of SIN3A complex (ARID4B, PHF12) also appeared to be important to the survival of indisulam–resistant SK-N-AS cells. However, the information on ARID4B and PHF12 in cancer, particularly in neuroblastoma, was scarce.

One question is how RBM39 affects epigenetic reprogramming. It is a common survival mechanism for cancer cells to alter their cell state under detrimental stresses, such as chemotherapy and targeted therapy. For example, prostate and breast cancers often undergo epigenetic reprogramming in response to hormone and targeted therapies[97]. Similarly, neuroblastoma cells frequently change their cell morphology and transcriptomic state when exposed to chemotherapeutic and differentiating agents. These dynamic cellular states are closely linked to changes in the activity of cell state–specific CRC TFs (i.e., PHOX2B and PRRX1)[98] and epigenetic remodelers (i.e., SWI/SNF)[99]. As a key splicing regulator essential for neuroblastoma survival, RBM39 may be functionally connected to CRC TFs and epigenetic

modifiers, given the tight coupling between splicing and gene transcription. Loss of RBM39 imposes significant survival pressure on neuroblastoma cells, forcing them to adapt by epigenetically reprogramming their cell state—either to restore RBM39 levels or to upregulate survival pathways and/or downregulate cell death programs. Our CRISPR screening revealed distinct epigenetic dependencies between indisulam-naïve and resistant neuroblastoma cells, which correlate with CRC TF dependency switching. This suggests that RBM39, CRC TFs, and epigenetic regulators act in concert to govern neuroblastoma cell fate. Overall, these data support that cell state switching is accompanied by the corresponding alterations of epigenetic and CRC TF dependency. This was also reflected by the distinct dependency of kinases for survival of naïve and indisulam–resistant SK-N-AS models, suggesting that therapies could potentially be developed to block the cell lineage switching to overcome therapy resistance. In the future, how epigenetic modifiers and CRC TFs cooperate to determine the cell state under different cellular contexts needs to be elucidated.

RNA-seq analysis reveals 27 immune cell subtypes in neuroblastomas, including distinct subpopulations of myeloid, NK, B, and T cells[100]. However, the biological functions of these immune cells in neuroblastoma are not well understood. Recent studies have shown that tumor regression induced by C-MYC loss appears to be NK cell–dependent[101]. *MYCN* is also an immunosuppressive oncogene that dampens the expression of ligands for NK-cell–activating receptors in human high-risk neuroblastoma[102]. Therefore, NK cell–mediated immune surveillance may be critical to constrain neuroblastoma. The application of anti-GD2 antibody in neuroblastoma treatment significantly improves survival of patients with high-risk disease, and anti-GD2 mainly acts through NK cell–mediated antibody–dependent cellular cytotoxicity (ADCC)[72,103,104]. However, not all high-risk patients respond to anti-GD2 therapy. Thus, improving NK–NK-mediated cancer killing for MYC–driven neuroblastoma is attractive. Interestingly, we found that the expression of GD2 in MES type cells was greatly upregulated once they switched to an ADRN state, suggesting that indisulam may potentially enhance the efficacy of anti-GD2 immunotherapy. Innate immune cells, such as NK cells, are known to mount rapid responses to damaged, infected, or stressed cells. Therefore, RBM39 loss may attract innate immune cells to the tumor, which becomes highly inflammatory due to cell damage. Additionally, RBM39 loss could lead to neoantigen production through splicing anomalies, potentially triggering a specific T cell–mediated immune response. While indisulam may be important to stimulate the activity of CD8 T cells to boost the efficacy of ICBs by producing neoantigens through alternating pre-mRNA splicing[105], our study demonstrated that NK cells might be more critical for eliminating cancer cells in our syngeneic *C-MYC* and transgenic *TH-MYCN/ALK^F1178L* neuroblastoma models. We found that indisulam treatment promoted a remarkable infiltration of NK cells in the c-MYC tumors implanted in Rag2$^{-/-}$ mice, and various immune cells, including T and NK cells in the transgenic *TH-MYCN/ALK^F1178L* tumors, suggesting that indisulam induced an "inflamed" tumor microenvironment. Although it awaits future studies to dissect the functions of each type of immune cells, including the CD8 T cells in indisulam–mediated anticancer activity, the durable complete responses induced by the combination of anti-GD2 immunotherapy with indisulam suggest that NK cells might play an important role. This role was further supported by the C-MYC syngeneic model in that tumors were completely eliminated in NK cell-intact Rag2$^{-/-}$ mice that lack functional T and B cells, but little effect was observed in NK/T/B cell-deficient NSG mice. The in vitro cell co-culture of NK92 cells and SK-N-AS cells demonstrated that indisulam can directly activate NK cells, leading to enhanced killing, which further supports this hypothesis. One recent study showed that modulation of RNA splicing by indisulam enhances NK cell-mediated tumor surveillance functions[82]. RBM39 in NK cells could possibly affect the anti-tumor

functions of NK cells, which needs to be tested in future studies by using a conditional *RBM39* knockout mouse. Overall, our data support that indisulam exerts its anticancer mechanism through three modes (Fig. 9d). First, indisulam degrades RBM39 in neuroblastoma cells, leading to cell death due to splicing defects, which is a major mechanism of indisulam in an immunodeficient setting. At the same time, indisulam induces an inflamed state in neuroblastoma tumors, leading to infiltration of immune cells and NK cell–mediated killing in immune competent or innate immunity competent settings. And last, when GD2 immunotherapy is applied, NK cells elicit antibody-dependent cellular cytotoxicity (ADCC) to further enhance the killing of neuroblastoma cells. All three mechanisms together lead to a durable complete response in immune competent neuroblastoma models. Our study also indicates that enhancing NK cell–mediated antitumor effect by indisulam could be applied in patients with other cancer types who receive antibody immunotherapy that elicits ADCC. However, the models we used have different genetic backgrounds, which may influence the interpretation of NK cell function in indisulam-mediated cancer cell killing. In future studies, we will validate these findings using models with matched genetic backgrounds.

## Methods

### Cell lines and reagents

SIMA (DSMZ, ACC164), SK-N-AS (ATCC, CRL-2137) naive and indisulam-resistant cell lines (R1-R3) were plated in RPMI 1640 (Corning, cat#: 10-040-CM) complete media (10% FBS, Gibco, cat#A5256701, & 1% Pen Strep, cat#:15140-122). All cells were maintained at 37 °C in an atmosphere of 5% CO$_2$. All human-derived cell lines were validated by short tandem repeat (STR) profiling using PowerPlex® 16 HS System (Promega) once a month. Once a month, a polymerase chain reaction (PCR)-based method was used to screen for mycoplasma employing the LookOut® Mycoplasma PCR Detection Kit (MP0035, Sigma-Aldrich) and JumpStart™ Taq DNA Polymerase (D9307, Sigma-Aldrich) to ensure cells were free of mycoplasma contamination.

Indisulam, obtained from MedKoo Biosciences (catalog #201540), was prepared as a stock concentration of 300 mM in DMSO. The process involved rigorous vortexing and sonication until achieving thorough mixing, resulting in a dark red solution. Indisulam was then formulated in 3.5% DMSO, 6.5% Tween 80 (Sigma, P1892-10mL), and 90% saline (Sigma, S8776-100mL), with a sequential addition of chemicals beginning with the indisulam in DMSO, followed by Tween-80, and finally saline. Vehicle solution for a 20 g mouse is composed of 7 µl DMSO, 13 µl Tween-80 and 180 µl Saline. Indisulam solution for a 20 g mouse is composed of 4.3 µl Indisulam (from the 300 mM stock), 2.7 µl DMSO, 13 µl Tween-80 and 180 µl Saline, which leads to a final concentration of 0.5 mg/200 mL for 25 mg/kg dose. Additionally, the Indisulam solution was further diluted for treatment at 10 mg/kg. In that case, Tween-80 and Saline were employed.

Gartisertib was purchased from MedChemExpress (Cat#HY-136270). To make a 2 mg/ml stock solution of gartisertib, it was suspended in 15% Captisol (Cat#HY-17031)/0.1 M hydrochloride acid (LabChem, LC152204).

### Hu 14.18 K322A antibody production

The seed train for the 5 L bioreactor was initiated from a single vial of the Master Cell Bank (Hu14.18 K322A YB2/0 CHIL-02 Ab Clone 134 Master Cell Bank) that was thawed and the cells placed into a 125 mL Erlenmeyer flask with cHSFM (HSFM + 2 g/L Soytone + 2 g/L Phytone™ + 4 mM GlutaMax™+ 1000 ng/mL Methotrexate (MTX)) at a cell density of 0.4 × 106 cells/mL and placed into a shaking CO$_2$ incubator set at 37 °C, 70% humidity and 8% CO$_2$ and 120 rpm. Cells are split every 48–72 hr reaching six 1000 mL Erlenmeyer flasks (200 mL volume) (Passage 9), providing cells to seed 5 L of cHSFM medium at 0.1 × 106 cells/ml in a 50 L WAVE bioreactor. The WAVE bioreactor is cultured until Day 4 post inoculation, with a minimum cell density of 1.00 × 106 cells/mL was

achieved. The Hu14.18 K332A cells from the Wave bioreactor were used to seed a starting volume of 2.3 L at 0.25 × 106 cells/mL in the Sartorius Stedim Biotech BIOSTAT 5 L glass bioreactors.

Engineering runs were set up in 5 L Sartorius bioreactors with initial charge of 2.4 L of initial charge of cells and HSFM media with 6 mM glutamax and 2 g/L of Soytone and Phytone. The agitation rate was set at 350 RPM, the pH was set at 6.9, and the dissolved oxygen setpoint was 50%. The cell concentration at the time of inoculation was 0.4 × 106 cells /ml. The bioreactors were fed using HSFM containing 6 mM glutamax and 10 g/L Soytone-Phytone at a constant flow rate starting after 72 h post-inoculation and continued for 10 days. The cells were harvested when the viability dropped below 50%.

Hu 14.18 K322A culture from a 5 L bioreactor was clarified by centrifugation and filtered using a 0.2 μ nominal filter and a 0.22 μ sterile filter. The filtered supernatant was stored at 4 °C until purification.

The Hu14.18K322A Ab was purified using MabSelect Sure Protein A resin (Cytiva) equilibrate with PBS. The antibody was eluted from the column using 0.2 M Glycine, pH 3.5. The MabSelect purified Hu14.18 K322A Ab was subjected to a low pH hold step at pH 3.7 for 30 min. The antibody was neutralized to pH 6.0 using 50 mM Malonate buffer and 1 M Tris-HCl, pH 8.0. The Hu14.18 K322A Ab was then purified using Capto SP ImpRes resin (Cytiva) equilibrated with 50 mM Malonate buffer, pH 6.0. The Hu14.18 K322A Ab was eluted from the column using 50 mM Malonate buffer, pH 6.0 plus 200 mM NaCL. The Hu14.18 K322A Ab was passed through a Sartobind Q membrane (Sartorius) to remove potential Host DNA and endotoxins. Hu14.18 K322A Ab that was filtered through Sartobind Q membrane filter was diafiltrated into PBS, pH 6.0, 100 mM Arginine and concentrated to approximately 10 mg/ml using a 30 kD Sartocon Slice 200 (Sartorius) membrane. To the concentrated Hu 14.18 K322A Ab, PBS pH 6, 100 mM Arginine, 10% Tween 80 buffer was added so that the final concentration of Tween 80 in the protein was 0.03%. The Hu 14.18 K322A Ab in formulation buffer was sterile filtered through a 0.22 μm Stericup filter (Millipore) and stored at 4 °C.

## Western blot

Cells were washed with 1x PBS pH = 7.4 while keeping the plate on ice. Removal of 1X PBS is followed by the addition of 2x sample loading buffer [0.1 M Tris HCl (pH 6.8), 200 mM dithiothreitol, 0.01% bromophenol blue, 4% SDS, and 20% glycerol] and scrapped from 6-well plate into a 1.5 mL Eppendorf tubes. Keeping the samples on ice, cell lysates were sonicated once with 30% amplitude output (sonics, VIBRA cell) for 5–10 seconds, followed by 10-minute heating at 95 °C. Cell lysates are then centrifuged at 1000 rpm for 2 min at room temperature before adding 10 μL of lysates into a 4–15% Mini-Protean TGX stain-free gel (BioRad, cat #: 4568086) and transferred into a methanol-soaked polyvinylidene difluoride membrane from (BioRad, cat# L002043B). Membrane is blocked using non-fat dry milk mixed with PBS supplemented with 0.1% Tween 20 (PBST) for one hour while on gentle shaking. Membrane is then washed with PBST for 15 min at room temperature, changing the wash solution every 5 min. Membrane is than incubated with anti-RBM39 (Sigma, cat#:HPA001591, RRID: AB_1079749;1:1000), anti-DCAF15 (St John's Laboratory, STJ194383) and anti-HSP90 (Santa Cruz: Cat#:SC13119, (1;1000 dilution, RRID#AB_675659) at 4 °C overnight with gentle shaking. The following day, the primary antibody is removed and washed for 5-minute intervals of PBST for a total of 15 min before adding anti-mouse or rabbit horseradish peroxidase-conjugated secondary antibody (Invitrogen: Ref: A24531 GTXRB IGG F(AB)'2 HRP XADS: 1:4000) mixed with blocking buffer. Membrane is incubated for 1 hour at room temperature with gentle shaking and washed with three 5-minute washes with PBST at room temperature. Lastly, the membranes were incubated for 2 min at room temperature with a 1:1 of Super Signal West Pico PLUS luminol/ enhancer [ThermoScientific: cat#: 1863096] and stable peroxide [ThermoScientific: cat#1863097] before visualizing antigen-antibody complexes on the Odyssey Fc Imaging System (serial #: OFC. 1358).

## Derivation of indisulam-resistant SK-N-AS cells from indisulam-resistant SK-N-AS tumors

The vehicle-treated SK-N-AS tumor ($n = 3$) and indisulam-resistant SK-N-AS tumors ($n = 3$) were excised and placed in a sterile tube containing phosphate-buffered saline (PBS) on wet ice during transport from the animal research facility to the research laboratory. Tumor samples were manually minced using a sterile scalpel and underwent an enzymatic digestion with collagenase IV (2 mg/ml; in 25 ml of RPMI medium) for 1 hour in a 37 °C rotor (Robbins Scientific Corporation, model 2000). After digestion, cells were filtered using a 70-μm sterile strainer and cultured in RPMI medium with 10% FBS and 1% penicillin and streptomycin.

## Cell viability assay

Cell viability for SK-N-AS naive and Indisulam-resistant cell lines (R1-R3) was determined by PrestoBlue cell viability reagent [Invitrogen, Cat#: A13262]. Following the manufacturer's instructions, cells were plated in two white 96-well plates (PerkinElmer cat#6005680). The first plate contained 90 μL of cells and served as a control plate before the treatment is added. 10 μL of PrestoBlue reagent was added to the cells to assess baseline activity. The second plate followed a 5-day treatment to assess $IC_{50}$. Cells were plated in 45 μL of 3000 cells/well and left to incubate at 37 °C with 5% $CO_2$ for 24 h. The following day, cells are treated with serially diluted 45 μL of 2x concentrations of indisulam until the final concentration within the well resulted in: 0 nM, 0.16 nM, 0.5 nM, 1.5 nM, 4.5 nM,13.7 nM, 41 nM,123 nM, 370 nM,1.1 μM, 3.3 μM, 10 μM. Each concentration is tested in 8 replicates. After 5-day treatment, 10 μL of 10X PrestoBlue reagent is added to plates and incubated at 37 °C with 5%$CO_2$ for 30 min. Fluorescence is read at 560-nm excitation/590-nm emission using the Synergy H1 microplate reader. Fluorescent values were normalized to the vehicle and graphed on PRISM9 GraphPad software. Normalized data is transformed into log10 and analyzed using nonlinear regression log(inhibitor) vs response (three parameters).

## In vitro NK-cell killing assay with caspase 3/7 dye

A total volume of 50 μL of RPMI media supplemented with 10% FBS was mixed with SKNAS cells, which were seeded at 5000 cells per well in a Corning 96-well Falcon plate (Cat. No. 353072) and allowed to adhere for 6 h. Once the cells had adhered to the plate, they were treated with either 50 μL of 0.05% DMSO (vehicle) or 50 μL of 300 nM Indisulam (resulting in a final concentration of 150 nM Indisulam). Next, NK92 cells were seeded at 1 × 106 cells in a T-25 flask containing MEM Alpha (1X, Cat. No. 0032) supplemented with L-glutamine, horse serum, folic acid, inositol, 2-mercaptoethanol, and IL-2. Both SKNAS and NK92 cells were incubated with either vehicle or Indisulam treatment for 24 h. After the incubation, the media from the 96-well plate were removed, and the cells were washed twice with warm 1X PBS (Gibco, PBS pH 7.4, Cat. No. 01751). Following the washes, 100 μL of RPMI media supplemented with 10% FBS was added to all wells, and the plate was incubated in the IncuCyte incubator for 30 min. SKNAS cells (Effector to target-3:1), both with and without Indisulam pre-treatment, were incubated with Sartorius Incu-Cyte Caspase 3/7 Green Dye (Cat. No. 4440) at a 1:1000 dilution. Afterward, 100 μL of NK92 cells in MEM Alpha were added to the plate. Imaging was conducted using 2-hour scans.

## GD2 flow cytometry analysis

The naïve and indisulam-resistant SK-N-AS cells were harvested as a single-cell suspension, washed with PBS, and blocked with blocking buffer. Subsequently, cells were incubated with anti-GD2 (Sigma Aldrich, Catalog # MAB2052) or Isotype control mAb (Thermo Fisher Scientific, Catalog # 14-4724-85) (1 μg per $10^6$ cells) master mix for 1 hour on ice. Following this, cells were washed with PBS supplemented with 5% FBS. Next, the cells were incubated with Alexa Fluor 594-conjugated anti-mouse IgG (1:1000) for 60 min in the dark. After three

washes, cells were spun down and resuspended in PBS for analysis. The experiment was acquired using BD Fortessa (San Jose, CA). Alexa Fluor 594 signals were detected with a 582/15 band-pass filter under a 562 nm laser excitation. Signal overlay analysis was conducted using FlowJo software for flow cytometry data.

## Sanger DNA sequencing of RBM39
Total RNA was isolated from tissue samples as per the manufacturer's instructions of the RNeasy Mini Kit (Qiagen, catalog no. 74106). cDNA was prepared from 1 µg of total RNA using a cDNA synthesis kit according to the manufacturer's protocol (Thermo Fisher Scientific, catalog no. 18091050). RBM39 gene was amplified by PCR using the cDNA as the template along with the following primers (forward: 5'-atttctagagccaccatggcagacgatattgatattgaagcaatgc-3'; reverse: 5'-attg-gatcctcatcgtctacttggaaccagtagc-3') using the Phusion High Fidelity PCR Kit (New England Biolabs, cat#E0553S). The resultant PCR product was gel-purified using a Qiagen gel extraction kit (Qiagen, cat#28706) according to the manufacturer's protocol. The DNA was sent for Sanger sequencing at Hartwell Center in St. Jude using the following primers: primer 288 F, ACAGAAGTCCTTACTCCGGACC; primer 388 R, ACTTTTGCTTCGGGAACGTCG; primer 602F, GTCGATGTTAGCTCAGTGCCTC. Primer 602F is used to detect the RBM39 mutation in the RRM2 motif.

## CRISPR library construction, screening and analysis
The customized gRNA libraries (epigenetics, transcription factors, and kinases, Supplementary files 1-3) were constructed by following the published protocol[106]. The 20 bp gRNAs prepended with extra sequences (TATCTTGTGGAAAGGACGAAACACCG for 5' and GTTTTAGAGCTAGAAATAGCAAGTTAAAAT for 3') were synthesized by GenScrip using GenTitan™ Oligo Pool service. Library amplification and Gibson Assembly into the pLentiguide-Puro backbone (Addgene #52963) was performed as previously described[107]. The plasmid library was amplified and validated in the Center for Advanced Genome Engineering at St. Jude Children's Research Hospital as described in the Broad GPP protocol (https://portals.broadinstitute.org/gpp/public/resources/protocols) except EnduraTM DUOs (Lucigen) electrocompetent cells were used for the transformation step. NGS sequencing was performed in the Hartwell Center Genome Sequencing Facility at St. Jude Children's Research Hospital. Single-end, 100-cycle sequencing was performed on a NovaSeq 6000 (Illumina). Validation to check gRNA presence and representation was performed using calc_auc_v1.1.py (https://github.com/mhegde/) and count_spacers.py, with qualities passing criteria: Percentage of guides that matched was 82.1%, 80.1% and 80.3%, respectively for epigenetic, kinase and transcriptional factor library (recommended >70%); Percentage of undetected guides was 0 for all three (recommended <0.05); Skew ratio of top 10% to bottom 10% was 1.56, 1.86, 2.66 respectively for epigenetic, kinase and transcriptional factor library (recommended <10); Area under Curve was 0.56, 0.57, 0.6, respectively for epigenetic, kinase and transcriptional factor library (recommended <0.7). For the lentivirus production of CRIPSR libraries, we used PEI-PRO (Polyplus, Cat #115-100) to transfect human embryonic kidney (HEK) 293 T cells with CRISPR-gRNA pool library constructs and accompanying helper plasmids: pCAG-kGP1-1R, pCAG4-RTR2, and pHDM-G. The following day, the plates were rinsed, and fresh medium was provided. Over a 72-hour period post-transfection, we gathered replication-deficient lentiviral particles every 12 h. These particles were concentrated through centrifugation at 28,000 rpm, followed by reconstitution in RPMI medium and subsequent storage at −80 °C in small aliquots. Estimations for the MOI were made for each library, aiming for an approximate -0.3 MOI. To achieve this, cells were transduced with the pooled CRISPR library in the presence of polybrene (8 µg/ml). After 24 h, cells underwent selection using puromycin (2 µg/ml) for an additional 48-hour duration to enable selection. Subsequently, cell counting was conducted, and live cells were quantified to determine the MOI.

SK-N-AS-Naïve and SK-N-AS-indisulam resistant cell (SKNAS-R2) derived from tumors underwent transduction utilizing Cas9 lentivirus (Addgene, Cat #52962) followed by a selection process employing blasticidin at a concentration of 10 µg/ml (Sigma, Cat #203350) for 7 days. Validation of Cas9 protein expression was performed via western blot analysis. Cas9 expressing SK-N-AS and SKNAS-R2 were transduced with the customized human CRISPR Knockout pooled library, known as epigenetic library, housing 4032 unique sgRNA sequences targeting 480 human genes (including 8 sgRNAs per gene and 192 non-targeting controls), or transcription factor library, housing 10744 unique sgRNA sequences targeting 2558 human genes (including 4 sgRNAs per gene and 512 non-targeting controls) or Kinase library, housing 4032 unique sgRNA sequences targeting 480 human genes (including 6 sgRNAs per gene and 192 non-targeting controls). To ensure effective barcoding of individual cells, we maintained a low MOI (-0.3). Post-transduction, cells underwent puromycin selection (2 µg/mL, Millipore Sigma) for 48 h, followed by removal of dead cell debris and maintenance in complete medium. The transduced cells were subjected to treatment with both vehicle and indisulam 250 nM, with concentrations chosen based on PrestoBlue assay (250 nM which show minimal effect on cell proliferation in resistant cells). A minimum of $5 \times 10^6$ cells for the transcription factor library and $3 \times 10^6$ for the epigenetic and Kinase libraries were collected for genomic DNA extraction, ensuring over 400x coverage of the libraries used for screening. Genomic DNA extraction was performed using a DNeasy Blood & Tissue Kit (Qiagen, Cat#69506) and quantified using a Nanodrop instrument. The sgRNA sequences were amplified via PCR using NEB Q5 polymerase (New England Biolabs, cat# M0491S). Purification of PCR products was done using AMPure XP SPRI beads (Beckman Coulter, Cat# A63881), and quantification was carried out using a Qubit dsDNA HS assay (Thermo Fisher Scientific, Cat# Q32851). Sequencing encompassed 4.5 million reads for the transcription factor library and 5 million reads for the epigenetic and Kinase libraries on an Illumina HiSeq sequencer at the Hartwell Center Genome Sequencing Facility at St. Jude Children's Research Hospital. NGS sequencing was carried out employing single-end, 100-cycle sequencing on a NovaSeq 6000 (Illumina). Validation to affirm gRNA presence and representation was conducted using calc_auc_v1.1.py (https://github.com/mhegde/) and count_spacers.py. CRIPSR data analysis was performed by using MAGeCK-VISPR software.

## RNA extraction from tumor tissues
Mice were humanely euthanized using $CO_2$. Tumor samples were excised promptly and flash-frozen in liquid nitrogen, then preserved at −80 °C until future utilization. A portion of the frozen sample was introduced into 500 µl RLT buffer (containing β-mercaptoethanol) and homogenized using a homogenizer (Pro Scientific). The resulting mixture underwent centrifugation at 13,000 rpm, and the supernatant was carefully transferred to a gDNA eliminator spin column, and further processed according to the manufacturer's instructions for RNA extraction by utilizing the RNeasy Plus Mini Kit (Qiagen, ref. #74136).

## RNA-seq and analysis
Total RNA from cells and tumor tissues were performed using the RNeasy Mini Kit (Qiagen) according to the manufacturer's instructions. Paired-end sequencing was performed using the High-Seq platform with 100 bp read length. Total stranded RNA sequencing data were processed by the internal AutoMapper pipeline. Briefly, the raw reads were first trimmed (Trim-Galore version 0.60), mapped to the human genome assembly (GRCh38) (STAR v2.7) and then the gene-level values were quantified (RSEM v1.31) based on GENCODE annotation (v31). For mouse tumors, the raw reads were first trimmed (Trim Galore

version 0.60), mapped to the mouse genome assembly (GRCm38, mm10) (STAR v2.7) and then the gene-level values were quantified (RSEM v1.31) based on GENCODE annotation (VM22). Low-count genes were removed from analysis using a CPM cutoff corresponding to a count of 10 reads and only confidently annotated (level 1 and 2 gene annotation) and protein-coding genes are used for differential expression analysis. Normalization factors were generated using the TMM method, counts were then transformed using voom and transformed counts were analyzed using the lmFit and eBayes functions (R limma package version 3.42.2). Then Gene Set Enrichment Analysis (GSEA) was carried out using gene-level log2 fold changes from differential expression results against gene sets in the Molecular Signatures Database (MSigDB 6.2) (gsea2 version 2.2.3). After mapping RNA-seq data, rMATS v4.1.0 was used for RNA alternative splicing analysis by using the mapped BAM files as input. Specifically, five different kinds of alternative splicing events were identified, i.e., skipped exon (SE), alternative 5′-splicing site (A5SS), alternative 3′-splicing site (A3SS), mutually exclusive exon (MXE), and intron retention (RI). To keep consistent, the same GTF annotation reference file for mapping was used for rMATS. For stranded RNA-seq data, the argument "--lib-Type fr-firststrand" was applied. To process reads with variable lengths, the argument "--variable-read-length" was also used for rMATS. To select statistically significantly differential splicing events, the following thresholds were used: FDR < 0.05 and the absolute value of IncLevelDifference > 0.1. For visualization, the IGV Genome Browser was used to show the sashimi plots of splicing events.

## Assay for Transposase-Accessible Chromatin using sequencing (ATAC-seq), ATAC-seq alignment, peak-calling and annotation

Library preparations for ATAC-seq were based on a published protocol with minor modifications[108,109]. Briefly, freshly cultured SKNAS cells (100,000 per sample, naïve and two resistant ones) were harvested and washed with 150 μl ice-cold Dulbecco's Phosphate-Buffered Saline (DPBS) containing protease inhibitor (PI). Nuclei were collected by centrifugation at 500 g for 10 min at 4 °C after cell pellets were resuspended in lysis buffer (10 mM Tris-Cl pH 7.4, 10 mM NaCl, and 3 mM $MgCl_2$ containing 0.1% NP-40 and PI). Nuclei were incubated with Tn5 transposon enzyme in transposase reaction mix buffer (Illumina, cat# 20034197) for 30 min at 37 °C. DNAs were purified from the transposition sample by using the Mini-Elute PCR purification kit (cat#28004, Qiagen) and measured by Qubit. Polymerase chain reaction (PCR) was performed to amplify with High-Fidelity 2X PCR Master Mix [72 °C/5 mins + 98 °C/30 s + 12 × (98 °C/10 s + 63 °C/30 s + 72 °C/60 s) + 72 °C/5 min]. The libraries were purified using the Mini-Elute PCR purification kit (Qiagen). ATAC-seq libraries were pair-end sequenced on HiSeq4000 (Illumina) in the Hartwell Center at St Jude Children's Research Hospital, Memphis, TN, USA.

The ATAC-seq raw reads were aligned to the human reference genome (hg38) using BWA[110] to and then marked duplicated reads with Picard (version 1.65), with only high-quality reads kept by samtools (version 1.3.1, parameter "-q 1 -F 1024")[111]. Reads mapping to mitochondrial DNA were excluded from the analysis. All mapped reads were offset by +4 bp for the + strand and −5 bp for the − strand[108]. Peaks were called for each sample using MACS2[112] with parameters "-q 0.01 −nomodel −extsize 200 −shift 100". Peaks were merged for the same cell types using BEDtools[113]. Peak annotation was performed using HOMER[114]. All sequencing tracks were viewed using the Integrated Genomic Viewer (IGV 2.3.82)[115].

## CUT&Tag and analysis

CUT&Tag DNAs from SKNAS-naïve, IR1, and IR2 cells and PDX-SJNB14-ctrl and PDX-SJNB14-IR tumors were prepared by following the protocol as described previously[116] (https://www.protocols.io/view/bench-top-cut-amp-tag-bcuhiwt6?step=1) with minor modifications. Briefly, for SJNB14-ctrl and SJNB14-IR tumor, nuclei extraction is following the

protocol of isolating nuclei from frozen tissues (https://support.missionbio.com/hc/en-us/article_attachments/4421562098967). For SKNAS-naïve, IR1 and IR2 cells were washed with wash buffer (20 mM HEPES pH 7.5; 150 mM NaCl; 0.5 mM Spermidine; 1× Protease inhibitor cocktail). Nuclei were isolated with cold NE1 buffer (20 mM HEPES−KOH, pH 7.9; 10 mM KCl 0.1%; Triton X-100; 20% Glycerol, 0.5 mM Spermidine; 1x Protease Inhibitor) for 10 min on ice. Nuclei were collected by 600 x g centrifuge and resuspended in 1 ml washing buffer containing with 10 μL of activated concanavalin A-coated beads (Bangs Laboratories, BP531) at RT for 10 min. Bead-bound nuclei were collected with placing tube on the magnet stand and removing the clear liquid. The nuclei bound with bead were resuspended in 50 μL Dig-150 buffer (20 mM HEPES pH 7.5; 150 mM NaCl; 0.5 mM Spermidine; 1× Protease inhibitor cocktail; 0.05% Digitonin; 2 mM EDTA) and incubated with a 1:50 dilution of H3K27ac (Abcam, ab4729; RRID:AB_2118291) overnight at 4 °C. The unbound primary antibody was removed by placing the tube on the magnet stand and withdrawing the liquid. The primary antibody-bound nuclei bead was mixed with Dig-150 buffer 100uL containing guinea pig anti-Rabbit IgG antibody (Antibodies, ABIN101961; RRID:AB_10775589) 1:100 dilution for 1 hour at RT. Bead-bound nuclei were washed using the magnet stand 3× for 5 min in 1 mL Dig-150 buffer to remove unbound antibodies. A 1:100 dilution of pA-Tn5 adapter complex was prepared in Dig-300 buffer (20 mM HEPES, pH 7.5, 300 mM NaCl, 0.5 mM Spermidine, 0.05% Digitonin, 1× Protease inhibitor cocktail). After removing the liquid on the magnet stand, 100 μL mixture of pA-Tn5 and Dig- 300 buffer was added to the nuclei-bound beads with gentle vortex and incubated at RT for 1 h. After 3 × 5 min in 1 mL Dig-300 buffer to remove unbound pA-Tn5 protein, nuclei were resuspended in 250 μL Tagmentation buffer (10 mM $MgCl_2$ in Dig-300 buffer) and incubated at 37 °C for 1 h. EDTA 10 μL of 0.5 M, 3 μL of 10% SDS and 2.5 μL of 20 mg/mL Proteinase K were added to stop tagmentation and incubated at 55 °C for 1 hour. DNA libraries were then purified with SPRIselect beads (Beckman Coulter, B23318) following manufacture instruction and then dissolved in water for Illumina sequencing.

The sequencing raw reads were aligned to the human reference genome (hg38) using BWA (version 0.7.12; BWA aln+sampe for CUT&Tag data). Duplicate reads were marked and removed by Picard (version 1.65). For CUT&Tag, only properly paired uniquely mapped reads were extracted by samtools (version 1.3.1 parameters used were -q 1 -f 2 -F 1804) for calling peaks and generating bigwig file. Narrow peaks were called by MACS2 (version 2.2.7.1) with parameters of " -t cut_tag_file -q 0.05 -f BED --keep-dup all". Peak regions were defined to be the union of peak intervals in replicates from control or treated cells, respectively. For peak overlap analysis, mergeBed (BEDtools version 2.25.0) was used to combine overlapping regions from multiple peak sets into a new region, and then a custom script was used to summarize common or distinct peaks and visualize in a Venn diagram. Promoter regions were defined as the regions 1.0 kb upstream and 1.0 kb downstream of the transcription start sites based on the human RefSeq annotation (hg38). Genomic feature annotation of peaks was done by annotatePeaks.pl, a program from the HOMER suite (v4.8.3, http://homer.salk.edu/homer/). We used genomeCoverageBed (BEDtools 2.25.0) to produce genome-wide coverage in a BEDGRAPH file and then converted it to a bigwig file by bedGraphToBigWig. The bigwig files were scaled to 15 million reads to allow comparison across samples. To show the average of several replicates as a single track in the browser, the bigwig files were merged to a single average bigwig file using UCSC tools bigWigtoBedGraph, bigWigMerge, and bedGraphToBigWig. The Integrated Genomics Viewer (IGV 2.3.82) was used for visual exploration of data. The HOMER software was used to perform de novo motif discovery as well as a check the enrichment of known motifs in a set of given peaks. Motif density histograms were created using HOMER for target regions. Background regions were generated by selecting DNA sequences of equal length at 10 kb

downstream of the target regions. The motif density at target regions was normalized to that at the control regions.

## Single-cell RNA sequencing cluster identification

The *TH-MYCN/ALK*[F1178L] mice were treated with vehicle or 25 mg/kg of indisulam for 3 days. The mouse tumor samples were collected for single-cell Multiome sequencing. Chromium Next GEM Single Cell Multiome ATAC + Gene Expression Reagent Kits (10X Genomics) were used to generate scRNA-seq and scATAC-seq libraries following the manufacturer's standard protocols. Sequencing was performed on an Illumina NovaSeq 6000. Single cell MultiOME ATAC libraries, a 50 × 49-bp paired-end configuration with single Index i7 −8cy and i5 −24cy. Single cell MultiOME RNA libraries were sequenced using a 28 × 90-bp paired-end configuration. The ScMultiome dataset were initially processed using Cell Ranger ARC v2.0.0 (10x Genomics), which aligned fastq files to the mm10 reference genome and generated a cell-versus-gene matrix of UMI counts for a gene expression assay and a cell-versus-fragment matrix for an ATAC assay. Subsequent analyses were performed using R (4.2.2) and Seurat package (5.0.0)[117,118] and Signac[119] with default parameters unless stated otherwise. For scRNA-seq data of each sample, cells with expression less than 200 genes were excluded. The maximum number of unique molecular identifiers (UMIs) was set at 10,000 counts, and the mitochondrial ratio was restricted to less than 0.1. Raw count matrices were normalized by total counts per cell and log-transformed. Principle Component Analysis (PCA) was performed on the scaled variable gene matrix, and the top 30 principal components were used for clustering and dimensional reduction. Batch effects between samples were removed using Harmony[120]. Clusters were defined using the shared nearest neighbor (SNN) method at a resolution of 0.1. The UMAP algorithm was applied to visualize cells in a two-dimensional space. Cells were annotated using a data transfer approach[117] implemented in Seurat, with cell type labels and single-cell transcripts by J Wienke et al.[85] as a reference. Additionally, the assigned labels were verified using SingleR (1.4.1)[121] with reference datasets (Immunological Genome Project (ImmGen) and Mouse RNA-seq) provided by celldex (1.0.0)[121]. Each cluster was annotated with a single cell type, determined by a majority vote on cell identifiers associated with the cell type labels within the cluster.

## Flow cytometry for immune cell profiling of transgenic TH-MYC/ALK[F1178L] tumors

To generate tumor single cell suspensions, tumors were excised, mechanically dissociated, and transferred to basal medium containing 0.1% Collagenase Type IV and 150 mg/mL DNase I, incubated for 30-minute shaking at 37 °C, then passed through a 70-μm nylon mesh. Cells were treated with ACK red blood cell lysis buffer and resuspended in PBS prior to further analysis. Cells were treated with a 1:100 dilution of CD16/32 blocking antibody (clone 2.4G2, Tonbo) for 10 min at 4 °C, then stained for 30 min at 4 °C with the following antibodies/dyes: B220 AF647 (clone RA3-6B2, BioLegend, lot# B301694, dilution 1:200) for B cells, CD3 PE/Dazzle594 (clone 17A2, BioLegend, lot# B307674, dilution 1:200) for T cells, CD4 BV570 (clone RM4-5, BioLegend, lot# B341697, dilution 1:200) for CD4 T cells, CD8a BV711 (clone 53-6.7, BioLegend, lot# B100747, dilution 1:200) for CD8 T cells, CD45 PE/Cy5 (clone 30-F11, BioLegend, lot# B340539, dilution 1:200), CD49b BV605 (clone HMα2S, BD Biosciences, lot# 1305998, dilution 1:200) for NK cells, Ghost Dye Violet 510 viability dye (Tonbo, 1:400). Data were acquired with a Cytek Aurora spectral flow cytometer and analyzed in FlowJo v10 (Treestar).

## Flow cytometry for NK cells in c-MYC tumors implanted in Rag2[-/-] mice

Tumors were excised, mechanically dissociated and transferred to basal medium containing 0.1% Collagenase Type IV (Worthington Biochemical Corporation, Cat#CLS-4) and 150 mg/mL DNase I

(Promega, Cat# M6101), incubated for 30-minute shaking at 37 °C, then passed through a 70-μm nylon mesh. Cells were treated with ACK red blood cell lysis buffer and resuspended in PBS prior to further analysis. Cells were incubated in C-RPMI containing Cell Stimulation Cocktail with protein transport inhibitors (eBioscience, cat# 00-4975-93) or protein transport inhibitors alone (eBioscience, cat# 00-4980-93) for 4 hr at 37 °C. Cells were then treated with a 1:100 dilution of Fc receptor blocking antibody (clone 2.4G2, Tonbo) for 10 min at 4 °C, and stained for 30 min at 4 °C with the following antibodies/dyes: CD3 PE/Dazzle594 (clone 17A2, BioLegend, lot# B307674, dilution 1:200), CD45 PE/Cy5 (clone 30-F11, BioLegend, lot# B340539, dilution 1:200), CD49b BV605 (clone HMα2S, BD Biosciences, lot# 1305998, dilution 1:200), Ghost Dye Violet 510 viability dye (Tonbo, lot# D0870071921133, 1:400). Cells were fixed and permeabilized with Cyto-Fast Fix/Perm Buffer Set (BioLegend, cat# 426803) according to manufacturer's instructions and stained with IFN-γ BV785 (clone XMG1.2, BioLegend, lot# B334622, dilution 1:100) for 30 min at 4 °C. After washing cells, data were acquired with a Cytek Aurora spectral flow cytometer. and analyzed in FlowJo v10 (Treestar).

## In vivo plasma pharmacokinetics study

The plasma pharmacokinetic (PK) profile of the RBM39 degrader indisulam (E7070) was evaluated in normal female CB17/SCID mice (Taconic Biosciences), approximately 12 weeks in age. Indisulam (MedKoo, CAT# 201540, LOT# KB60825) was dissolved in 5% DMSO, 4.75% Tween 80 in saline, at 5 mg/mL for a 25 mg/kg free base equivalent dose as a 5 mL/kg IV bolus dose via the tail vein. Two survival blood samples were obtained from each mouse via retro-orbital plexus using Drummond EDTA 75 mm capillary tubes, and a third final sample by cardiac puncture, all using KEDTA as the anticoagulant. Samples were obtained at various times up to 36 h post-dose, immediately processed to plasma, and stored at −80 °C until analysis. The remaining dosing solution was submitted for verification of potency and chemical and physical stability during the study period. Plasma samples were analyzed for indisulam using a qualified liquid chromatography–tandem mass spectrometry (LC-MS/MS) assay. Plasma calibrators and quality controls were spiked with solutions, corrected for salt content and purity as necessary, and prepared in DMSO.

Plasma samples, 25 μL each, were protein precipitated with 100 μL of 30 ng/mL U-104 (MedChemExpress, CAT# HY-13513/CS-4495, LOT# 17909) in methanol as an internal standard (IS). A 3 μL aliquot of the extracted supernatant was injected onto a Shimadzu LC-20ADXR high-performance liquid chromatography system via a LEAP CTC PAL autosampler. The LC separation was performed using a Phenomenex Kinetex EVO C18 (2.6 μm, 50 mm × 2.1 mm) column maintained at 50 °C with gradient elution at a flow rate of 0.6 mL/min. The binary mobile phase consisted of UP water-methanol-formic acid, (90:10:0.1 v/v) and methanol-formic acid (100:0.1 v/v) in reservoir B. The initial mobile phase consisted of 20% B with a linear increase to 50% B in 1.5 min followed by a 1.0-minute hold at 50% B. The column was then rinsed for 2.0 min at 100% B and then equilibrated at the initial conditions for 2.0 min for a total run time of 6.5 min. Under these conditions, the analyte and IS eluted at 1.63 and 1.07 min, respectively. Analyte and IS were detected with tandem mass spectrometry using a SCIEX QTRAP 5500 in the negative ESI mode, and the following mass transitions were monitored: indisulam 384.0 → 320.2, U-104 308.1 → 171.1. The method qualification and bioanalytical runs all passed the acceptance criteria for non-GLP assay performance. A linear model (1/X2 weighting) fit the calibrators across the 0.250 to 500 ng/mL range, with a correlation coefficient (R) of ≥ 0.9993. The lower limit of quantitation (LLOQ), defined as a peak area signal-to-noise ratio of 5 or greater versus a matrix blank with IS, was 0.250 ng/mL. Sample dilution integrity was confirmed, and no matrix effects were observed in blank experimental CB17/SCID plasma. The intra-run precision and accuracy were ≤ 6.32% CV and 94.7% to 105%, respectively.

Plasma concentration-time (Ct) data for indisulam were grouped by matrix and nominal time point, and summary statistics were calculated. The arithmetic mean Ct values were subjected to non-compartmental analysis (NCA) using Phoenix WinNonlin 8.1 (Certara USA, Inc., Princeton, NJ). The IV bolus model was applied, and the area under the Ct curve (AUC) values were estimated using the "linear up log down" method. The terminal phase was defined as at least three time points at the end of the Ct profile, and the elimination rate constant (Kel) was estimated using an unweighted log-linear regression of the terminal phase. The terminal elimination half-life (T1/2) was estimated as 0.693/Kel, and the AUC from time 0 to infinity (AUCinf) was estimated as the AUC to the last time point (AUClast) + Clast (predicted)/Kel. Other parameters estimated included observed maximum concentration (Cmax), time of Cmax (Tmax), concentration at the last observed time point (Clast), time of Clast (Tlast), back extrapolated initial concentration (C0), clearance (CL = Dose/AUCinf), and volume of distribution at steady state (Vss). A clinically relevant dose (CRD) for mice was estimated from unbound plasma PK and exposure. The CRD was defined as the mouse dose regimen achieving a predicted mean unbound plasma average concentration over time (Cavg,t) similar to humans with the selected dose level and regimen of interest. The Cavg,t was calculated as the AUC from 0 to time t divided by time t. Dose proportional, linear, and time-invariant PK for mice was assumed. Human and mouse plasma protein binding were assumed to be similar since data were not available. This corresponds to the clinical relevance approach proposed by Spilker[122], which uses unbound plasma average steady state concentrations to define CRDs. Some latitude in dose rounding was permitted in the CRD recommendation, and an unbound exposure within 2-fold of the clinical target was considered acceptable. Additional considerations influenced the final recommended mouse dose, including mouse dosing regimens prevalent in the literature and the tolerability of the compound in mice.

## Animals and Therapies

Sex as a biological variable. Both genders of Th-MYCN/ALK[F1178L], C57BL/6, Rag2[-/-] and NSG mice were used. All murine experiments were done in accordance with a protocol approved by the Institutional Animal Care and Use Committee of St. Jude Children's Research Hospital. Subcutaneous xenografts were established in 4-6 weeks old CB17 SCID mice (CB17 scid, Taconic) or NOD.Cg-Prkdcscid Il2rgtm1Wjl/SzJ (NOD scid gamma (NSG, St Jude Children's Research Hospital, bred in house ARC) mice by implanting $5 \times 10^6$ cells (SK-N-AS and SIMA) in Matrigel. Patient-derived xenograft (SJNB14 PDX) and mouse-derived xenograft (C-MYC MDX) tumors were sectioned into small fragments using a Razor blade (Electron Microscopy Sciences, Cat#71980), loaded into a syringe (21-gauge needle, BD: Ref:305167), and subcutaneously injected into mice. Following genotyping, *TH-MYCN/ALK[F1178L]* mice were gender-segregated and assigned to imaging groups. Inclusion criteria were the presence of the *TH-MYCN* allele, heterozygosity for one *ALK[F1178L]* allele, and an age range from 1 to 3 weeks postweaning. Either gender was used and noted. Criteria for analysis were the presence or absence of an ultrasound-detectable tumor (quantified by staff not directly involved in this study) at any point within a 7-week period, where mice were imaged once per week. To mitigate gender bias, both genders were employed in NSG, *Rag2[-/-]*, or transgenic *TH-MYCN/ALK[F1178L]* mouse models. Tumor measurements were done weekly using electronic calipers for xenograft models, and volumes calculated as π/6 × d3, where d is the mean of two diameters taken at right angles. The starting treatment points and dosing schedules were documented in each figure legend. Indisulam was administered to CB17 SCID or NSG mice via tail vein injection, while *TH-MYCN/ALK[F1178L]*, C57BL/6, or *Rag2[-/-]* mice received treatment through IP injection.

For ATR inhibitor gartisertib in combination with indisulam, once the tumor volume reached around 200 mm³, mice were randomly assigned to 4 groups (*n* = 5 mice per group). Mice were administered with vehicle (15% Captisol /0.1 M hydrochloride acid, oral gavage), indisulam (10 mg/kg, tail vein injection, 5 days on and 2 days off, for three weeks), gartisertib (10 mg/kg/day, oral gavage, once per week for three weeks), and indisulam (10 mg/kg, tail vein injection, for three weeks) combined with gartisertib (10 mg/kg/day, oral gavage once per week for three weeks) except the mice that reached humane endpoints. The groups with indisulam alone and combination therapy were given another two-week cycle treatment when the tumors in mice with indisulam alone treatment relapsed. The tumor volume and mice weight were measured twice in a week. Mice were euthanized when the tumor volume reached 20% of the body weight or the mice became moribund.

## Ultrasound imaging of transgenic TH-MYCN/ALK[F1178L] mouse

Fur was removed from the ventral side of each animal using Nair. Technicians in the St. Jude Center for In Vivo Imaging and Therapeutics performed ultrasound scanning on mice weekly using VEVO-3100 and determined tumor volumes using VevoLAB 5.7.1 software. All ultrasound data were acquired in a blinded fashion.

## Quantification and statistical analysis

All data in this study are displayed as the mean ± SEM, unless indicated elsewhere. Comparison between the two groups was determined using Student's *t*-test. The Wilcoxon rank sum test (two-sided) was used to compare the tumor volumes between the two groups at every time point. P-values across multiple time points were adjusted for multiple comparisons using the Benjamini-Hochberg method. Kaplan-Meier survival was analyzed using log-rank (Mantel-Cox) method in the Prism program.

## Ethics statement

All research was conducted in accordance with relevant ethical guidelines. Animal procedures were reviewed and approved under the protocol 615 issued to Jun Yang in accordance with the guidelines outlined by the St Jude Children's Research Hospital Institutional Animal Care and Use Committee (IACUC). Mice were housed with ambient temperature and humidity with 12 h light /12 h dark cycle controlled under specific-pathogen-free conditions (SPF) at the St Jude Children's Research Hospital mouse facility. Mice were allowed to feed and drink ad libitum. The maximum tumor volume is limited to 20% of the mouse body weight. If the tumor volume had not yet reached this endpoint during one measurement but exceeded 20% of body weight at a subsequent measurement, the mice were immediately euthanized, as permitted by the St. Jude IACUC protocol. For subcutaneous tumor models, tumor volume was estimated using a digital caliper by measuring the length and width (including the mouse skin). This is because we had to monitor tumor growth longitudinally; we did not excise and weigh tumors directly. Instead, we calculated tumor volume using the equation: $V = (\pi/6) \times [(W+L)/2]^3$, where L is the length and W is the width, with the assumption that $1 \text{ cm}^3 \approx 1 \text{ g}$. However, we acknowledge that the actual tumor weight may be much lower than this estimate, as subcutaneous volume measurements included skin thickness, and necrosis within tumors can make them lighter than the calculated value. For spontaneous tumor models, three-dimensional ultrasound imaging was performed using the Vevo system (FUJIFILM VisualSonics, Inc.). High-resolution, near-isotropic datasets were acquired and analyzed using Vevo Lab software. Tumor perimeters were manually traced in serial image slices, and calibrated volumes were calculated by the software through 3D reconstruction. Transgenic mice were euthanized through $CO_2$ inhalation with 3 Liters/min in the mouse cage and followed by cervical dislocation when moribund or determined by a veterinarian in the Animal Research Center at St Jude. For therapy studies in subcutaneous xenograft mouse models, the mice were euthanized through $CO_2$ inhalation with 3 Liters/min in the mouse cage and followed by cervical dislocation when the mice reached the endpoint.

**Reporting summary**

Further information on research design is available in the Nature Portfolio Reporting Summary linked to this article.

## Data availability

Raw RNA-seq, ATAC-seq data, CUT&Tag, and single-cell multiomics data in this study have been deposited in the Gene Expression Omnibus (GEO) and are publicly available with the SuperSeries accession number GSE251920 (GSE251918 and GSE164505 for RNA-seq, GSE251915 for CUT&Tag, GSE293865 for multiomics). GSE251920 SuperSeries [https://www.ncbi.nlm.nih.gov/geo/query/acc.cgi?&acc=GSE251920]. GSE251918. GSE164505. GSE251915. GSE293865. Target NBL data can be freely accessed through https://portal.gdc.cancer.gov/projects/TARGET-NBL[21]. St Jude PCGP data can be freely accessed through https://stjude.cloud/[59]. RNA-seq for SEQC neuroblastoma cohort (GSE62564[83]) can be accessed through https://hgserver1.amc.nl/cgi-bin/r2/main.cgi. The remaining data are available within the Article or Supplementary Information, or Source data file. Source data are provided with this paper.

## Code availability

All samples were processed by internal software AutoMapper, which includes all pre-defined NGS mapping pipelines, supported by the Center for Applied Bioinformatics (CAB), St Jude Children's Research Hospital. The pseudo-codes and detailed parameters of pre-processing pipelines for RNAseq, ATACseq, and CUT&Tag data in this study can be found in the GitHub repository (https://doi.org/10.5281/zenodo.15930501).

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

## Acknowledgements

We thank the staff in St. Jude Hartwell Center for their dedication and expertise. We thank Dr. Dongli Hu for his monthly mycoplasma testing and STR assay. This work was partly supported by American Cancer Society-Research Scholar (130421-RSG-17-071-01-TBG, J.Y.) and National Cancer Institute (1R01CA229739-01, 1R01CA266600, 1R01CA289881-01A1, 1R01CA303799, J.Y.), Comprehensive Cancer Center core grant

P30 CA021765, and the American Lebanese Syrian Associated Charities (ALSAC). The content is solely the responsibility of the authors and does not necessarily represent the official views of the National Institutes of Health. The authors have declared that no conflict of interest exists.

## Author contributions

S.S. performed animal experiments and CRISPR screening with the help of A.C. A.C. performed in vitro co-culture assay. J.F. performed ATAC-seq and CUT&TAG. H.J., J.C., and X.C. performed bioinformatics analysis. S.N. and J.E. performed scRNA-seq. L.V., P.J.M., and P.T. provided transgenic TH-MYCN/ALK$^{F1178L}$ mice, immune cell infiltration analysis. E.P. and L.C. performed animal studies for the TH-MYCN model. Q.W. analyzed the GD2 expression with help from L.H. and performed quality control of indisulam. W.Q. examined RBM39 mutations. C.L.M. provided PDX. C.L.M, M.A.W with W.V.C and B.B.F III. performed a PK study. J.P.C., J.A.S., and S.M.P-M provided CRISPR library expansion and pooled screen analysis. R.T., G.T., T.C., and M.J. performed small animal imaging and analysis. T.L. provided anti-GD mAb. T.O. provided indisulam and consultation. R. W generated the Dbh-iCre/CAG-MYC allograft model. A.J.M. provided editing and consultation. P.T., A.M.D., J.R.P., and J.Y. conceived the project. J.Y. designed the study, analyzed data, and wrote the manuscript with help from all authors.

## Competing interests

Jun Yang is a subject Editor of the British Journal of Cancer. Other authors declare no competing interests.

## Additional information

Shivendra Singh[1,17], Jie Fang[1,17], Hongjian Jin[2,17], Lee-Ann Van De Velde [3,17], Andrew Cortes[1,17], Jiani Chen[4], Sivaraman Natarajan [4], Evon Poon[5], Qiong Wu[1], Christopher L. Morton[1], Mary A. Woolard[1], Waise Quarni[1], Jacob A. Steele [6], Jon P. Connelly[6], Liusheng He[3], Rebecca Thorne[7], Gregory Turner[7], Thomas Confer [7], Melissa Johnson [7], William V. Caufield [8], Burgess B. Freeman III [8], Timothy Lockey[9], Andrew J. Murphy[1], Peter J. Murray [10], Takashi Owa[11], Shondra M. Pruett-Miller [6], Ruoning Wang [12], Louis Chesler[5], Julie R. Park [13], Andrew M. Davidoff [1,14,15], John Easton [4], Xiang Chen [4], Paul G. Thomas [3] & Jun Yang [1,15,16] ✉

[1]Department of Surgery, St. Jude Children's Research Hospital, Memphis, TN, USA. [2]Center for Applied Bioinformatics, St. Jude Children's Research Hospital, Memphis, TN, USA. [3]Department of Host Microbe Interactions, St. Jude Children's Research Hospital, Memphis, TN, USA. [4]Department of Computational Biology, St. Jude Children's Research Hospital, Memphis, TN, USA. [5]Division of Clinical Studies, The Institute of Cancer Research, London, UK. [6]Center for Advanced Genome Engineering (CAGE) and Department of Cell and Molecular Biology, St. Jude Children's Research Hospital, Memphis, TN, USA. [7]Center for In Vivo Imaging & Therapeutics, St. Jude Children's Research Hospital, Memphis, TN, USA. [8]Preclinical Pharmacokinetics Shared Resource, St. Jude Children's Research Hospital, Memphis, TN, USA. [9]Therapeutics Production and Quality, St. Jude Children's Research Hospital, Memphis, TN, USA. [10]Max Planck Institute of Biochemistry, Martinsried, Germany. [11]Eisai Inc., Nutley, NJ, USA. [12]Center for Childhood Cancer Research, Abigail Wexner Research Institute, Nationwide Children's Hospital, The Ohio State University, Columbus, OH, USA. [13]Department of Oncology, St. Jude Children's Research Hospital, Memphis, TN, USA. [14]Department of Surgery, The University of Tennessee Health Science Center, Memphis, TN, USA. [15]Department of Pathology, College of Medicine, The University of Tennessee Health Science Center, Memphis, TN, USA. [16]College of Graduate Health Sciences, University of Tennessee Health Science Center, Memphis, TN, USA. [17]These authors contributed equally: Shivendra Singh, Jie Fang, Hongjian Jin, Lee-Ann Van de Velde, Andrew Cortes. ✉e-mail: Jun.Yang2@stjude.org

