## [Transparent Peer Review file · Nature Communications]

RBM39 degrader invigorates innate immunity to eradicate neuroblastoma despite cancer cell plasticity

Corresponding Author: Dr Jun Yang

Version 0:

Reviewer comments:

Reviewer #1

(Remarks to the Author)

In this study, Singh and colleagues investigated high-risk neuroblastoma models that had developed resistance to the RBM39 degrader, indisulam. They observed a multi-directional switch of cell states in indisulam-resistant cells, reminiscent of the cellular plasticity seen in neural crest cells. Notably, the lineage alterations are coupled with epigenetic reprogramming. Additionally, indisulam induced an inflammatory tumor microenvironment, enhancing natural killer (NK) cell activity. When combined with anti-GD2 immunotherapy, indisulam produced durable, complete responses in preclinical models. Overall, this novel study proposes a promising therapeutic strategy: targeting RBM39 in combination with anti-GD2 therapy for treating high-risk neuroblastoma patients, regardless of their potential to switch cell states. This intriguing study not only contributes to the discovery of novel cell phenotypic switching under drug treatment but also introduces a novel approach to combination therapy for eradicating neuroblastoma cells, irrespective of their cell state switching potential. The discoveries are of importance and interest, and the experiments were performed competently.

I have some concerns and comments for the authors to consider addressing:

1. The authors found that indisulam resistance is associated with lineage switches in cell states. In transgenic MYCN/ALKF1178L mouse models, this switch occurs from ADRN to MES and Schwann precursor cells (Fig. 1). In human MYCN-amplified neuroblastoma xenograft model, the switch is from ADRN to MES and a melanocytic state (Fig. 2). In c-MYC overexpressed neuroblastoma xenograft model, the switch is from MES to ADRN state (Fig. 3). Additionally, indisulam treatment led to an increased percentage of NK cells, CD4 cells, and CD8 cells in transgenic MYCN/ALKF1178L mice (Fig. 7). To further investigate lineage switches, cell states, and the inflammatory tumor microenvironment induced by indisulam, the authors should consider performing a single-cell RNA sequencing experiment in one of the preclinical models. This will help dissect the cells undergoing lineage switches at the single-cell level. Moreover, single-cell RNA sequencing will aid in characterizing the immune cell infiltration, transcriptional profile of immune cells, and their functions both with and without indisulam treatment. For instance, conducting such scRNA-seq analysis in the transgenic MYCN/ALKF1178L mouse model experiment setup shown in Fig. 7E,F would be interesting. If the authors still have frozen tissues from this experiment, they could explore single-nuclei RNA sequencing as an approach.
2. Considering that indisulam treatment in mice likely impacts tumor microenvironments, immune cells, stromal cells, fibroblasts, etc. by targeting RBM39, how genetic silencing of RBM39 in neuroblastoma cells affects lineage switches and immune cell recruitment in xenografts? Please discuss.
3. In Fig. 1C, the authors described that this is a GSEA analysis result of RNA-seq data of naïve tumors vs relapsed tumors. In Fig. 1D,H,I GSEA assay, is it also naïve tumors vs relapsed tumors? Do naïve tumors vs relapsed tumors mean normalized naïve tumors RNA read counts/relapsed tumors RNA read counts? How were the genes ranked? Was it based on T-score or fold change? For me, the labels of some of the GSEA graphs in this study are confusing. For instance, in Fig. 1H, although the authors claimed that the neural crest stem cell signature genes are positively enriched in the resistant population, which is true if the GSEA graph was generated using the ranked gene list of naïve tumors vs relapsed tumors, however, from the graph, it looks like the neural crest stem cell signature genes are negatively enriched in the resistant population since beneath the graph it was labeled with Naïve on the left, and labeled Resistant on the right, and NES score is negative. I feel it might be easier to understand if the authors labeled the graph differently, such as by labeling “ranked gene list (naïve tumors vs relapsed tumors)” beneath the graph. The other way to do it is to analyze the gene list of relapsed tumors vs naïve tumors, then the neural crest stem cell signature genes NES score will be positive. Please clarify.
4. In Fig. 1J, how was the heatmap generated? Is it based on the normalized read counts, fold change, or z-score? Similarly,

how was the heatmap generated for Fig. 2E? A scale bar is needed for this heatmap.

5. On page 9, starting from line 298, the authors described many ADRN genes such as PHOX2A and MYCN as well as the RBM39 locus showed increased chromatin accessibility in the resistant cells (Fig. 3G). In the figure legend of Fig. 3G, it says naïve vs resistant, which indicates the normalized ATAC-seq read counts in naïve cells/in resistant cells. However, Fig. 3G showed that PHOX2A, SOX11 etc. are on the right side of the volcano plot, with positive fold change. From my understanding, this means that associated ATAC-seq peaks of these genes have higher signals (increased chromatin accessibility) in the naïve cells compared to the signals in the resistant cells, while PRRX2, GL2, SOX9 on the left side of the volcano plot have lower ATAC-seq signals in the naïve cells compared to the signals in the naïve cells. Please clarify the discrepancy between the main text and the graph.

6. Page 9, line 306, Figure 3G is a typo, which should be Figure 3H.

7. On page 9, starting from line 310, it says that GSEA analysis of the differential peaks near the annotated genes revealed that the 20q locus (where RBM39 resides) ranked on the top in the resistant cells while the c-MYC targets ranked on top in the naïve SK-N-AS cells (Figure 3K). However, from what I understand, Fig. 3K indicates that 20Q is high and MYC is low in naïve cells. Please clarify.

8. In Fig. 3L, how was the heatmap generated? What does the color mean? Are they based on H3K2ac signals or motif numbers? Please clarify with detailed information. Moreover, a scale bar is needed for this heatmap.

9. Fig. 7G,H, is the difference statistically significant? What is the p-value?

10. Discuss how RBP39 affects epigenetic reprogramming.

Reviewer #2

(Remarks to the Author)

In the manuscript "RBM39 degrader invigorates nature killer cells to eradicate neuroblastoma despite cancer cell plasticity," Singh et al. use a combination of genomics and Xenograft animal models to determine the mechanisms through which Neuroblastomas driven by hyperactivation of C-Myc/N-Myc develop resistance to the RBM39 degrader. In samples that are initially characterized by the adrenergic state (ADRN) they find that in response to prolonged exposure to the RBM39 degrader Indisulam, the tumor cells that have developed resistance upregulate expression of RBM39 and shift to a neural crest stem cell-like state and activate genes that are typically associated with the Mesenchymal (MES) state. They then show a Neuroblastoma line, SK-N-AS, that is associated with the MES state, when grown as a Xenograft and treated with indisulam, also acquires resistance, and these resistant cells downregulate the MES genes and upregulate genes associated with the ADRN state. Using CRISPR screens they show these "lineage-switching" events are coupled to changes in dependencies of the tumor cells on cell-type specific transcription factors, and chromatin remodelers, and they also identify general dependencies on kinases and and indisulam-resistant specific dependencies on druggable kinases. They go on to characterize the expression of GD2, a potential target of immuno-therapies, and find that GD2 remains expressed in the ADRN and MES states. By comparing tumor clearance in Xenograft models that are depleted for T, B and NK cells (NSG) to Xenograft models that are depleted for only T and B cells (Rag2^{-/-}), they suggest that NK cells are the critical mediator of tumor clearance, and that indisulam may be aiding in this response by inducing inflammation within the tumor. They then show that indisulam can be used in combination with GD2-directed immunotherapy to induce complete remission of MYC driven neuroblastomas. The development of preclinical models to show indisulam induces NK-cell-mediated tumor clearance that works in combination with GD2 inhibitors, make this manuscript of significant interest to the Neuroblastoma field, and provide an interesting example of how splicing inhibitors can be incorporated into other therapeutic regimens. However, the mechanism of action of indisulam in this context requires further clarification, and would benefit from a more uniform and complete analysis of the genomics data, and more complete characterization of the immune cells in these preclinical models.

(1) The lineage-switching mechanism of indisulam resistance that is proposed is often supported by a seemingly random choice of marker genes, "cherry picked" by the authors to support their hypothesis. In the MYCN/ALKF1178L and SK-N-AS models, the terms the "Neural crest stem cell", "NBL_MES", and "ADRN" come out of an unbiased GSEA. Is this sufficient to meet the definition of "lineage-switch" in this model? If so, the SJNB14 model fails to meet this definition, and weakens the point. To be taken seriously, the authors must do a more thorough, convincing, and uniform job of establishing the lineage identity of the cells before and after lineage switching.

(2) Related to the first point; using single-cell genomics of healthy tissues to compare marker gene expression can help to establish cell-type identities, but in Figure 1K the authors show only the expression of Sox10, and this does little to strengthen their conclusion. Can the authors color this UMAP according to the composite or average score of genes downregulated in the resistant cells (presumably the ADRN signature) and the genes upregulated in the resistant cells (The MES signature).

(3) Currently, every figure highlights a different panel of marker genes, and the data is presented in different formats making it very difficult to compare one model to another. As one example: Figure 1J shows a panel of 8 SCP genes that are upregulated in the resistant cells. Figure 2E shows a panel of ~30 genes. The only gene shown in both figure panels is S100B and yet we do believe these two Neuroblastoma models are engaging in a similar lineage-switching mechanism of resistance?

(4) Related to point 3; In figure 4 the authors show the key dependencies that increase in the SK-N-AS model after the development of resistance are HAND2, PHOX2B and TBX2. Yet, these genes are totally absent from Figure 3 which shows the changes in gene expression in the SK-N-AS model during the development of resistance and transition to an MES state.

These kind of incongruencies make this paper extremely challenging to follow and critically assess.

(5) Due to the previous issues, I am left with the impression that lineage switching in this context has no formal definition, and thus can be used to mean anything the authors choose. Please explain why a shift from ADRN does > MES state promote resistance to indisulam and a shift from an MES > ADRN state also promote resistance to indisulam. Is one state more resistant or not? Are the authors suggesting that it is the act of lineage switching that promotes resistance, and that the directionality of the lineage switch is inconsequential. If so, the authors must provide a rationale that is supported by evidence.

(6) The argument that the immune clearance is driven by NK cells is based primarily in the difference between RAG2^{-/-} and NSG mice. In addition to presence of NK cells in RAG2^{-/-} mice, these models could have many other differences that contribute to apparent changes in tumor responses to indisulam. Can the authors provide additional conclusive evidence the NK cells make the difference. For example using immune reconstitution experiments etc.

(7) How can the authors be sure that indisulam is acting on the Neuroblastoma cells and not the NK cells?

Minor points:

- Line 118 reads "that tumor cells acquired MES gene features when developed resistance." Please reword to something like: "the resistant tumor cells acquired MES gene features."
- Line 183 reads "the we projected the expression of Sox10..." this should read "then we colored the UMAP projection by the expression of Sox10".
- Line 211-212 reads "It is known that SCP is a melanocyte progenitor." This should read, "It is know that SCPs are melanocyte progenitors."
- Line 310 reads "GSEA analysis of the differential peaks." Should read "GSEA of the differential peaks."
- Line 315 reads "the RBM39 locus but not the RBM23 locus showed a greater increase in H3K27ac peak..." Should read "the RBM39 locus showed a greater increase in H3K27ac than the RBM23."
- Line 345 refers to a gene SMARCE1, but there is no SMARCE1.
- Line 360 "We identified 43, 31 and 25 kinases that were essential for either naïve..." only three numbers are listed but four states are referred to.
- Line 531 "... that drive high rate ..." please reword to "that drive high rates"

Reviewer #3

(Remarks to the Author)

This is an interesting manuscript which identifies that acquired resistance to indisulam in neuroblastoma may be due to a change in phenotypic cell states where apparently the new state amplifies RBM39 resulting in less degradation in resistant cells. Additionally, the authors suggest that RBM39 degradation may promote the activity of anti-GM2 monoclonal antibody therapy. They suggest that the therapeutic activity of RBM39 degraders may be related to NK cell mediated killing. Overall, the manuscript is novel but the data around NK cells is quite modest and not currently well supported by the data presented in the manuscript.

-In the first section of the Results and Figure 1, it is unclear if RBM39 protein level is actually increased in resistant tumors and if RBM39 is degraded in resistant tumors. Is DCAF15 still expressed in the resistant tumors throughout the manuscript?

-It is unclear if the RBM39 degrader resistant cells still rely on RBM39 or not. It appears as though resistance to these agents may be due to amplification of RBM39 which prevents effective RBM39 protein downregulation. The authors should clarify this point more in the manuscript.

-The authors claim to have tested the impact of RBM39 degradation versus RBM39 knockdown when referring to Figure S3C but Figure S3C only shows data from RBM39 degrader treatment but this figure only shows analysis of RBM39 degrader treated cells. This is an important point because it is not clear if the degree of drug-induced splicing alterations would be the same in the resistant cells treated with drug as when RBM39 is genetically suppressed.

-It is not clear if the anti-tumor response to indisulam requires NK cells. This should be directly tested. The fact that indisulam treatment showed little efficacy in NSG mice but had efficacy in C57BL/6 and Rag2^{-/-} mice does not necessarily indicate involvement in NK cell killing as NSG mice have macrophages as well.

-Are NK cells required for response to anti-GD2 mAb?

-What is the impact of indisulam treatment in vivo on NK cell numbers and phenotypes?

-The analysis of the immune microenvironment with indisulam and anti-GD2 mAb is quite limited (the authors only investigated a small number of immune cell markers by conventional FACS). A more detailed analysis of types of innate and adaptive immune cells (by more comprehensive FACS and/or RNA-sequencing analyses) would be very helpful.

Reviewer #4

(Remarks to the Author)

This is an interesting and data rich paper.

As an immunologist, the data linking Indisulam to NK cell mediated rejection of neuroblastoma is intriguing.

However, the extensive characterisation of the models (figures 1-6) is not matched at all by the depth of the immunological responses shown in Figure 7.

main problems with Figure 7.

1. The mouse strains used have different backgrounds (C57Bl6 and RAG KO are the same but the NSG mice are normally CB17). The genetic background of the mouse tumour used in this study is not evident to me but it clearly cannot match two different strains.

2. Data in figure 7B and C looks promising (strain caveat aside). For rigour, I suggest that Ab mediated NK cell depletion from the C57Bl/6 mice (anti-NK1.1) is performed, alongside NK cell adoptive therapy from C57bl/6 mice into NSG mice. Adoptive transfer experiments might also help to evaluate whether Indisulam effects are wholly tumour intrinsic or also has effects on NK cells etc.

3. The numbers of replicates and absence of stats in figs 7G and H is a concern. Are the changes in cell infiltration statistically significant (NK cells in 7H seems to have substantial variation in the three replicates).

4. In vitro experiments would enhance the conclusions; does the drug enhance killing by purified NK cells in vitro? It might be possible to perform in vitro experiments (or adoptive transfers) using NK cells from mice lacking CD16 to demonstrate the pathway of GD2 Ab recognition as suggested in Figure 8.

I found the earlier part of the paper hard to follow. Admittedly this is outside of my core expertise. Nevertheless, it lacked detail in terms of rationale and approaches used (sacrificed for brevity). It was hard to tell how many tumours were analysed in places (e.g. figure 2a). I struggled with identifying the core data revealed by the different tumour models (e.g. different markers and pathways identified in different models). Is all of this data really necessary to get to Figure 7, which the title and abstract flag up.

Again, stats are missing from many key data. For example, traces showing read counts over exons are attractive but are qualitative only without statistical testing.

Version 1:

Reviewer comments:

Reviewer #1

(Remarks to the Author)

The authors have satisfactorily addressed my comments and concerns.

Reviewer #2

(Remarks to the Author)

In their revised manuscript Singh et al. have added several experiments to test the alterations to tumors and the microenvironment in response to indisulam. They add a single-cell RNA-seq experiment that shows an increase in Schwann-like cells Mesenchyme-like and Immune cells in the indisulam resistant tumors and find that in vitro, indisulam can stimulate NK cells. They also add important western blots showing that although indisulam resistance is associated with upregulation of RBM39 at the RNA level the protein is depleted. Together these results have the potential to provide a more complete picture of the resistance mechanisms of indisulam. However, the manuscript currently suffers from issues related to (1) poor organization of the data presentation primary in the first three figures and (2) broad sweeping claims based on an N of 2, a GEM and a PDX model that are driven by different mutations, and (3) the use of different genetic models that are not properly controlled to make claims about the influence of immune cells in tumor clearance.

I understand this manuscript has been in review for over a year, and so I will provide feedback that should be able to be addressed without additional experimentation.

- The Results section is very difficult to read. Both myself and Reviewer 3 pointed out that the bulk RNA-seq comparisons made in Figure 1, 2 and 3 are overly drawn out and do not add much to the story. The authors show some genes are upregulated in one resistant tumor, and other genes are upregulated in a different resistant tumor, and then suggest cell-state changes underly resistance. This is not convincing. If the authors have evidence that the "cell state change" is important for resistance, I missed it. I suggest consolidating Figures 1-3 in order to make the points that the authors want to make clear and concise. Although the authors remove the term "lineage switch" which was certainly warranted, this does not fix my larger issue that the gene expression and epigenetic alterations they observe are not consistent between models. Currently, they show a series of gene signatures that are inconsistent from one model to the next. If the point is that the tumor diversifies the regulatory landscape in response to prolonged Indisulam, this is shown most clearly by their new single cell experiment. And bulk data and single cell data could be consolidated to make this point. Also, with the addition of their own single-cell data, do the authors still need to show SOX10 expression in isolation on a previously published dataset/UMAP? Please consider reorganizing this section and using your new single cell experiment to make the point regarding cellular plasticity as clearly and concisely as possible.

- Although Reviewer 1 asked for clear labels on panels, this still needs work. Here are some examples Fig. 1G what is "Indisulam vs copy"? Fig 3G and H the text is too small to read, and I am not sure what this adds to the narrative. Also, which side is which (right or left) is still not clearly presented for these panels. Please make your data presentation as easy to follow as possible for the readers.

- The authors state they "comprehensively defined the transcriptomic and epigenetic map of ADRN and MES types of

neuroblastomas using human and murine models." This statement needs to be revised to indicate what the authors actually did. An N of 2 with ATAC-seq and H3K27ac data is not comprehensive.

- Several of the reviewers had issues with the genetic models used to claim the NK cells are the key mediator tumor clearance. The data is interesting, but the authors must revise the text to be transparent about the limitations of these models. The addition of the in vitro cell killing assay helps to make the point that NK cells play an important role and are stimulated by indisulam.

After reading this revised manuscript several times, I appreciate the value of the current study. By reorganizing the presentation to make a clear concise narrative, the authors will ensure the study is readily digestible by a broader readership.

Reviewer #3

(Remarks to the Author)

The authors have addressed my initial comments and questions.

Reviewer #4

(Remarks to the Author)

My expertise lies in NK cell biology and I have concentrated my review on these aspects.

The authors have strengthened their arguments for the role of NK cells in the response to indisulam therapy and neuroblastoma and my concerns have been addressed.

The manuscript is interesting, a detailed study has been undertaken and the findings have implications for neuroblastoma treatment. I recommend that the article be published.

RESPONSE TO REVIEWER COMMENTS

Reviewer #1 - Neuroblastoma, epigenetics (Remarks to the Author):

In this study, Singh and colleagues investigated high-risk neuroblastoma models that had developed resistance to the RBP39 degrader, indisulam. They observed a multi-directional switch of cell states in indisulam-resistant cells, reminiscent of the cellular plasticity seen in neural crest cells. Notably, the lineage alterations are coupled with epigenetic reprogramming. Additionally, indisulam induced an inflammatory tumor microenvironment, enhancing natural killer (NK) cell activity. When combined with anti-GD2 immunotherapy, indisulam produced durable, complete responses in preclinical models. Overall, this novel study proposes a promising therapeutic strategy: targeting RBM39 in combination with anti-GD2 therapy for treating high-risk neuroblastoma patients, regardless of their potential to switch cell states. This intriguing study not only contributes to the discovery of novel cell phenotypic switching under drug treatment but also introduces a novel approach to combination therapy for eradicating neuroblastoma cells, irrespective of their cell state switching potential. The discoveries are of importance and interest, and the experiments were performed competently.

We are highly grateful for your positive comments and insightful suggestions. We have performed additional experiments and made extensive revisions to address the concerns of this reviewer.

I have some concerns and comments for the authors to consider addressing:

1. The authors found that indisulam resistance is associated with lineage switches in cell states. In transgenic MYCN/ALKF1178L mouse models, this switch occurs from ADRN to MES and Schwann precursor cells (Fig. 1). In human MYCN-amplified neuroblastoma xenograft model, the switch is from ADRN to MES and a melanocytic state (Fig. 2). In c-MYC overexpressed neuroblastoma xenograft model, the switch is from MES to ADRN state (Fig. 3). Additionally, indisulam treatment led to an increased percentage of NK cells, CD4 cells, and CD8 cells in transgenic MYCN/ALKF1178L mice (Fig. 7). To further investigate lineage switches, cell states, and the inflammatory tumor microenvironment induced by indisulam, the authors should consider performing a single-cell RNA sequencing experiment in one of the preclinical models. This will help dissect the cells undergoing lineage switches at the single-cell level. Moreover, single-cell RNA sequencing will aid in characterizing the immune cell infiltration, transcriptional profile of immune cells, and their functions both with and without indisulam treatment. For instance, conducting such scRNA-seq analysis in the transgenic MYCN/ALKF1178L mouse model experiment setup shown in Fig. 7E,F would be interesting. If the authors still have frozen tissues from this experiment, they could explore single-nuclei RNA sequencing as an approach.

We thank this reviewer for the suggestion. We have performed scRNA-seq in the MYCN/ALK^{F1178L} mouse model and indeed, we are able to validate the immune cell infiltration and enrichment of Schwann cell population after indisulam treatment. We have included the data in Figure 8A, B. and Supplementary Fig. 8-10.

Please see details from line 535-569.

		annotate_percent				
		label	grouping	count	count.total	percent
Control	Endothelial	Ctrl	138	25475	0.54%	
	Mesenchyme-like	Ctrl	204	25475	0.80%	
	Schwann-like	Ctrl	75	25475	0.29%	
	Immune	Ctrl	488	25475	1.92%	
	Tumor	Ctrl	24570	25475	96.45%	
Indisulam	Endothelial	Ind	83	6358	1.30%	
	Mesenchyme-like	Ind	269	6358	4.23%	
	Schwann-like	Ind	271	6358	4.26%	
	Immune	Ind	2492	6358	39.20%	
	Tumor	Ind	3243	6358	51.01%	

Supplementary Figure 8. Percentages of cell types in control tumors and indisulam-treatment tumors. Count = cell numbers for each cell type. Count.total = total cell numbers in each group. Percent = count/count.total

Supplementary Figure 9. Indisulam induces changes in cell composition. (a) UMAP showing 5 major cell types in Th-MYCN/ALK^{F1178L} tumors treated with vehicle control and indisulam (25mg/kg) for 5 days. (b-i) UMAP showing markers for each cell population.

Supplementary Figure 10. Indisulam induces immune cell infiltration. (a) UMAP showing Ptpcr (CD45) positive immune cell populations that are further classified into 8 clusters based on differential gene expression. (b) Summary of cell counts and percentage of each cluster of immune cell type in Th-MYCN/ALK^{F1178L} tumors treated with vehicle control and indisulam (25mg/kg) for 5 days. (c) Heatmap showing the expression of differential genes in each cluster of immune cells. (d) Bubble plot showing the gene set enrichment for each class of immune cells.

2. Considering that indisulam treatment in mice likely impacts tumor microenvironments, immune cells, stromal cells, fibroblasts, etc. by targeting RBM39, how genetic silencing of RBM39 in neuroblastoma cells affects lineage switches and immune cell recruitment in xenografts? Please discuss.

This is an excellent question. Cancer cells are able to alter their cell state to survive under detrimental stresses, including chemotherapy and targeted therapy. As this reviewer knows, neuroblastoma cells frequently undergo changes in morphology and transcriptomic state when exposed to therapeutic agents. Specifically, the loss of RBM39 may impose significant

survival pressure, prompting neuroblastoma cells to adapt by epigenetically reprogramming their cell state. This adaptation could enable cells to either increase RBM39 expression to compensate for its loss or, alternatively, upregulate survival pathways and/or downregulate the cell death programs.

Innate immune cells, such as NK cells, are known to mount rapid responses to damaged, infected, or stressed cells. Therefore, RBM39 loss may attract innate immune cells to the tumor, which becomes highly inflammatory due to cell damage. Additionally, RBM39 loss could lead to neoantigen production through splicing anomalies, potentially triggering a specific T cell-mediated immune response.

We have incorporated these points into the **Discussion** section (line 664-678, 697-699).

3. In Fig. 1C, the authors described that this is a GSEA analysis result of RNA-seq data of naïve tumors vs relapsed tumors. In Fig. 1D,H,I GSEA assay, is it also naïve tumors vs relapsed tumors?

The reviewer is right. Fig. 1C provides a summary of all GSEA results while Fig. 1D, H and I are extracted from 1C to demonstrate each specific gene set in our analysis.

Do naïve tumors vs relapsed tumors mean normalized naïve tumors RNA read counts/relapsed tumors RNA read counts?

Yes, they are normalized RNA read counts. However, counts are transformed to log₂ counts per million reads (log₂CPM). We have added this in the Figure legend for clarity.

How were the genes ranked? Was it based on T-score or fold change?

GSEA ranks the genes based on a measure of each gene's differential expression with respect to the two phenotypes (for example, tumor versus normal) or correlation with a continuous phenotype. Then the entire ranked list is used to assess how the genes of each gene set are distributed across the ranked list.

Here we used the Metric for ranking genes with the parameter “log₂_Ratio_of_classes”. We have added this in the figure legend.

For me, the labels of some of the GSEA graphs in this study are confusing. For instance, in Fig. 1H, although the authors claimed that the neural crest stem cell signature genes are positively enriched in the resistant population, which is true if the GSEA graph was generated using the ranked gene list of naïve tumors vs relapsed tumors, however, from the graph, it looks like the neural crest stem cell signature genes are negatively enriched in the resistant population since beneath the graph it was labeled with Naïve on the left, and labeled Resistant on the right, and NES score is negative. I feel it might be easier to understand if the authors labeled the graph differently, such as by labeling “ranked gene list (naïve tumors vs relapsed tumors)” beneath the graph. The other way to do it is to analyze the gene list of relapsed tumors vs naïve tumors, then the neural crest stem cell signature genes NES score will be positive. Please clarify.

As the reviewer suggested, we have labeled the GSEA as “ranked gene list (naïve tumors vs relapsed tumors)” in Figure 1D, H, I.

4. In Fig. 1J, how was the heatmap generated? Is it based on the normalized read counts, fold change, or z-score? Similarly, how was the heatmap generated for Fig. 2E? A scale bar is needed for this heatmap.

The Morpheus program generated z-score based interactive heatmap after log₂ transformed expression data. We have added this information in Figure legend Fig. 1J.

A similar method was used to generate heatmap for Fig 2E. We apologize for missing the scale bar during figure assembly and now it has been added.

5. On page 9, starting from line 298, the authors described many ADRN genes such as PHOX2A and MYCN as well as the RBM39 locus showed increased chromatin accessibility in the resistant cells (Fig. 3G). In the figure legend of Fig. 3G, it says naïve vs resistant, which indicates the normalized ATAC-seq read counts in naïve cells/in resistant cells. However, Fig. 3G showed that PHOX2A, SOX11 etc. are on the right side of the volcano plot, with positive fold change. From my understanding, this means that associated ATAC-seq peaks of these genes have higher signals (increased chromatin accessibility) in the naïve cells compared to the signals in the resistant cells, while PRRX2, GL2, SOX9 on the left side of the volcano plot have lower ATAC-seq signals in the naïve cells compared to the signals in the naïve cells. Please clarify the discrepancy between the main text and the graph.

We are sorry for the confusion.

In the figure legend, we have changed to “Volcano plot shows the differential peaks of ATAC-seq on or at nearby genes in resistant vs naïve tumors. The positive value indicates high ATAC-seq signals in the resistant tumors while the negative value indicates high ATAC-seq signals in the naïve tumors. “

6. Page 9, line 306, Figure 3G is a typo, which should be Figure 3H.

We have corrected this typo.

7. On page 9, starting from line 310, it says that GSEA analysis of the differential peaks near the annotated genes revealed that the 20q locus (where RBM39 resides) ranked on the top in the resistant cells while the c-MYC targets ranked on top in the naïve SK-N-AS cells (Figure 3K). However, from what I understand, Fig. 3K indicates that 20Q is high and MYC is low in naïve cells. Please clarify.

Sorry for the confusion. Actually, 20Q is high and MYC is low in resistant cells. We have changed the labeling to “resistant vs naïve” in Fig. 3K, and accordingly, in the figure legend.

8. In Fig. 3L, how was the heatmap generated? What does the color mean? Are they based

on H3K2ac signals or motif numbers? Please clarify with detailed information. Moreover, a scale bar is needed for this heatmap.

We apologize for the missing information.

The heatmap is generated based on the H3K27ac peak numbers that bear the corresponding motifs of transcription factor binding. We have added this information in the Figure legend, and the scale bar as well.

9. Fig. 7G,H, is the difference statistically significant? What is the p-value?

We have made a statistical comparison. While the total number of immune cells is significantly higher (Fig. 7G, now as Fig. 8C), the p value did not reach significance when each immune type is individually examined. This is probably due to the low sample size and variations of immune cell FACS sorting. We have added the p values in the Fig. 8C in our revision.

10. Discuss how RBM39 affects epigenetic reprogramming.

We have added the below sentences in Discussion (line 664-678):

One question is how RBM39 affects epigenetic reprogramming. It is a common survival mechanism for cancer cells to alter their cell state under detrimental stresses, such as chemotherapy and targeted therapy. For example, prostate and breast cancers often undergo epigenetic reprogramming in response to hormone and targeted therapies⁹⁶. Similarly, neuroblastoma cells frequently change their cell morphology and transcriptomic state when exposed to chemotherapeutic and differentiating agents. These dynamic cellular states are closely linked to changes in the activity of cell state-specific CRC TFs (i.e., PHOX2B and PRRX1)⁹⁷ and epigenetic remodelers (i.e., SWI/SNF)⁹⁸. As a key splicing regulator essential for neuroblastoma survival, RBM39 may be functionally connected to CRC TFs and epigenetic modifiers, given the tight coupling between splicing and gene transcription. Loss of RBM39 imposes significant survival pressure on neuroblastoma cells, forcing them to adapt by epigenetically reprogramming their cell state—either to restore RBM39 levels or to upregulate survival pathways and/or downregulate cell death programs. Our CRISPR screening revealed distinct epigenetic dependencies between indisulam-naïve and resistant neuroblastoma cells, which correlate with CRC TF dependency switching. This suggests that RBM39, CRC TFs, and epigenetic regulators act in concert to govern neuroblastoma cell fate

Reviewer #2 - Epigenetics, resistance (Remarks to the Author):

In the manuscript “RBM39 degrader invigorates nature killer cells to eradicate neuroblastoma despite cancer cell plasticity,” Singh et al. use a combination of genomics and Xenograft animal models to determine the mechanisms through which Neuroblastomas driven by hyperactivation of C-Myc/N-Myc develop resistance to the RBM39 degrader. In samples that are initially characterized by the andrenergic state (ADRN) they find that in response to prolonged exposure to the RBM39 degrader Indisulam, the tumor cells that have develop resistance upregulate expression of RBM39 and shift to a neural crest stem cell-like state and activate genes that are typically associated with the Mesenchymal (MES) state. They then show a Neuroblastoma line, SK-N-AS, that is associated with the MES state, when grown as a Xenograft and treated with indisulam, also acquires resistance, and these resistant cells downregulate the MES genes and upregulate genes associated with the ADRN state. Using CRISPR screens they show these “lineage-switching” events are coupled to changes in dependencies of the tumor cells on cell-type specific transcription factors, and chromatin remodelers, and they also identify general dependencies on kinases and and indisulam-resistant specific dependencies on druggable kinases. They go on to characterize the expression of GD2, a potential target of immuno-therapies, and find that GD2 remains expressed in the ADRN and MES states. By comparing tumor clearance in Xenograft models that are depleted for T, B and NK cells (NSG) to Xenograft models that are depleted for only T and B cells (Rag2^{-/-}), they suggest that NK cells are the critical mediator of tumor clearance, and that indisulam may be aiding in this response by inducing inflammation within the tumor. They then show that indisulam can be used in combination with GD2-directed immunotherapy to induce complete remission of MYC driven neuroblastomas. The development of preclinical models to show indisulam induces NK-cell-mediated tumor clearance that works in combination with GD2 inhibitors, make this manuscript of significant interest to the Neuroblastoma field, and provide an interesting example of how splicing inhibitors can be incorporated into other therapeutic regimens. However, the mechanism of action of indisulam in this context requires further clarification, and would benefit from a more uniform and complete analysis of the genomics data, and more complete characterization of the immune cells in these preclinical models.

We highly appreciate this reviewer for his/her positive comments and great summary of our study. We have performed additional experiments as suggested which we believe have significantly improved our study.

(1) The lineage-switching mechanism of indisulam resistance that is proposed is often supported by a seemingly random choice of marker genes, “cherry picked” by the authors to support their hypothesis. In the MYCN/ALKF1178L and SK-N-AS models, the terms the “Neural crest stem cell”, “NBL_MES”, and “ADRN” come out of an unbiased GSEA. Is this sufficient to meet the definition of “lineage-switch” in this model? If so, the SJNB14 model fails to meet this definition, and weakens the point. To be taken seriously, the authors must

do a more thorough, convincing, and uniform job of establishing the lineage identity of the cells before and after lineage switching.

We agreed with this reviewer that “lineage switching” is a strong word. We now have changed this term to “cell state change”, which might be more appropriate.

(2) Related to the first point; using single-cell genomics of healthy tissues to compare marker gene expression can help to establish cell-type identities, but in Figure 1K the authors show only the expression of Sox10, and this does little to strengthen their conclusion. Can the authors color this UMAP according to the composite or average score of genes downregulated in the resistant cells (presumably the ADRN signature) and the genes upregulated in the resistant cells (The MES signature).

The UMAP in Figure 1K projected Sox10 into the SCP population and the expression was not from our own experimental data. We now have performed scRNA-seq using the Th-MYCN/ALK^{F1178L} model and demonstrated that the SCP population was upregulated after indisulam treatment. This has been added in our revision as Fig. 8A, B, and Supplementary Figure 8,9. Please see details (line 535-569).

(3) Currently, every figure highlights a different panel of marker genes, and the data is presented in different formats making it very difficult to compare one model to another. As one example: Figure 1J shows a panel of 8 SCP genes that are upregulated in the resistant cells. Figure 2E shows a panel of ~30 genes. The only gene shown in both figure panels is S100B and yet we do believe these two Neuroblastoma models are engaging in a similar lineage-switching mechanism of resistance?

This is a very insightful question. We tried to find if both models share common gene signatures including the SCP gene signature after tumors developed resistance to indisulam. However, the upregulated gene signatures in both models were barely overlapping. This could be due to multiple reasons. First, neuroblastoma is a very heterogeneous disease and could occur at different developmental stages of neural crest cell lineage. Neural crest cells are a transient stem cell population that develops into different cell lineages under different development cues. Second, the species specificity (mouse vs human) could also make a difference. Third, the tumor drivers are different. While the tumor model in Figure 1 was primarily driven by transgenic MYCN and ALK^{F1178L}, the model in Figure

2 was a PDX model with MYCN amplification derived from a relapsed patient that received intensive chemotherapy. Nevertheless, both models exhibited gene signatures that can be projected to some stages of neural crest lineage during development, as characterized by Schwann cell precursor in Fig. 1 and melanocytic markers in Fig. 2. These data support the lineage plasticity of neuroblastoma cells in therapy resistance, and this may be associated with differentiation and de-differentiation of neural crest lineages. Dr. John Maris group recently characterized chemotherapy-resistant high-risk neuroblastoma persister cells and found that these persisters were not a uniform population, rather, they were composed of 4 different populations and exhibited distinct gene signatures (Cancer Discov. 2024 Dec 2;14(12):2387-2406). Interestingly, this study identified some persister tumors had elevated Schwann cell signatures while some showed elevated neuronal signatures, similar as seen in Figure 1 (Th-MYCN/ALK^{F117BL} genetic model) and Figure 3 (c-MYC overexpressing human SKNAS model). This study further suggested that the neuroblastoma cells can go to different directions after developing therapy resistance.

We have included this discussion in our manuscript (line 304-319).

(4) Related to point 3; In figure 4 the authors show the key dependencies that increase in the SK-N-AS model after the development of resistance are HAND2, PHOX2B and TBX2. Yet, these genes are totally absent from Figure 3 which shows the changes in gene expression in the SK-N-AS model during the development of resistance and transition to an MES state. These kind of incongruencies make this paper extremely challenging to follow and critically assess.

Again, this reviewer has raised a great point. We also felt surprised that HAND2, PHOX2B and TBX2 did not show significant upregulation in the resistant tumors. However, regardless of cell state (ADRN vs MES), we found that nearly all neuroblastoma lines expressed HAND2 and PHOX2B. This raised one hypothesis that cells may lean on ADRN CRC TFs (i.e., PHOX2B and HAND2) for survival when they switch their cell state from MES to ADRN, even the expression levels of PHOX2B and HAND2 showed no significant upregulation. Additionally, the transcriptional activity of PHOX2B, HAND2 and TBX2 could be significantly enhanced during the cell state transition from MES to ADRN. However, we acknowledged that further work needs to be done to understand the underlying mechanism.

We have added this in the Discussion (line 642-648).

(5) Due to the previous issues, I am left with the impression that lineage switching in this context has no formal definition, and thus can be used to mean anything the authors choose. Please explain why a shift from ADRN does > MES state promote resistance to indisulam and a shift from an MES > ADRN state also promote resistance to indisulam. Is one state more resistant or not? Are the authors suggesting that it is the act of lineage switching that promotes resistance, and that the directionality of the lineage switch is inconsequential. If so, the authors must provide a rationale that is supported by evidence.

Yes, we believe that it is the act of lineage switching that promotes resistance, and that the directionality of the lineage switch is inconsequential or less important. Please see our response to the comment 3 from this reviewer,

(6) The argument that the immune clearance is driven by NK cells is based primarily in the difference between RAG2^{-/-} and NSG mice. In addition to presence of NK cells in RAG2^{-/-} mice, these models could have many other differences that contribute to apparent changes in tumor responses to indisulam. Can the authors provide additional conclusive evidence the NK cells make the difference. For example using immune reconstitution experiments etc.

To address this reviewer's concern, we have performed additional experiments:

- (1) We have expanded our study by using the human SKNAS xenografts implanted into RAG2^{-/-} mice. Again, indisulam treatment led to durable and complete responses in this model (**Figure 7D, E**). However, all SKNAS xenografts were relapsed in NSG mice (**Supplementary Fig. 5A**).
- (2) We have directly assessed whether NK cells can be activated by indisulam to enhance neuroblastoma cell killing. We utilized a co-culture system by mixing NK92 cells (a human lymphoma-derived cell line phenotypically similar to activated NK cells) with SKNAS cells. Both cell types were pretreated with a suboptimal dose of indisulam (150nM, to avoid inducing cell death) for 24 hours and co-cultured at a 3:1 effector-to-target (E:T) ratio for 12 hours(**Figure 7G**). SKNAS cell viability was assessed by measuring apoptosis through caspase 3/7 activity (**Figure 7H**). Interestingly, we found that pretreatment of SKNAS cells with indisulam did not enhance NK92-mediated cell killing. In contrast, pretreatment of NK92 cells with indisulam significantly increased their cytotoxic activity against SKNAS cells (**Figure 7H**). These data demonstrated that indisulam can directly activates NK cells to kill neuroblastoma cells.

(3) We have tested the efficacy of combination therapy (indisulam + anti-GD2) in two additional neuroblastoma models: Dbh-iCre:LSL-MYCN model (C57BL6 background) and a temozolomide-resistant Th-MYCN model (129 X1/SvJ background). Both models exhibited complete and durable responses to the combination therapy while the monotherapy only achieved transient responses (**Figure 8F-I**). These data indirectly support the NK cell involvement as the NK cell mediated ADCC is thought to be the main anticancer mechanism of anti-GD2 immunotherapy.

(4) However, we agree with this reviewer that the difference between NSG and RAG2^{-/-} mice could contribute the efficacy of indisulam. We have tried to deplete NK cells from RAG2^{-/-} mice to see if it could induce disease relapse. Unfortunately, we observed disease relapse in only one out of four mice. We were not confident if NK cells were completely deleted and decided not to include this data. However, we do believe NK cells were involved in indisulam-mediated cancer cell killing, at least based on the co-culture data.

(7) How can the authors be sure that indisulam is acting on the Neuroblastoma cells and not the NK cells?

We sincerely appreciate this reviewer for asking this question and the suggestion for reconstitution experiment in comment 6. As we have shown in Figure 7G, H, it seemed that indisulam activates NK cells for cancer cell killing, but not the other way. We have obtained an NK cell specific Cre and floxed RBM39 mice, which will allow us to further differentiate the action of indisulam. Considering there are still remaining questions to be answered for the exact roles of NK cells and/or other types of immune cells, we have replaced NK cells with innate immunity in our title.

Minor points:

- Line 118 reads “that tumor cells acquired MES gene features when developed resistance.” Please reword to something like: “the resistant tumor cells acquired MES gene features.”

Thanks for this suggestion. We have made the change.

- Line 183 reads “the we projected the expression of Sox10...” this should read “then we colored the UMAP projection by the expression of Sox10”.

We have made the change.

- Line 211-212 reads “It is known that SCP is a melanocyte progenitor.” This should read, “It is know that SCPs are melanocyte progenitors.”

We have made the change.

- Line 310 reads “GSEA analysis of the differential peaks.” Should read “GSEA of the differential peaks.”

We have made the change.

- Line 315 reads “the RBM39 locus but not the RBM23 locus showed a greater increase in H3K27ac peak...” Should read “the RBM39 locus showed a greater increase in H3K27ac than the RBM23.”

We have made the change.

- Line 345 refers to a gene SMARCE1, but there is no SMARCE1.

Please see SMARCE1 in the right panel of Fig. 4G.

- Line 360 “We identified 43, 31 and 25 kinases that were essential for either naïve...” only three numbers are listed but four states are referred to.

This was for the description of Venn diagram (Fig. 5B), and we also mentioned “and 13 of them were commonly shared”.

- Line 531 “... that drive high rate ...” please reword to “that drive high rates”

We have made the correction.

Reviewer #3 - RBM39 degraders (Remarks to the Author):

This is an interesting manuscript which identifies that acquired resistance to indisulam in neuroblastoma may be due to a change in phenotypic cell states where apparently the new state amplifies RBM39 resulting in less degradation in resistant cells. Additionally, the authors suggest that RBM39 degradation may promote the activity of anti-GM2 monoclonal antibody therapy. They suggest that the therapeutic activity of RBM39 degraders may be related to NK cell mediated killing. Overall, the manuscript is novel but the data around NK cells is quite modest and not currently well supported by the data presented in the manuscript.

We sincerely thank this reviewer for his/her positive and constructive comments. As this reviewer may notice, our focus in this study was not NK cells as my laboratory lacks expertise in this field. However, we have performed additional experiments to support the hypothesis that NK cells are involved in indisulam-mediated cancer cell killing by performing additional experiments. Please see our response to each comment from this reviewer.

-In the first section of the Results and Figure 1, it is unclear if RBM39 protein level is actually increased in resistant tumors and if RBM39 is degraded in resistant tumors. Is DCAF15 still expressed in the resistant tumors throughout the manuscript?

To answer this reviewer's question, we performed western blot to detect the expression of RBM39 and DCAF15 from tumor samples. Surprisingly, the RBM39 protein was actually downregulated to some degree, which was in contrast to its mRNA expression. This data suggest that cells strived to survive by producing more mRNAs to compensate for the RBM39 protein degradation. We have added this data as Fig. 1F in our revision.

-It is unclear if the RBM39 degrader resistant cells still rely on RBM39 or not. It appears as though resistance to these agents may be due to amplification of RBM39 which prevents effective RBM39 protein downregulation. The authors should clarify this point more in the manuscript.

To answer if the resistant cells are still dependent on RBM39, we introduced DACF15 into the indisulam-resistant SKNAS cells, which led to a remarkable degradation of RBM39 and cell death. These data indicate that the indisulam-resistant cells are still dependent on RBM39, at least in this tested model. We have added the data as Supplemental Figure 5D, E, F,G.

-The authors claim to have tested the impact of RBM39 degradation versus RBM39 knockdown when referring to Figure S3C but Figure S3C only shows data from RBM39 degrader treatment but this figure only shows analysis of RBM39 degrader treated cells. This is an important point because it is not clear if the degree of drug-induced splicing alterations would be the same in the resistant cells treated with drug as when RBM39 is genetically suppressed.

We totally agree with this reviewer but unfortunately we do not have the genetic data of RBM39 knockdown in this model, partly due to the difficulty in genetically manipulating patient-derived xenografts. We therefore cited our previously published data which had

matched shRNA and indisulam treatment in two neuroblastoma cell lines, which showed a good correlation in splicing changes (PMID: 34788094).

-It is not clear if the anti-tumor response to indisulam requires NK cells. This should be directly tested. The fact that indisulam treatment showed little efficacy in NSG mice but had efficacy in C57BL/6 and Rag2^{-/-} mice does not necessarily indicate involvement in NK cell killing as NSG mice have macrophages as well.

To address this reviewer's concern, we have performed additional experiments:

- (1) We have expanded our study by using the human SKNAS xenografts implanted into RAG2^{-/-} mice. Again, indisulam treatment led to durable and complete responses in this model (Figure 7D, E). However, all SKNAS xenografts were relapsed in NSG mice (Supplementary Figure 5A).
- (2) We have directly assessed whether NK cells can be activated by indisulam to enhance neuroblastoma cell killing. We utilized a co-culture system by mixing NK92 cells (a human lymphoma-derived cell line phenotypically similar to activated NK cells) with SKNAS cells. Both cell types were pretreated with a suboptimal dose of indisulam (150nM, to avoid inducing cell death) for 24 hours and co-cultured at a 3:1 effector-to-target (E:T) ratio for 12 hours (Figure 7G). SKNAS cell viability was assessed by measuring apoptosis through caspase 3/7 activity (Figure 7H). Interestingly, we found that pretreatment of SKNAS cells with indisulam did not enhance NK92-mediated cell killing. In contrast, pretreatment of NK92 cells with indisulam significantly increased their cytotoxic activity against SKNAS cells (Figure 7H). These data demonstrated that indisulam can directly activate NK cells to kill neuroblastoma cells.

(3) We have tested the efficacy of combination therapy (indisulam + anti-GD2) in two additional neuroblastoma models: Dbh-iCre:LSL-MYCN model (C57BL6 background) and a temozolomide-resistant Th-MYCN model (129 X1/SvJ background). Both models exhibited complete and durable responses to the combination therapy while the monotherapy only achieved transient responses (**Figure 8F-I**). These data indirectly support the NK cell involvement as the NK cell mediated ADCC is thought to be the main anticancer mechanism of anti-GD2 immunotherapy in neuroblastoma treatment.

(4) However, we agreed with this reviewer that the difference between NSG and RAG2^{-/-} mice could contribute the efficacy of indisulam. We have tried to deplete NK cells from RAG2^{-/-} mice to see if it could induce disease relapse. Unfortunately, we observed disease relapse in only one out of four mice. We were not confident with this result because we were

not sure if NK cells were completely depleted during the treatment course and therefore decided not to include this data in our revision. However, we do believe NK cells were involved in indisulam-mediated cancer cell killing, at least based on the co-culture data.

(5) This reviewer also mentioned macrophage, which led us to hypothesize that indisulam may activate macrophages to exert the cancer cell killing. Tumors resist macrophage phagocytosis through expression of the checkpoint molecule CD47 ('Don't eat me' signal).

We therefore tested if the combination of indisulam with anti-CD47 antibody could enhance the efficacy in a MYCN syngeneic neuroblastoma model that we recently developed (Figure 7F). Surprisingly, anti-CD47 seemed to block, to some degree, the indisulam effect, albeit an effect that was not statistically significant. We therefore terminated this experiment. While we believe there are a lot of questions that remain to be answered, we decided to wrap up the story since we have been revising this manuscript for over one year.

-Are NK cells required for response to anti-GD2 mAb?

Based on the published in vitro, preclinical and clinical studies, NK cells play a critical role in anti-GD2-mediated anticancer effect (PMID: 1988079, PMID: 10663607, PMID: 20935224, PMID: 22863621, PMID: 27069083, PMID: 28972044, PMID: 29327110, PMID: 36822669, PMID: 30232225). However, the in vitro studies showed that granulocytes such as neutrophils also had cancer cell killing effect at a higher E:T ratio (5:1 to 50:1) in the presence of anti-GD2 antibody (PMID: 1988079, PMID: 10663607). However, the neutrophil effect was not noticed when neutrophils and neuroblastoma cells were mixed at a lower E:T ratio (PMID:10663607). Lack of evidence for anti-GD2 treatment in NSG mice (which have functional granulocytes) suggests that neutrophils may not play an important role in mediating anti-GD2 activity.

It is also reported that anti-GD2 can trigger complement activation by the C1q-antibody interaction, leading to complement lysis of neuroblastoma cells (PMID: 16288049). However, the anti-GD2 (Hu14.18K322A) antibody used in this study has a mutation on K322 that abrogates its interaction with complement (in order to reduce the pain caused by complement effect on neurons). Therefore, this mechanism may be negligible in our study.

We have added the references related to anti-GD2 mediated NK effect in Discussion (line 693).

-What is the impact of indisulam treatment in vivo on NK cell numbers and phenotypes?

We have performed scRNA-seq analysis using the Th-MYCN/ALKF1178L genetic model treated with indisulam. Indeed, indisulam treatment induced enrichment of immune cells in tumors after indisulam treatment (Supplementary Figure 10B). However, due to the low absolute number of these immune cells (particularly NK and T cells) it is very challenging to further define the subclusters for each group. We are currently expanding the study as a new project and hopefully we can report our results in the near future.

cluster	cell type	Ctrl (cell count)	Ind (cell count)	Ctrl(%)	Ind(%)
c1	NK	24	39	0.09%	0.61%
c2	T	14	61	0.05%	0.96%
c3	B	10	19	0.04%	0.03%
c4	DC	24	85	0.09%	1.30%
c5	Macrophage	85	1330	0.33%	20.92%
c6	Macrophage	83	861	0.33%	13.54%
c7	Macrophage	170	97	0.67%	1.53%
c8	Macrophage	78	0	0.31%	0.00%

-The analysis of the immune microenvironment with indisulam and anti-GD2 mAB is quite limited (the authors only investigated a small number of immune cell markers by conventional FACS). A more detailed analysis of types of innate and adaptive immune cells (by more comprehensive FACS and/or RNA-sequencing analyses) would be very helpful.

We acknowledge this limitation and also please see our response to the above comment.

Considering there are still remaining questions to be answered for the exact roles of NK cells and/or other types of immune cells, we have replaced NK cells with innate immunity in our title.

Reviewer #4 - NK cells, immunotherapy (Remarks to the Author):

This is an interesting and data rich paper.

As an immunologist, the data linking Indisulam to NK cell mediated rejection of neuroblastoma is intriguing.

However, the extensive characterisation of the models (figures 1-6) is not matched at all by the depth of the immunological responses shown in Figure 7.

We sincerely thank this reviewer for his/her positive and constructive comments and suggestions. While Figures 1-6 could serve as a whole story, we eventually decided to add Figure 7 in our manuscript. We hope these data could encourage and convince the neuroblastoma community that the combination of indisulam with anti-GD2 antibody is a promising approach for the high-risk patients. The publication of these data will help us to move forward for a clinical trial. We are currently collaborating with Eisai, a pharmaceutical company who will provide us GMP-grade drug for clinical trials. However, we acknowledge that Figure 7 was premature and lacked a full understanding of NK cell function, this is partly because my laboratory lacks expertise in this field. We highly appreciate this reviewer's insightful suggestions including the in vitro assay approach and CD16 deficient mice, which helped us a lot for testing our hypothesis and preparing for future experiments. While we cannot accomplish all experiments suggested due to time pressure and limited expertise, we have performed additional experiments to support the hypothesis, at least partly, that NK cells are indeed involved in indisulam-mediated cancer cell killing.

main problems with Figure 7.

1. The mouse strains used have different backgrounds (C57Bl6 and RAG KO are the same but the NSG mice are normally CB17). The genetic background of the mouse tumour used in this study is not evident to me but it clearly cannot match two different strains.

We acknowledge that the genetic difference among these strains could cause the differential response. We have added this in our revision (line 512-513).

2. Data in figure 7B and C looks promising (strain caveat aside). For rigour, I suggest that Ab mediated NK cell depletion from the C57Bl/6 mice (anti-NK1.1) is performed, alongside NK cell adoptive therapy from C57bl/6 mice into NSG mice. Adoptive transfer experiments might also help to evaluate whether Indisulam effects are wholly tumour intrinsic or also has effects on NK cells etc.

Using a human neuroblastoma model (SKNAS) implanted in RAG2^{-/-} mice, we have tried to deplete NK cells to see if it could induce disease relapse. Unfortunately, we observed disease relapse in only one out of four mice although no mouse had disease relapse in mice without NK cell depletion. We were not confident with this result

because we were not sure if NK cells were completely depleted during the therapy course in each mouse and therefore decided not to include this data in our revision.

NK cell adoptive therapy is a great idea. Unfortunately, we lack expertise to isolate NK cells for the experiment. Instead, we have taken other approaches as listed below to test if NK cells are involved in indisulam-mediated cancer cell killing:

- (1) We have expanded our study by using the human SKNAS xenografts implanted into RAG2^{-/-} mice. Again, indisulam treatment led to durable and complete responses in this model (Figure 7D, E). However, all SKNAS xenografts were relapsed in NSG mice (Supplementary Figure 5A). One out of 4 RAG2^{-/-} mice had disease relapse when they were administered with anti-NK1.1 (please see above).
- (2) We have directly assessed whether NK cells can be activated by indisulam to enhance neuroblastoma cell killing. We utilized a co-culture system by mixing NK92 cells (a human lymphoma-derived cell line phenotypically similar to activated NK cells) with SKNAS cells. Both cell types were pretreated with a suboptimal dose of indisulam (150nM, to avoid inducing cell death) for 24 hours and co-cultured at a 3:1 effector-to-target (E:T) ratio for 12 hours (Figure 7G). SKNAS cell viability was assessed by measuring apoptosis through caspase 3/7 activity (Figure 7H). Interestingly, we found that pretreatment of SKNAS cells with indisulam did not enhance NK92-mediated cell killing. In contrast, pretreatment of NK92 cells with indisulam significantly increased their cytotoxic activity against SKNAS cells (Figure 7H). These data demonstrated that indisulam can directly activate NK cells to kill neuroblastoma cells.

(3) We have tested the efficacy of combination therapy (indisulam + anti-GD2) in two additional neuroblastoma models: Dbh-iCre:LSL-MYCN model (C57BL6 background) and a temozolomide-resistant Th-MYCN model (129 X1/SvJ background). Both models exhibited complete and durable responses to the combination therapy while the monotherapy only achieved transient responses (**Figure 8F-I**). These data indirectly support the NK cell involvement as the NK cell mediated ADCC is thought to be the main anticancer mechanism of anti-GD2 immunotherapy in neuroblastoma treatment.

3. The numbers of replicates and absence of stats in figs 7G and H is a concern. Are the changes in cell infiltration statistically significant (NK cells in 7H seems to have substantial variation in the three replicates).

We have calculated the p values and they are not significant probably due to the low mouse number. We have added the p values in the figure (now as Figure 8C).

4. In vitro experiments would enhance the conclusions; does the drug enhance killing by purified NK cells in vitro? It might be possible to perform in vitro experiments (or adoptive transfers) using NK cells from mice lacking CD16 to demonstrate the pathway of GD2 Ab recognition as suggested in Figure 8.

We have used an in vitro approach (NK92) as this reviewer suggested. Please see the above response to this reviewer's comment.

We appreciate the suggestion to use mice lacking CD16, which is a great idea. Unfortunately, Jax Laboratory does not have live mice for this model. Considering the prolonged time from rederiving the mice to breeding for experiments, we did not explore this approach immediately. However, we will get this mouse strain rederived from JAX to test the idea as this reviewer suggested.

Considering there are still remaining questions to be answered for the exact roles of NK cells and/or other types of immune cells, we have replaced NK cells with innate immunity in our title.

I found the earlier part of the paper hard to follow. Admittedly this is outside of my core expertise. Nevertheless, it lacked detail in terms of rationale and approaches used (sacrificed for brevity). It was hard to tell how many tumours were analysed in places (e.g. figure 2a). I struggled with identifying the core data revealed by the different tumour models (e.g. different markers and pathways identified in different models). Is all of this data really necessary to get to Figure 7, which the title and abstract flag up.

We are sorry for the difficulty for this reviewer to follow. We have tried our best to add enough details in our revision. We agreed that this is a complex story that consisted of different neuroblastoma models that behaved differently when they developed drug resistance. However, this was also the point we were trying to make that the cellular plasticity of neuroblastoma cells can direct them to distinct developmental stages under the therapeutic pressure. Regardless of the cellular plasticity, the host immune system can be leveraged to eradicate the residue disease that is responsible for cancer relapse.

Again, stats are missing from many key data. For example, traces showing read counts over exons are attractive but are qualitative only without statistical testing.

Due to the low sample number ($n=2$) per group, the p value was not statistically significant. We have added the \log_2 fold change =0.73 in the manuscript.

Reviewer #1 (Remarks to the Author):

The authors have satisfactorily addressed my comments and concerns.

We highly thank this reviewer for his/her time and effort, and insightful comments that greatly improved our study.

Reviewer #2 (Remarks to the Author):

We thank the reviewer for their continued engagement and valuable suggestions, which have helped us improve the manuscript. Below, we respond to each point raised.

Comment: The manuscript currently suffers from poor organization of the data presentation primarily in the first three figures... suggest consolidating and focusing on the single-cell data.

Response: We appreciate the suggestion and understand the desire for a more concise presentation. However, we believe that the bulk RNA-seq gene expression data presented in Figures 1–3 are essential for illustrating the cellular plasticity of neuroblastoma. As noted in our previous response to this reviewer during the first-round revision, our central premise is that neuroblastoma cells exhibit dynamic plasticity, shifting toward distinct developmental states under therapeutic pressure. Importantly, despite this plasticity, the host immune system can still be harnessed to eliminate residual disease responsible for relapse.

We have retained the SOX10 panel in Figure 1, as we believe it provides valuable developmental context by highlighting the neural crest lineage, the cell of origin for neuroblastoma, which may be especially helpful for readers less familiar with neuroblastoma biology.

Comment: The results from bulk RNA-seq and ATAC-seq are inconsistent across models; this weakens the claim of cell-state changes as a resistance mechanism.

Response: In our first-round revision, we have addressed these differences across models reflecting context-dependent plasticity and tumor evolution under treatment pressure, rather than a single conserved resistance program. This heterogeneity itself underscores the adaptability of neuroblastoma tumors to therapeutic stress.

Comment: Figure labels and text are still unclear in places (e.g., “Indisulam vs copy”, text size in Figs. 3G and 3H, unclear left/right groups).

Response: We have clarified the labels by changing “indisulam vs copy” to “indisulam vs gene copy” in Fig 1G, and increased font size and improved panel readability in Figures 3G and 3H.

Comment: “Comprehensively defined the transcriptomic and epigenetic map...” is overstated.

Response: We have removed “Comprehensively”.

Comment: Several of the reviewers had issues with the genetic models used to claim the NK cells are the key mediator tumor clearance. The data is interesting, but the authors must revise the text to be transparent about the limitations of these models. The addition of the in vitro cell killing assay helps to make the point that NK cells play an important role and are stimulated by indisulam.

Response: We acknowledged the genetic differences among these models in our first-round revision. We appreciate the reviewer's comment: "The addition of the in vitro cell killing assay helps to make the point that NK cells play an important role and are stimulated by indisulam." In the revised manuscript, we have further addressed the limitations of these models in the Discussion section, stating: "However, the models we used have different genetic backgrounds, which may influence the interpretation of NK cell function in indisulam-mediated cancer cell killing. In future studies, we will validate these findings using models with matched genetic backgrounds." (line 726-729)

Reviewer #3 (Remarks to the Author):

The authors have addressed my initial comments and questions.

We highly thank this reviewer for his/her time and effort, and insightful comments that greatly improved our study.

Reviewer #4 (Remarks to the Author):

My expertise lies in NK cell biology and I have concentrated my review on these aspects. The authors have strengthened their arguments for the role of NK cells in the response to indisulam therapy and neuroblastoma and my concerns have been addressed.

The manuscript is interesting, a detailed study has been undertaken and the findings have implications for neuroblastoma treatment. I recommend that the article be published.

We highly thank this reviewer for his/her time and effort, and insightful comments that greatly improved our study.